# Demixing fluorescence time traces transmitted by multimode fibers

Caio Vaz Rimoli ®[1,2], Claudio Moretti ®[1], Fernando Soldevila[1], Enora Brémont ®[2], Cathie Ventalon ®[2,3] ✉ & Sylvain Gigan ®[1,3] ✉

Optical methods based on thin multimode fibers (MMFs) are promising tools for measuring neuronal activity in deep brain regions of freely moving mice thanks to their small diameter. However, current methods are limited: while fiber photometry provides only ensemble activity, imaging techniques using of long multimode fibers are very sensitive to bending and have not been applied to unrestrained rodents yet. Here, we demonstrate the fundamentals of a new approach using a short MMF coupled to a miniscope. In proof-of-principle in vitro experiments, we disentangled spatio-temporal fluorescence signals from multiple fluorescent sources transmitted by a thin (200 μm) and short (8 mm) MMF, using a general unconstrained non-negative matrix factorization algorithm directly on the raw video data. Furthermore, we show that low-cost open-source miniscopes have sufficient sensitivity to image the same fluorescence patterns seen in our proof-of-principle experiment, suggesting a new avenue for novel minimally invasive deep brain studies using multimode fibers in freely behaving mice.

Fluorescence-based techniques are providing researchers with different ways to collect functional readouts from neuronal activity in the brain[1–10]. However, measuring neuronal activity at depths greater than 1 mm is still challenging mainly due to issues resulting from light scattering, especially in prominent paradigms such as freely behaving animals[11–15]. To address this problem, neuronal microendoscopy methods have emerged as complementary alternatives to linear and nonlinear fluorescence microscopy techniques for studying neuronal activity in deep brain regions using genetically encoded calcium indicators (GECI)[14–17]. Among these methods, conventional microendoscopic methods that use a single gradient index (GRIN) lens optics[17–20], as well as fiber photometry recordings using multimode fiber (MMF)[16,21–24], have been successfully used to obtain functional neuronal activity signals in deep brain regions in freely behaving mice[16,23–25]. Nonetheless, direct imaging techniques and fiber photometry approaches bring peculiar tradeoffs in terms of spatial and temporal discerning capabilities[11,14,26,27]. On one hand, albeit GRIN lens microendoscopy retrieves calcium transients with cellular resolution, it

demands a somewhat invasive surgical procedure to implant the GRIN lens into the mouse brain. Commercial GRIN lenses are relatively thick (≥500 μm), and oftentimes they necessitate the removal of a significant amount of brain tissue to effectively conduct the experiment[13,27]. On the other hand, the use of thin multimode fibers (<500 μm diameter) in photometric recordings, as well as in optogenetics experiments, has a significantly less invasive surgical procedure, which does not require any brain tissue removal, but only a careful penetration of the thin fiber through the mouse brain[6,16,26]. It is known that the implantation of multiple multimode fibers (up to a maximum of 48 fibers[28]) to optogenetically control and/or photometrically probe different regions in freely-behaving mouse brains is already a reality in neuroscience labs[27–29]. However, the light wavefront propagating inside multimode fiber gets spatio-temporally scrambled due to multimodal mixing (internal scattering)[30–32]. Generally, that is not a limitation for delivering light (optogenetics) to an ensemble of neurons in a given depth (unless one wants to probe specific neurons within the fiber field of view, FoV), but it poses a challenge for fiber photometry methods

[1]Laboratoire Kastler Brossel, ENS-Université PSL, CNRS, Sorbonne Université, Collège de France, 24 Rue Lhomond, Paris F-75005, France. [2]Institut de Biologie de l'ENS (IBENS), Département de biologie, École normale supérieure, CNRS, INSERM, Université PSL, 75005 Paris, France. [3]These authors jointly supervised this work: Cathie Ventalon, Sylvain Gigan. ✉e-mail: cathie.ventalon@bio.ens.psl.eu; sylvain.gigan@lkb.ens.fr

which limits the technique's potential to resolve (demix) time traces from individual neurons. Consequently, fiber photometry time traces coming from a whole population of neurons transmitted through MMF are ensemble integrated during detection, and therefore, fast single-pixel detectors are frequently chosen to optimize the detection speed and sensitivity[16,27]. While the use of fast scientific Complementary Metal-Oxide-Semiconductor (sCMOS) cameras to simultaneously probe multiple fiber photometric signals has been demonstrated[27,29,33–36], the mixing between all the spatial patterns transmitted by each MMF prevented the individual retrieval of each neuron time trace[27]. Recently, researchers have developed novel techniques that utilize the deterministic nature of the multimode fiber transmission matrix (TM) to perform bioimaging[31,32,37–48]. These approaches have enabled the acquisition of diffraction-limited images of fluorescently labeled brain structures and neuronal activity, even in deep brain regions of head-fixed mice, using a multimode fiber microendoscope[32,47,48]. To achieve this, however, an extensive characterization of MMFs transmission properties is necessary, ideally taking into account TM changes whenever the MMF fiber is bending or changing its transmission properties during an experiment, as well discussed in previous research[37,48,49]. Consequently, while these techniques provide a minimally invasive method to obtain diffraction-limited resolution in deep brain regions, they are complex to implement and require a wavefront shaping device (e.g., spatial light modulator, SLM) to compensate for the fluorescence randomized wavefronts through a lengthy calibration procedure. Moreover, the calibration can be even more complex if the experiment is not performed in head-fixed mice, but in freely behaving mice, such as those in long-term social behavior studies[32,37,47,48]. Finally, the use of spatial light modulators and complex distal optical devices poses an extra challenge in future use in miniaturized wireless systems.

In this article, we propose a novel approach to perform minimally invasive fiber photometry experiments disentangling single-source time traces transmitted by short and thin multimode fibers (≈200 μm diameter and <10 mm length). We take advantage of the short length of the multimode fibers, which makes them naturally rigid (bending resistant) and therefore suitable to be used in long-term freely-moving mice neuroscience experiments. Our method involves the demixing of fluorescence spatiotemporal signals by applying a single post-processing step on the recorded video data of 2D scattered fluorescence patterns transmitted by the fiber. By substituting the bucket detector with a camera (i.e., a pixelated detector such as CMOS sensor), we can profit from using the spatial information of the fluorescence patterns transmitted by the multimode fiber, enabling single-source temporal activity resolution. Analysis of the recorded video is performed employing a simple unconstrained Non-negative Matrix Factorization (NMF) algorithm that separates each spatial scattering pattern component with its corresponding temporal trace (singular trace)[49–52]. With this approach, we show that it is possible to extract single-source time traces in fiber photometry without the need to perform any complicated calibration procedure. This work builds up on previous work from some of the authors, which showed that it is possible to spatiotemporally demix fluorescence scattering patterns (*speckles*) transmitted through a highly scattering media (e.g., mouse skull) by using a NMF algorithm[51,52]. This algorithm relies on the premise that the input data matrix only contains non-negative values, and it has been used to decompose datasets into their representative parts or components[50–61]. Here, we apply the same algorithm to the video data from scattering patterns that are characteristic of the light transmitted through short multimode fibers of the same length as the ones implanted in fiber photometry[27,62–64]. It is well known that multimode fibers randomize the fluorescence wavefront propagating within it, acting as a scattering media over the fiber length due to multimodal mixing. Nevertheless, the multimode fibers (<10 mm) typically implanted in living mice for chronic behavioral experiments are too

short to generate a fully evolved speckle wavefront[62,64]. In fact, the light wavefront that emerges from such short fibers displays a very peculiar spatial distribution of light, which is structurally mixed, but not fully spatially randomized/sparse[62,63], and whose shape depends mostly on the multimode fiber core geometry[65]. Here, we call these short MMF patterns as scattering fingerprints.

In the present work, we design in vitro proof-of-principle experiments and we show that a simple unconstrained NMF algorithm can disentangle scattering fingerprints transmitted by short MMFs and retrieve the corresponding time traces. We demonstrate that one may now temporally resolve and count the number of sources with singular time traces transmitted by short, minimally invasive MMFs. Thus, the results of this paper consist of a proof of concept on how to obtain individual time trace resolution in fiber photometry methods. Starting with a simple proof-of-principle experiment with only a few fluorescent beads located right below the fiber, we progressively validate our approach towards more realistic conditions, such as demixing fluorescence signal from tens of bead sources buried behind a scattering media (plastic paraffin: Parafilm M®) including a component for neuropil activity, and by selectively probing a few structurally Gad-eGFP labeled neurons in a 50 μm fixed brain slice with literature-available time traces to mimic neuronal activity. We also validate the method when the signal from the mimicked neuropil is dominant compared to the signal corresponding to the mimicked cell bodies (somata). Finally, we propose a novel method for probing neuronal microendoscopic signals by simply combining a miniscope and an implantable short multimode fiber, which we call MiniDART (for Miniaturized Deep Activity Recording with high Throughput). For that, we demonstrate that the inexpensive and commercially available open-source miniscope (Open Ephys Miniscope-v4.4) has already enough sensitivity and illumination power to detect the typical intricated patterns of short MMFs.

## Results

### Proof-of-principle experiment using phantom samples made of 10 μm diameter fluorescent beads

To demonstrate the validity of the method, we implemented an optical setup using a digital micromirror device (DMD), which was used to generate different excitation ground truth (GT)[51,52,66,67] activity traces for each fluorescent source (10 μm diameter fluorescent beads ≈ neuron soma size). Each source emits fluorescence that is collected and transmitted by the multimode fiber (see Fig. 1 and methods for details). Upon propagating through the MMF, the fluorescence wavefront undergoes scrambling, resulting in the emergence of fluorescence patterns upon exiting the fiber (scattering fingerprints). The controlled excitation guarantees that each fluorescent source generates a fingerprint pattern whose intensity transiently fluctuates accordingly with the chosen GT time trace profile (see transient patterns in Fig. 1). We designed GT time traces to be equivalent to optical recording experiments of GECI time traces where calcium signals had F0 set to zero. We then recorded a video of the transient patterns that emerge from a short multimode fiber and applied NMF to the recorded raw data without doing any pre-processing step. In other sets of experiments later, we selected one or more sources available in the FoV to mimic neuropil signal, by exciting them using a non-sparse GT signal (see sections below).

In Fig. 2 we show the results retrieved by NMF and compared with the ground truth. The results consist of individual spatial fingerprint patterns (Fig. 2f) and, most importantly, their corresponding single-activity time traces (Fig. 2g-i) that without NMF would be mixed.

As can be seen in Fig. 2b, the scattering patterns transmitted by the short MMF of 6 fluorescent bead sources (Fig. 2d) have a significant overlap in space when all of the sources are simultaneously excited with the DMD (Fig. 2b). On the other hand, whenever a fluorescent bead is excited individually, each detected spatial pattern has a very

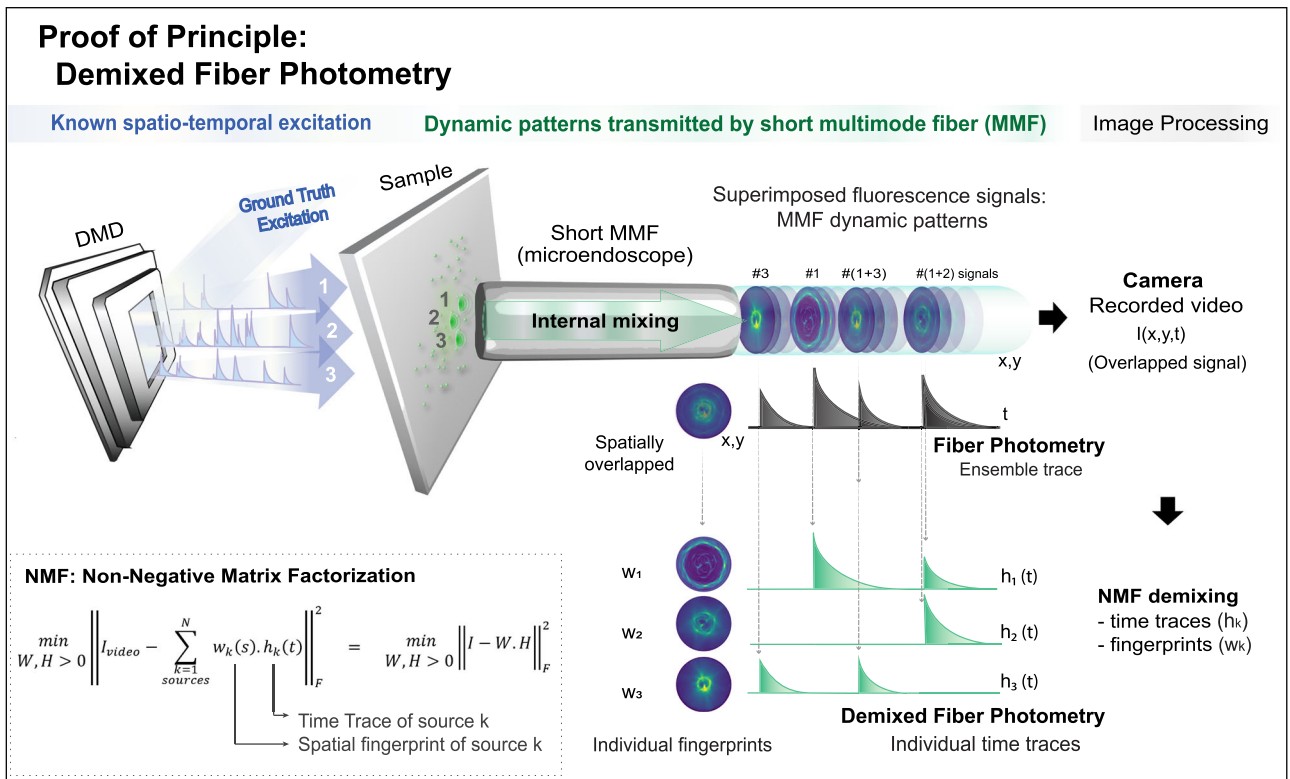

**Fig. 1 | Concept of single-source resolved fiber photometry (demixed fiber photometry).** From left to right: ground-truth excitation mimicking neuronal activity is performed by using a DMD, which can selectively excite a set of fluorescent emitters on the sample with a given time trace, likewise in[51,52]. A short (8 mm long) multimode fiber typically implanted in optogenetics or fiber photometry experiments (NA = 0.39, 200 μm core diameter, step-index fiber) is placed almost touching the sample (distance of ≈ 50 μm) to collect the fluorescence dynamics of each source. Due to its proximity to the sample, the fiber's effective FoV is expected to be slightly larger than the core size. Fluorescence light inside the multimode fiber is subject to multimodal mixing during propagation, which scrambles/mixes

the emitters' wavefront similarly to any scattering media. The transmitted superimposed signal (#1, #2, and #3) consists of fluorescence transient patterns, i.e., 2D patterns (w1, w2, w3) that fluctuate in intensity over time with typical calcium transient profiles (h1, h2, h3)[66,67]. A video is recorded with a camera and a postprocessing step using a spatio-temporal demixing algorithm (unconstrained NMF) is applied to disentangle the overlapped transient patterns into individual 2D spatial fingerprints and their corresponding singular time trace profiles that should match the GT excitations. The optical setup and raw data videos details are fully described in Fig. S1.

different spatial structure/morphology (see the GT scattering fingerprints in Fig. 2e). After applying a simple unconstrained NMF on the recorded video data, the demixed spatio-temporal result by NMF had an overall good agreement with the GT. The ensemble superimposed signal (photometry) was decomposed on its individual fingerprint-trace components (Fig. 2g). Not only the NMF retrieved well each singular temporal activity trace (Fig. 2g-i), but also the individual spatial fingerprint patterns (Fig. 2e-f). Since we know the GT activity, we sorted and assigned the source indexes in descending order of the correlation between their time trace obtained by the NMF and the GT. As we can see in Fig. 2h, the GT-NMF temporal correlation coefficient values were high for all the first 5 beads, which were localized closer to the central region of the fiber (Fig. 2d). More specifically, we obtained an average value of $<\delta_{g,n}> = \delta_{avg} = 85.4\%$ with a standard deviation of std = 3.6% (Fig. 2h). However, the NMF algorithm could not reliably recover the time trace and fluorescence patterns corresponding to bead #6 due to low SNR, probably because it was localized too far from the center, i.e. at the edge of the field of view (fiber core border, see Fig. 2d), as suggested by the low intensity of GT scattering pattern (Fig. 2e, index 6). In the analysis process, the input rank of NMF determines the number of components the algorithm demixes the spatio-temporal signal. When we choose an NMF's rank value higher than the real number of fluorescence sources, we obtain replicas of the scattering fingerprints and the background (see Supplementary Note 1), similar to what happened in a previous work[56]. Thus, counting

the maximum number of unique patterns demixed by NMF could be a way to count the real number of sources probed by the fiber. A more technical analysis of it can be found in the Supplementary Note 1 and Figs. S2, S3, S4, S5.

## NMF demixing of densely superimposed spatiotemporal signals including neuropil dynamic background

Due to the fiber geometry, it is expected that symmetrically probed sources should generate similar spatial fingerprint signals that could in principle limit the capacity of NMF to disentangle fluorescence time traces. Hence, we designed and performed an experiment to address a more realistic video recording than before, such as on a new bead sample with a higher spatial density of beads (see "Methods"); with a fiber field of view where multiple fluorescence sources would have similar radial distance (equidistant) from the center of fiber; with more complex time traces with several overlapping peaks; and by setting one of the sources as a "noisy and dynamic background source" that would yield a rapidly fluctuating non-sparse signal throughout the experiment (e.g., mimicking a neuropil fluctuations). Interestingly, as we can see on Fig. 3, the NMF algorithm was able to successfully demix around 22 fluorescence time traces out of 26 probed sources (see Fig. 3g–i). Note that, all of the 4 poorly retrieved time traces were from sources that were actually close to the fiber edge (c.f. beads 23, 24, and 25 in Fig. 3d) or even outside of the fiber FoV (c.f. bead 26 in Fig. 3d) as expected.

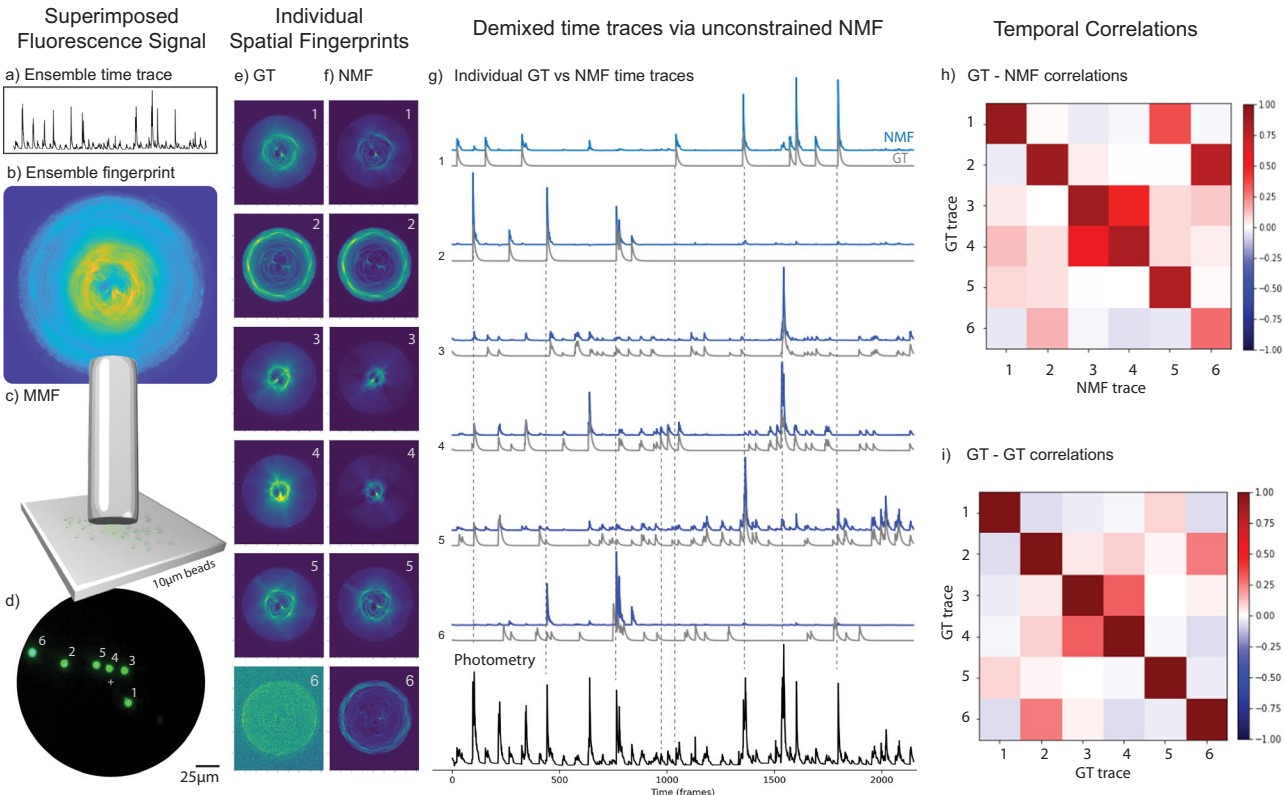

**Fig. 2 | Results of a proof-of-principle experiment performed with 6 fluorescent beads.** From (**a**) to (**d**) we have: (**a**) the ensemble temporal activity (fiber photometry), (**b**) the superimposed pattern image of 6 fluorescent bead fingerprints when simultaneously excited (imaged detected via sCMOS camera) (see "Methods" and Fig. S1); (**c**) the short MMF located at a distance of $60 \pm 10 \mu m$ from the fluorescent beads; (**d**) a CMOS Basler camera with the ground truth image of the sample (see Fig. S1 for setup scheme details). **e** The ground truth (GT) fingerprint patterns obtained from each bead when they were individually excited. **f** The fingerprint patterns obtained via NMF demixing are to be compared with the GT patterns in (**e**). **g** The individual temporal activity traces of the sources obtained with NMF (blue) and their corresponding GT traces (gray). The NMF trace (#6) was not recovered well by NMF since bead #6 was localized very close to the fiber core edge, therefore yielding low signal/contrast of its pattern (see GT scattering

fingerprint of bead #6 in (**e**)). **h** The GT-NMF time trace correlations. The average diagonal value of the first 5 beads was $<\delta_{g,n}> = \delta_{avg} = 85.4\%$ with $\sigma_\delta = 3.6\%$. To better evaluate the off-diagonal elements (time trace cross-talk), we subtract them from their corresponding GT-GT coefficients. Then, we averaged the absolute values of these differences and we obtained the mean absolute error of $\zeta_{avg} = 7.06\%$ with a standard deviation of $\sigma_\zeta = 7.29\%$ for the first 5 beads (see Supplementary Note 2). **i** The GT-GT temporal trace correlation table. Importantly, the GT-GT correlation coefficients show that although each GT trace was unique over time (singular profile), GTs from different sources were not fully uncorrelated. For example, GT traces of beads #3 and #4 were fairly correlated ($\gamma_{3,4} = \gamma_{4,3} = 31.2\%$, in (**i**)) and had a very clear spatial overlap (see GT and NMF scattering fingerprints #3 and #4 in (**e**) and (**f**)).

## NMF demixing of photometric signals emitted from multiple sources buried below a scattering layer

The previous experiments mimicked a condition where there is no scattering media (e.g., brain tissue layer) in between the fluorescence sources and the fiber tip. In this section, we designed a similar experiment as before, but including a layer of plastic paraffin (Parafilm M®) in between the fluorescence beads and the fiber. Parafilm M® is a well-known scattering media and it has similar scattering properties to biological tissue, as the brain[56,68]. We assume that one layer (~120 μm) of Parafilm M® mimics well ~120 μm of brain tissue slice (a brief discussion about why Parafilm M® is a good material to model for biological tissue scattering can be found in the Supplementary Note 3). With this experiment, we expect to show (1) that our method could demix fluorescence spatiotemporal signals transmitted by short MMF coming from sources concealed beneath a scattering layer and (2) that the scattering layer scrambles more the fluorescence wavefront and, consequently, breaks the residual symmetry of the fingerprint patterns we currently obtained, thus affording depth sensitivity to the method. In an extreme case, where a strong scattering medium is present in between the sources and the fiber, multimode fibers are expected to transmit a fully developed speckle as fluorescence

patterns, which our team has already demonstrated that NMF can successfully be employed[51].

To challenge NMF in this experiment with Parafilm M®, we chose a distal FoV where tens of fluorescent beads were actually touching each other and had similar localization around the fiber center (equidistant) (Fig. 4). As shown in Fig. 4g, we chose one of the sources (bead #13) to mimic the dynamic neuropil background during the video acquisition. Despite these multiple challenges, NMF was capable of demixing most of those signals (20 time traces out of 26 sources) as we can see in Fig. 4. As expected, the scattering patterns from this experiment (Fig. 4) are different from the previous cases (Figs. 2, 3). They became less symmetric (which facilitates NMF demixing) and spatially noisier (i.e., with decreased intensity contrast), reflecting the addition of an element in the optical pathway (the parafilm layer) that scatters light and does not exhibit cylindrical symmetry. The latter might explain why some sources could not be demixed well even though they were close to the fiber center (see beads #23 and #24). Indeed, those beads they had a strong spatial overlap with the neuropil (bead #13), which made the signals more difficult to demix. Nevertheless, based on previous works, it is reasonable to expect that a longer acquisition would help on this issue because there would be more frames to be used in the matrix decomposition[51].

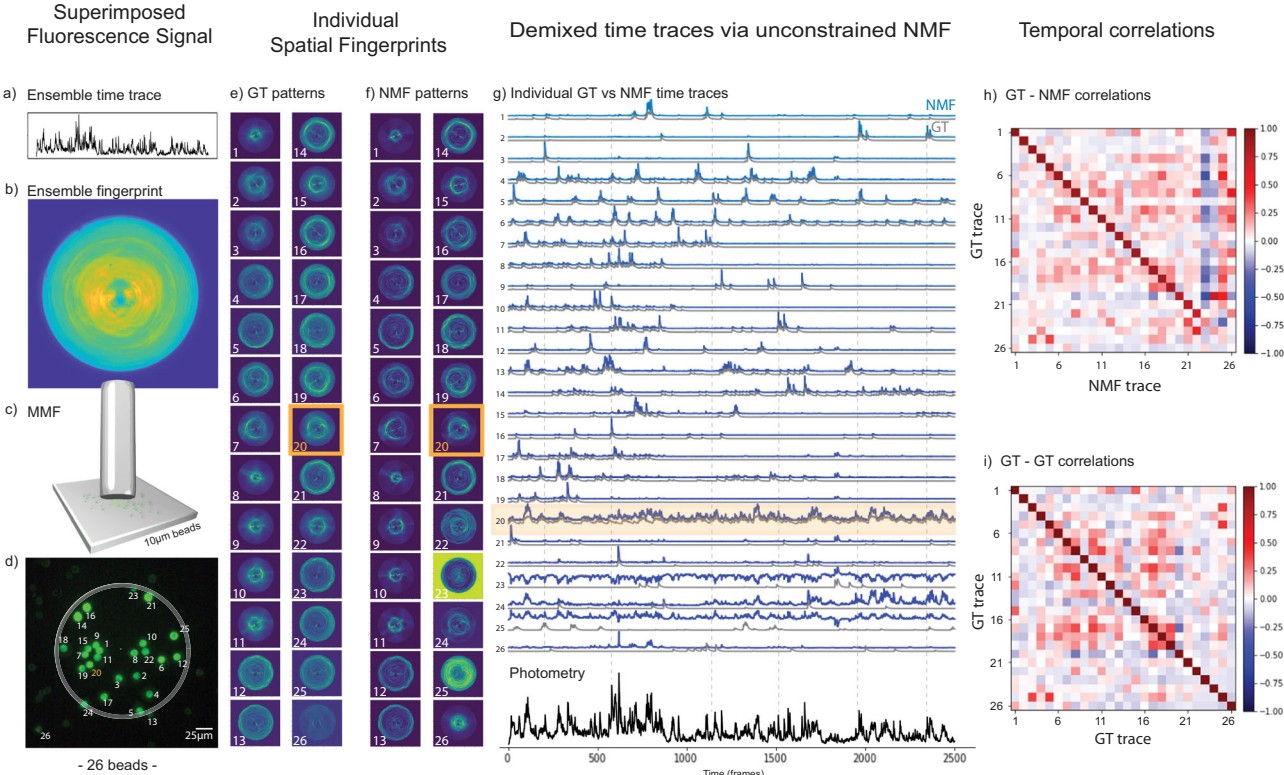

**Fig. 3 | Results of a proof-of-principle experiment performed with 26 fluorescent beads including a neuropil background source.** The bead #20 is modeling the neuropil (highlighted in orange). From (**a**) to (**d**) we have: (**a**) the photometric (ensemble) time trace, which is the sum of 26 time traces; (**b**) the sCMOS detected image of the spatially overlapped fingerprint patterns from 26 fluorescent beads probed by the short MMF (see "Methods"); (**c**) the short MMF located at a distance of $60 \pm 10\,\mu m$ from the sample; (**d**) the ground truth image of the sample (backpropagated fluorescence image detected from a CMOS Basler camera, see details of the setup in Fig. S1). **e** The ground truth (GT) fingerprint patterns obtained from each bead when they were individually excited. **f** The fingerprint patterns obtained via NMF are to be compared with the GT patterns in (**e**).

**g** Top: the individual temporal activity traces obtained with NMF (blue) and their corresponding GT traces (gray). Bottom: the photometric ensemble signal from the recorded video (black line), which is the sum of all individual traces. The fluorescence intensity in all traces in the figure are normalized to 1. **h** The GT-NMF time trace correlations. The average diagonal value of the first 22 beads was $<\delta_{g,n}> = \delta_{avg} = 86.0\%$ with $\sigma_{\delta} = 5.4\%$. To better evaluate the off-diagonal elements (time trace cross-talk), we subtract them from their corresponding GT-GT coefficients. Then, we averaged the absolute values of these differences and we obtained the mean cross-talk of $\zeta_{avg} = 4.4\%$ with a standard deviation of $\sigma_{\zeta} = 3.7\%$ for the first 22 beads (see Supplementary Note 2). **i** The GT-GT temporal trace correlation table showing that the ground truth traces were not orthogonal.

Interestingly, however, NMF is widely employed to denoise image datasets[57,60,61,69,70], and the results from the Fig. 4 strongly suggests that NMF is inherently denoising our data. Such denoising effects were already observed in the previous figures (see the comparison of GT and NMF patterns in Figs. 2, 3). A few examples have been further highlighted in Fig. S6. Consequently, our findings suggest a multifaceted role for NMF, wherein it not only demixes signals and performs image segmentation (fingerprints), but also naturally denoises the data, thereby exemplifying its potential in simultaneously addressing multiple aspects of fluorescence imaging data extraction[60,61].

### NMF demixing of signal from multiple sources hidden by dominant neuropil activity

Previous work by some of the authors has already shown that unconstrained NMF can successfully demix fluorescence time traces from overlapping speckle patterns from a high level of fluorescence background[51,52]. In the last experiments we showed here, we chose to have only one source mimicking neuropil signal, with an amplitude comparable to the amplitude of the bright peaks of each of the other individual time traces. Thus, it remained an open question whether unconstrained NMF could demix the time traces from overlapping scattering fingerprints transmitted by short multimode fibers in conditions where the fluorescence background is dominant, as demonstrated in the previous works.

To address this question, we performed a new set of experiments in which we progressively increased the number of sources mimicking neuropil, and thus progressively increased the strength of the neuropil signal compared to the target sources. More specifically, we chose a new FoV containing 21 beads in total, and we excited 1, 5, or 11 sources with the same neuropil-like non-sparse temporal signal (see Fig. 5, S7, S8). As expected, unconstrained NMF was able to successfully demix most of the individual time traces of the target beads mimicking cell bodies ("target sources") with high temporal correlation accuracy (>80%), even when 11 of 21 sources were mimicking neuropil (see Fig. 5). In such an extreme case, the signal from the total neuropil-like background was approximately 6x stronger on average than the whole ensemble target signal (see Fig. S7m), and approximately 10x larger than the maximum peak of each target source signal. Yet, NMF was able to retrieve 9 of the 10 remaining target sources with very high accuracy (temporal correlation with GT time traces larger than 80%, with an average value of 86% and standard deviation of 5.5%, see Fig. 5). Importantly, as can be seen in Fig. 5d, the target sources were spatially aggregated with many neuropil sources, which is often the case in GECI imaging experiments.

### Validation of the method while probing structurally GFP-labeled neurons in a 50 μm thick fixed brain slice

Next, we investigated if NMF could demix the fluorescence activity of structurally labeled GFP neurons in a fixed brain slice, whose signal has

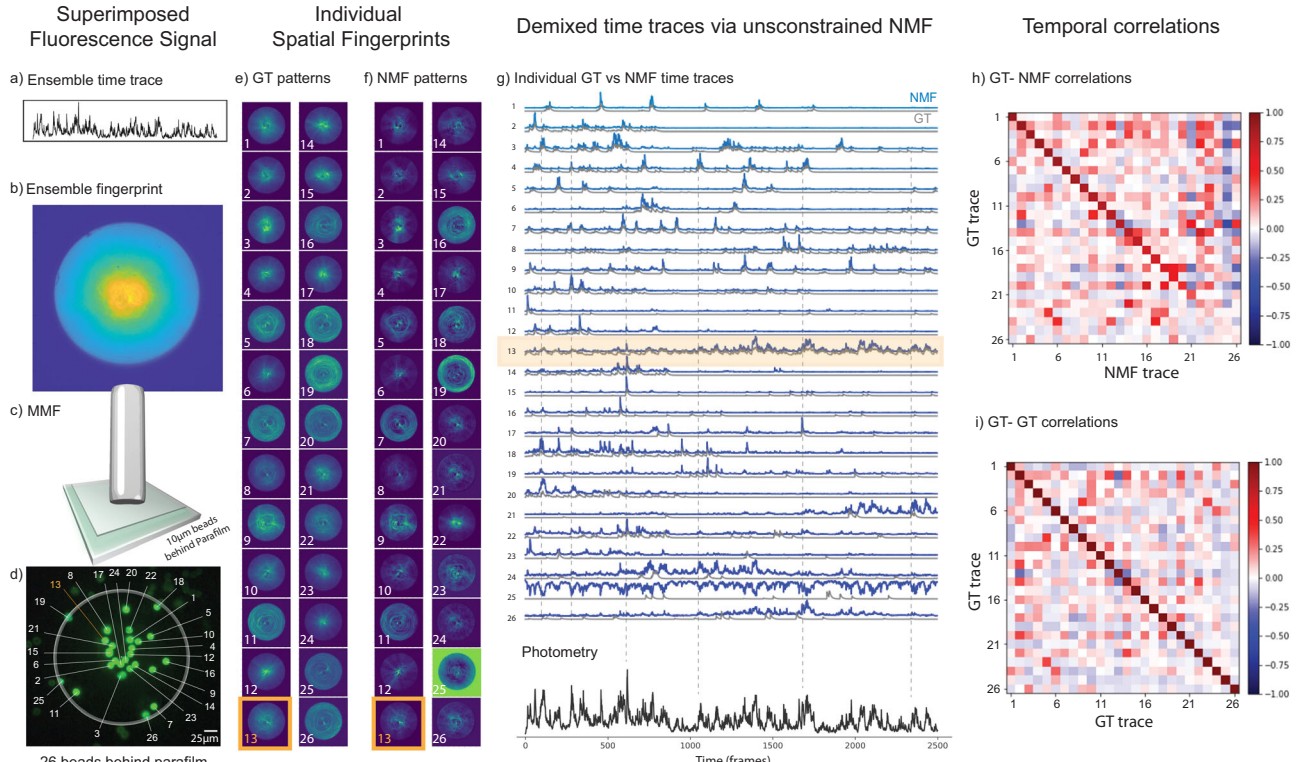

**Fig. 4 | Results of a proof-of-principle experiment performed with 26 fluorescent beads behind a Parafilm M® layer.** The bead #13 is the source mimicking neuropil background signal (highlighted in orange). From (**a**) to (**d**) we have: (**a**) the photometric (ensemble) time trace, which is the sum of 26 time traces; (**b**) the sCMOS detected image of the spatially overlapped fingerprint patterns from 26 fluorescent beads simultaneously probed by the short MMF (see "Methods"); (**c**) the short MMF located at a distance of $60 \pm 10\,\mu$m from the sample; (**d**) the ground truth image of the sample (backpropagated fluorescence image detected from a CMOS Basler camera, see setup in Fig. S1). **e** The ground truth (GT) fingerprint patterns obtained from each bead when they were individually excited. **f** The fingerprint patterns obtained via NMF are to be compared with the GT patterns in (**e**). **g** Top: the individual temporal activity traces obtained with NMF (blue) and their corresponding GT traces (gray). Bottom: the photometric signal from the recorded video (black line), which is the sum of all individual traces. The fluorescence intensity in all traces in the figure are normalized to 1. **h** The GT-NMF time trace correlations. The average diagonal value of the first 20 beads was $\langle\delta_{g,n}\rangle = \delta_{avg} = 77.0\%$ with $\sigma_\delta = 11.9\%$. To better evaluate the off-diagonal elements (time trace cross-talk), we subtract them from their corresponding GT-GT coefficients. Then, we averaged the absolute values of these differences and we obtained the mean cross-talk of $\zeta_{avg} = 5.9\%$ with a standard deviation of $\sigma_\zeta = 5.6\%$ for the first 20 beads (see Supplementary Note 2). **i** The GT-GT temporal trace correlation table showing that the ground truth traces were not orthogonal - some of them were correlated.

a higher background and lower fluorescence brightness compared to fluorescent bead samples[44]. Therefore, in another set of experiments, we changed our fluorescent sample to a 50 μm fixed brain slice (Gad-eGFP labeled neurons, see methods for more details), and we carried out the same type of GT excitation on a few selected neurons in the fiber FoV (mimicking the neuronal activity in a real brain environment). After recording a video of the scattering transient patterns and applying the unconstrained NMF, we confirmed again (see Fig. 6) a good retrieval of the number of neurons, their scattering fingerprints (Fig. 6e, f), and their individual activity traces (see Fig. 6g–i). As in the previous experiment, the GT-NMF temporal trace correlation values obtained were high, with an average value of $\langle\delta_{g,n}\rangle = \delta_{avg} = 86.7\%$, and a standard deviation of std = 2.8%.

### Pattern sensitivity evaluation of low-cost miniscopes while probing a single fluorescence source

Finally, we investigated if an inexpensive miniscope (Open Ephys Miniscope-v.4.4) coupled with a MMF would have enough sensitivity to excite and image an individual fluorescence fingerprint pattern emitted from a single fluorescence source (Fig. 7). In such conditions, both the LED excitation from the miniscope and the fluorescence signal from the sample would be transmitted within the short MMF before imaging. This is an important question because the miniscope has simple, inexpensive, and compact components, such as the commercial-grade CMOS detector, rather than a high-end sCMOS camera as in our proof-of-principle table-top experiment (see "Methods" and Fig. S1). To tackle this question, we combined a miniscope with a short MMF (the MiniDART) and tested the miniscope sensitivity with a very sparse bead sample: a single fluorescent bead that we can displace laterally in the fiber FoV (see methods and Fig. 7b, c). When we set the miniscope for low LED power and with no camera gain (LED = 20% corresponds to a transmitted power through the short MMF of $P_{MMF} < 10\,\mu$W), it was already possible to detect short MMF scattering fingerprint patterns with very good contrast at the fastest frame rate available in the miniscope control software (FPS = 30 Hz, corresponding to 33 ms exposure time, see Fig. S9). Interestingly, whenever raising the LED power, decreasing the framerate speed, or raising the camera gain value, the detected patterns by the miniscope got saturated, suggesting that the miniscope CMOS has already enough sensitivity to probe scattering fingerprints through short MMFs even from less bright fluorescent sources than the ones used here, especially when the fluorescent source is close to the center of the fiber core (see Fig. 7c). At distances d < 20 μm from the core center, the CMOS got saturated while the miniscope GUI settings were: LED power = 20% (where $\text{LED}_{max} = 100\%$, corresponding to a transmitted power through the short MMF of $P_{MMF}^{max} = 125\,\mu$W), FPS = 10 Hz (where $\text{FPS}_{max} = 30$ Hz), and Gain = 1 (where $\text{Gain}_{max} = 3.5$).

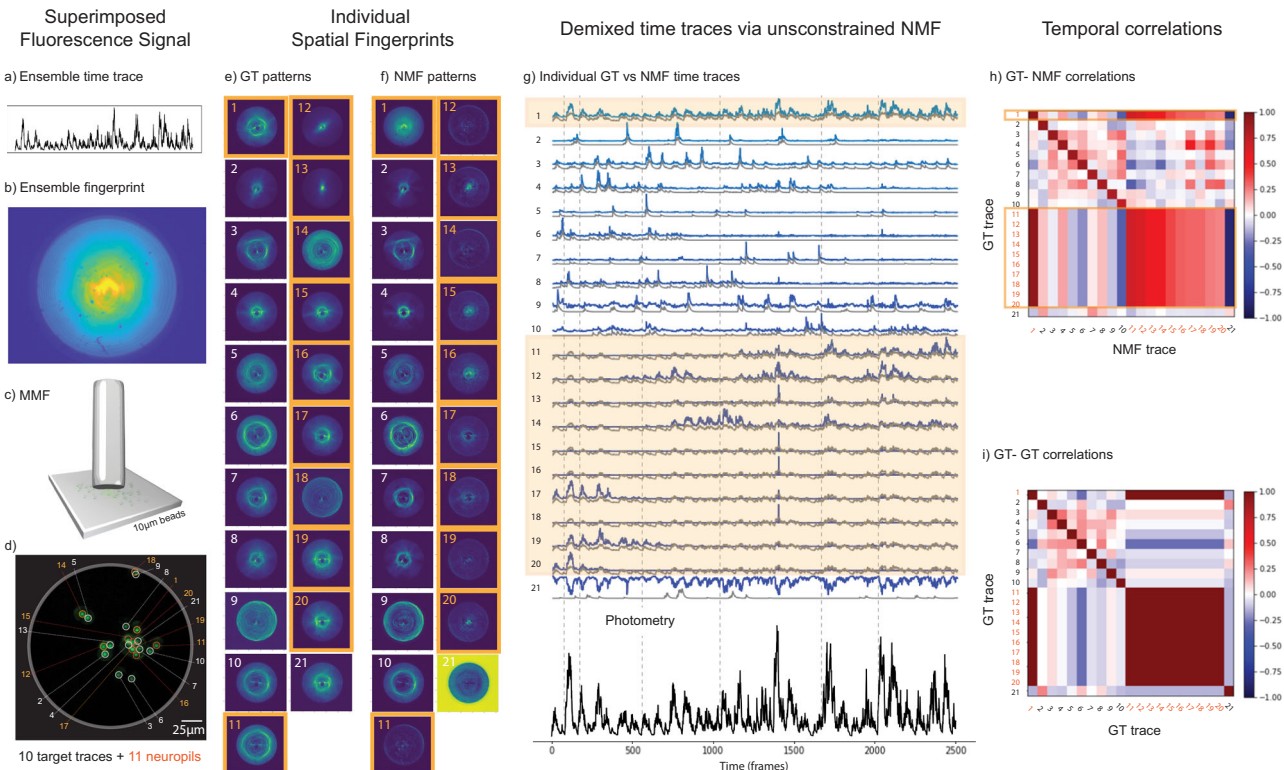

**Fig. 5 | Results of a proof-of-principle experiment performed in conditions simulating dominant neuropil activity.** The sample consists of 21 fluorescent beads, where 10 beads had sparse and unique time traces and the other 11 had the same non-sparse neuropil-like time trace (dynamic background). The beads with "neuropil-like" background activity are highlighted in orange (#1, #11, #12, #13, #14, #15, #16, #17, #18, #19, and #20). The remaining beads (#2, #3, #4, #5, #6, #7, #8, #9, #10, and #21) had unique sparse neuronal activity time traces mimicking signal from neuronal cell bodies (target sources). **a** The photometric (ensemble) time trace, which is the sum of all the 21 time traces; (**b**) the sCMOS detected image of the spatially overlapped fingerprint patterns from 21 fluorescent beads simultaneously probed by the short MMF (see "Methods"); (**c**) the short MMF located at a distance of $60 \pm 10\,\mu m$ from the sample; (**d**) the ground truth image of the sample (backpropagated fluorescence image detected from a CMOS Basler camera, see setup in Fig. S1). **e** The ground truth (GT) fingerprint patterns obtained from each

bead when they were individually excited. **f** The fingerprint patterns obtained via NMF. Note that the NMF pattern #1 is the neuropil pattern due to the spatial overlap of 11 sources (highlighted with orange squared boxes). **g** Top: the individual time traces obtained with NMF (blue) and their corresponding GT traces (gray). Bottom: the photometric signal from the recorded video (black line), which is the sum of all individual traces. The fluorescence intensity in all traces in the figure are normalized to 1. **h** The GT-NMF time trace correlations. The average diagonal value of the first 10 beads was $\langle \delta_{g,n} \rangle = \delta_{avg} = 86.0\%$ with $\sigma_\delta = 5.5\%$. To better evaluate the off-diagonal elements (time trace cross-talk), we subtract them from their corresponding GT-GT coefficients. Then, we averaged the absolute values of these differences and we obtained the mean cross-talk of $\zeta_{avg} = 6.1\%$ with a standard deviation of $\sigma_\zeta = 5.7\%$ for the first 10 beads (see Supplementary Note 2). **i** The GT-GT temporal trace correlation table showing that the ground truth traces were not orthogonal.

Fundamentally, the MiniDART is not designed for imaging neurons or localization (like in other works that used learning algorithm methods[62,63]). In many experiments, the relevant biological information does not depend on the shape or local position of the neurons, but mostly on the activity of each of them and their number. To extract this information, just demixing fluorescence time trace signals with NMF would be enough for relevant applications, since the number of neurons (NMF rank) could be obtained by counting the number of unique MMF fingerprints, and possibly confirmed by a *post-mortem* evaluation of the brain region right below the thin hole made by the MMF. In addition, recording the spatial fingerprints of each source would be useful for chronic experiments (to match the sources from one session to the next). Nevertheless, as we can also see in Fig. 7, the morphology of short MMF fingerprints provides some interesting insights into the point source position at the fiber distal end - without the need for any computational learning method. As we move a single bead laterally in a radial manner (in the x,y-plane at the fiber's distal end), a few easily interpretable geometrical properties of the fluorescence patterns systematically change in the fiber's proximal end (Fig. 7c). More specifically, when the bead is displaced from the center towards the fiber border (of a distance d) a bright ring (of radius ρ) with some spiral ramifications is deterministically formed (see Supplementary Note 4).

Larger d distances yield wider rings (i.e., larger ρ, see Fig. 7c). Thus, each fingerprint's bright ring diameter is encoding radial information about the bead lateral localization at the distal end. The dependence of fingerprint shape features on the individual point source position is further discussed in the Supplementary Note 4.

## Discussion

Multimode fibers are well known to be minimally invasive micro-endoscopic probes that could be promptly combined with optogenetics manipulation of neuronal activity for behavioral neuroscience studies in living mice[16,28,29,47,64]. It is well known from the literature that long multimode fibers (MMF length > 100 mm) could be used to transmit fluorescence activity signals and that would naturally generate a speckle wavefront due to internal multimodal mixing[31,38,71,72]. However, as well described in wavefront shaping experiments, speckle wavefronts are extremely sensitive to fiber bending, torsion movements, and temperature changes along the fiber, which demands a long and meticulous wavefront propagation characterization (e.g., calibration using spatial light modulators, SLM) to compensate for all the changes in spatial properties of the speckles[37,48,49]. In other words, there would be many experimental conditions to take into account to guarantee that the speckle patterns from each source would remain

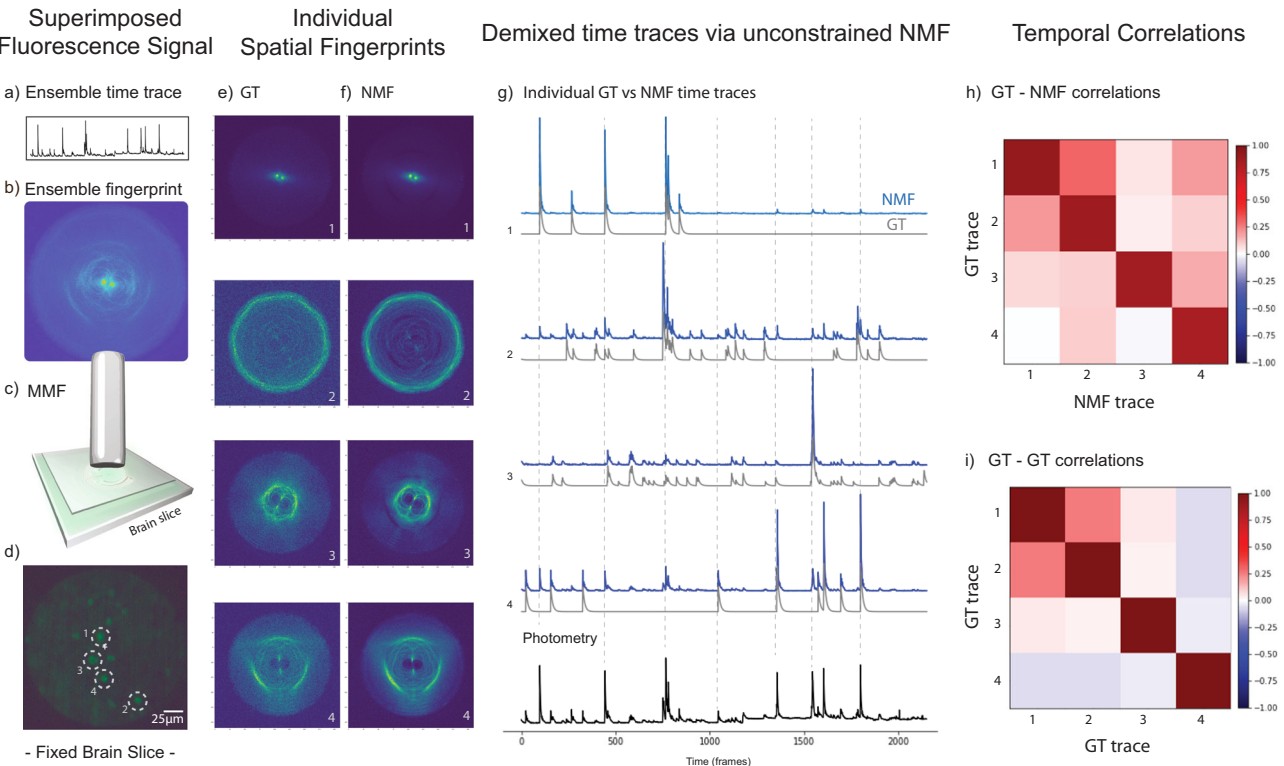

**Fig. 6 | Validation of the concept of single-activity resolved fiber photometry with short multimode fibers in a brain tissue environment (in vitro).** Sample: Gad-EGFP neurons fixed in a 50 µm brain slice, sealed in between 2 coverslips to keep the humidity of the tissue (see "Methods"). **a** The ensemble photometric time trace of this experiment. **b** The fiber proximal end image of 4 neurons' fingerprint patterns spatially overlapped on the sCMOS camera chip. **c** An illustration of the short MMF placed above the top coverslip of the sample, at a distance of ≈ 60 ± 10 µm from it; and (**d**) the GT image of the sample highlighting the 4 selected neurons to be excited (structurally labeled). **e** The GT fingerprint patterns are obtained from each neuron when individually excited. **f** The fingerprint patterns retrieved via NMF are in good agreement with the GT patterns in (**e**). **g** The demixed temporal activity traces are sorted in descending GT-NMF correlation order

(from the most correlated time traces on top to the least correlated time traces on the bottom). Traces in blue are retrieved by NMF and temporal traces in gray are their GT. **h** The GT-NMF temporal trace correlation coefficients. **i** The GT-GT temporal correlations. The average diagonal value in (**h**) of the 4 neurons was $<\delta_{g,n}> = \delta_{avg} = 86.7\%$, with standard deviation of = 2.8%. Regarding the non-diagonal elements (cross-talk), the mean absolute error taking into account the GT-GT coefficients was $\zeta_{avg} = 8,95\%$ with a standard deviation of $\sigma_\zeta = 8.02\%$ (see Supplementary Note 2). Again, although each GT trace was unique in time (singular), they were not fully uncorrelated as we can see in the GT-GT correlation traces (**i**). Interestingly, neurons #1 and #2 (i.e., the two best NMF retrieved results) were also the most temporally correlated ones in the GT excitation ($\gamma_{1,2} = \gamma_{2,1} = 25.0\%$).

the same over time when using a long MMF as a photometric probe in freely-behaving mice experiments. That is why using only a short and stable multimode fiber for this type of application could be a very pragmatic solution.

To this end, we designed an experiment with a known ground truth excitation to be able to evaluate if NMF could demix the individual spatio-temporal readouts characteristic of short MMFs and GECI recordings. In this paper, we demonstrate that it is possible to demix such spatio-temporal signals in vitro, using a simple and general unconstrained NMF algorithm on the video data recorded in our proof-of-principle experiment. We designed in vitro samples to mimic as much as possible the real brain: we used tens of sources embedded in agarose (where a few of them are touching each other), with multiple overlapping temporal transient peaks signals, and we chose one or several sources to mimic fluorescent neuropil. In one experiment, we added a scattering layer between the sources and the fiber, with scattering properties equivalent to that of a 120 µm brain layer. In all of these experiments, we demonstrate that we are able to successfully demix most of the sources with NMF. In one challenging case, where the sources were touching one another below a scattering layer, we show that NMF retrieved 20 of the 26 sources with high fidelity. In another extreme case, we included a dominant neuropil-like background signal to compete against the signal from the target sources (mimicking activity from neuronal cell bodies). We designed this

"neuropil-like" signal to be a surrounding, non-sparse, and dynamic fluorescent background emitted from sources that were in the vicinity or even aggregating with the target sources (Fig. 5), with a total amplitude that was 6 times larger than the ensemble signal from the target sources. Despite these stringent conditions, we demonstrated that a general unconstrained NMF algorithm could successfully demix the signals from most of the target sources (9 out of 10). Therefore, this work opens a promising direction to improve fiber photometry fluorescence experiments by reaching single-source temporal activity resolution.

We finally suggest that this method could be applied in vivo in freely behaving animals by coupling the short implanted fiber with a miniscope (MiniDART concept). Indeed, we showed that patterns measured with a miniscope for single sources are very similar to patterns measured with the benchtop microscope used in our in vitro experiments. Therefore, one could benefit from the short MMF's scattering fingerprint patterns and rigidity by directly imaging the proximal end of the short (implantable) MMF with a miniscope. That would be different from the typical scheme in fiber photometry, where there is an optical coupling between the transmitted light from the implantable MMF to another long MMF (relay). In this sense, we tested and confirmed that even a low-cost miniscope would have enough LED power and CMOS speed/sensitivity to image the intricated scattering patterns transmitted by short MMFs emitted from just a single source.

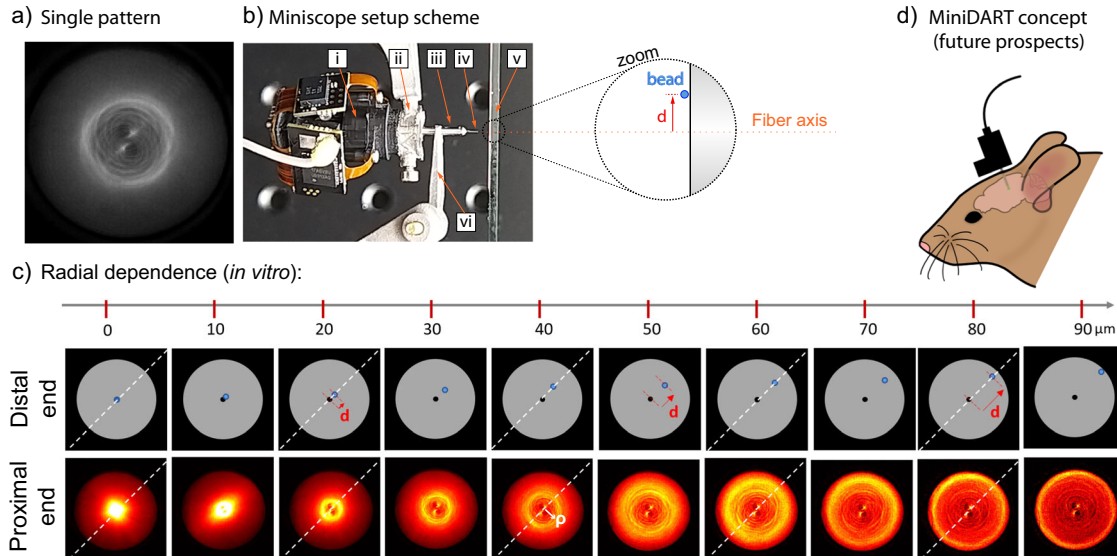

**Fig. 7 | Novel microendoscopy concept using a short MMF and a miniscope: the MiniDART. a** A typical fingerprint pattern from a single-fluorescent bead (10 μm diameter) probed by using only the miniscope excitation and miniscope detection through the multimode fiber. **b** The experimental setup to probe scattering patterns from the short MMF includes (i) the miniscope, (ii) a customized titanium base plate (YMETRY®) to hold the miniscope, (iii) a ferrule (Thorlabs SFLC230-10) that rigidly holds the multimode fiber (iv) within it, (v) a sample consisting of a single fluorescent 10 μm bead (spatial density <1 bead/cm²), and (vi) a customized titanium tweezer (YMETRY®) to hold the ferrule. **c** Scattering fingerprint patterns at the proximal end (bottom row) depending on the radial position (**d**) of the single-bead at the distal end (bottom row). Position (**d**) (red arrow) is indicated in relation to the fiber axis (the bead is represented as a blue spot in the zoom of (**b**) and the top row images of (**c**), while the axial center of the fiber is represented as a fixed black dot in the top row images of (**c**)). Each pattern acquisition in the proximal end (bottom row of (**c**)) corresponds to 10 μm steps of the bead from the fiber central

axis in the distal end (top row of (**c**)). The bigger the distance d (red arrow; top) of the bead from the center of the fiber, the larger the radius ρ (white arrow; bottom) of the bright spiral-ring pattern in the proximal end. The diagonal white dashed line is the azimuthal orientation of the red vector d, which always coincides with the alignment angle of the 2 central bright points of the fingerprint patterns in the proximal end (see Fig. S10 for details). The highest LED power values measured at the distal end of the fiber (whose core is 200 μm in diameter) were around 9.5 μW, which yields an excitation intensity of 0.3 mW/mm² at the output of the fiber core, and $2.4 \times 10^{-5}$ mW excitation power per bead area. Exposure time: 100 ms (Miniscope FPS = 10 Hz). For more details, see Figs. S9, S10. **d** The concept of doing experiments with a MiniDART device, which combines a miniscope and a short implantable multimode. For future in vivo experiments, the MMF and miniscope baseplate should be glued on the mouse skull with dental cement in the same way typical miniscope experiments are performed with GRIN lenses.

Both the multimode fiber and the miniscope (Open Ephys Miniscope-v4.4) are affordable and available in the market to carry out experiments in living mice. Therefore, in this paper, we show a low-cost and simple idea to expand the capacity of fiber photometry methods in resolving neuronal activity circuitry (that have already been used to investigate deep brain regions in chronic behavioral experiments in freely-behaving mice). In addition, as a minor result, we point out in Fig. 7 that the short MMF scattering fingerprint morphology can have some interpretable geometry that has a relation with the source (≈10 μm size) localization at the distal end of the fiber. The relationship between the source localization and its fluorescence pattern shape is discussed in the Fig. 7 and S10, but the key message for demixing is that the scattering fingerprint patterns do not seem to be ambiguous depending on the source lateral position from the fiber axis.

To apply this method in vivo, a few difficulties could arise, but we think they can be circumvented:

Fluorescence background: In calcium imaging experiments, there are two types of background: a static background (corresponding to the neurons resting fluorescence F0) and a dynamic background (typically the activity of the neuropil). In our experiments, we didn't model the resting fluorescence of the neurons (F0 = 0). However, recent GECI indicators such as GCaMP8 have low resting fluorescence and show large transient signals, with a ΔF/F0 corresponding to one action potential ranging between 40% and 100% depending on the variant used[73]. Therefore, this static background should remain moderate, and we expect NMF to be able to extract it (for example using rank-1 matrix factorization, see supp info of[74]). On the other hand, dynamic background should be less straightforward to subtract. In this work, to evaluate if NMF could find and remove a non-sparse dynamic

background component from our current signal, we included one (Figs. 3, 4) or several (Fig. 5, S7, S8) additional sources to model a neuropil signal. These sources were chosen to have a spatial fingerprint overlapped with many other sources in the FoV, and to exhibit a non-sparse, fast oscillating signal, with amplitudes on the same order of magnitude as the transient peaks of the other individual sources. In the two experiments with one neuropil source, we first showed that a simple NMF algorithm could successfully retrieve more than 20 out of 26 spatiotemporal sources from the sample, including the neuropil-like source. Then, in a more challenging experiment (Fig. 5, S7, S8) where 11 out 21 sources were mimicking synchronous neuropil, NMF successfully demixed 9 out 10 remaining target sources with 86% average temporal correlation accuracy. These results suggest that our method could be suitable to extract activity from individual neurons in conditions where the average neuropil fluorescence is several times larger in amplitude than the remaining ensemble signal from the neurons in the FoV.

In addition, previous work from the team addressed NMF performance to extract activity from target sources in strong fluorescence background[51,52]. NMF performance was evaluated by quantifying the cross-correlation statistics of the retrieved time traces over different experimental background conditions. Typically, NMF performance starts to *slowly* get impaired when the max value of the background signal is 1.5x bigger than the max value of the sources' activity time trace signal. In this case, the median temporal cross-correlation gets lower than 80%, with the first quartile of ~65%. Nevertheless, even when the max background over the max activity ratio is of 2.0, the median cross-correlation in the time trace is still above 70%, with the first quartile above 60%[51].

Moreover, the performance of NMF could be improved compared to what has been shown here. Indeed, our team has already shown that the cross-correlation between NFM and GT traces increases with the number of frames in the recording[51], thus longer recordings are preferable for noisier data. In addition, it has been shown that for cases in which the background is larger than the signal from the somata, constrained NMF could retrieve well the time traces from neurons recorded with wide-field microscopy[60,61]. In our case, we could apply similar spatiotemporal constraints (adapted to the fingerprint patterns we see here), which should allow significant improvement of demixing performances. Finally, when designing in vivo experiments, we would advise using GCaMP8 indicators targeted to the soma[75] to minimize the neuropil component and therefore facilitate signal analysis. In the first experiments, sparsity of expression could be adjusted so that only 20 to 30 neurons are labeled within the illumination volume of the fiber. In this case, we expect that spatiotemporal unmixing of most of the sources should be possible. Indeed, neurons distributed in the 3D scattering tissue should produce unique fingerprint patterns at the camera and should therefore be properly unmixed, similar to what we demonstrated in in vitro proof-of-principle experiments.

Sample motion: In the case of 2-photon imaging experiments in the cortex with a cranial window, motion artifacts were 2–4 μm at z distances shorter than 150 μm from the optical window[1,76]. In our case, we expect similar motion artifacts when exploring shallow regions of the brain, and smaller artifacts for deep regions. Indeed, this is what has been observed for 2-photon imaging with GRIN lenses[77,78]. In addition, since in our experiments the patterns smoothly change upon source motion in the distal FoV (see Fig. 7 and S9), we expect that motion artifact to remain small, periodic, and restricted in space, and we expect NMF to extract an average pattern for each source.

Fiber bending: Typical studies on multimode fiber bending effects are done with relatively long fibers (>100 mm, typically around 300 mm long), which are quite flexible to be bent (Thorlabs manufacturer recommends bending fibers until a maximum of 21 mm of radius of curvature for 200 μm diameter core fibers). However, light propagation studies dealing with extreme bending cases of fibers with similar core as the one we used (200 μm diameter) display typical smallest bending radius of around 5 mm, which is the typical length of the short MMF we use[48]. Like any other solid material, shorter multimode fibers (~10 mm) are way stiffer to bending than long fibers (>100 mm) (Euler–Bernoulli beam theory of solid materials mechanics)[74,79–81]. Typically, even long fibers have critical ~6 mm bending radius (with typical bending stress of ~700 MPa), where the Young modulus of multimode fibers changes very little (less than 1%) and its value is for practical purposes considered constant[80]. Therefore, short fibers as the ones we used (~8 mm long) are significantly rigid and very difficult to be bent. This is particularly true for in living mouse experiments, where one end of the fiber (proximal) will be glued on the mouse's skull, and only the distal end of the fiber will be "free" to be bent (i.e., this end will be actually be surrounded by the mouse brain, dumping the small internal movements[77,78]). In fact, most of the fiber length is rigidly glued within the ferrule, letting only a small portion of the fiber "free" to be bent (typically, in between ~1 mm and ~4 mm long sticking out of the ferrule, which is below the critical length of the fiber). Therefore, we expect very little bending of the fiber during in vivo recordings.

Bit depth of the miniscope camera: The current 8-bit depth CMOS camera of the miniscope might experimentally limit the potential of the method in probing a large number of sources because the miniscope camera can get more easily saturated due to the current short dynamic range (shallow bit depth, see Fig. 7c and Fig. S9). Saturated camera pixels should be avoided during the video recording because they do not allow one to distinguish GECI dynamics within the saturated frames. Probably, when N number of pixels are saturated over a finite number of F frames, it might create a cross-talk in all retrieved time traces over the F frames whose fingerprint signals depend on those N saturated pixels. In our tabletop experiment, the sCMOS camera we used had a larger bit depth (16-bit dynamic range) and we demonstrated that a simple unconstrained NMF was able to demix more than 20 sources with a significant spatiotemporal overlap with other sources and non-sparse neuropil activity. In our experiments the camera dynamic range (DR) in the ensemble time trace were around: DR = 10.000 for Fig. 2 (Proof-of-principle with 6 beads), DR = 1.500 for Figs. 3, 4 (Proof-of-principle with 26 beads with and without Parafilm M®), DR = 450 for Fig. 6 (Proof-of-principle with 50 μm thick brain slice). Therefore, most of the experiments used a moderate dynamic range compared to the available 16-bit. We provide all the tiff files of the videos in the data repository referenced below.

Regarding the current literature, previous works have demonstrated methods to obtain some degree of readout specificity in photometry experiments. For example, in Bianco M. et al. APL Photonics (2022)[33], the authors have tapered the end of a multimode fiber so that fluorescence light coming from different depths of the taper is spatially separated in the far field of the optical fiber, at a camera. Although this technique is a nice improvement of fiber photometry, it does not give single source specificity, but rather it allows to obtain signals from spatial ensembles located at different depths in the tissue. Our approach is very different: we are not relying on spatial information only to separate the sources, but we take advantage of the temporal fluctuations of each source to extract both the spatial fingerprint and the temporal trace corresponding to each source, using a NMF algorithm. As a consequence, we manage to reach single source specificity. Moreover, in our approach we show that the raw video data doesn't need to be transformed or pre-process to obtain more specificity in the photometric readouts, although we could promptly do it as an extra strategy to improve its performance. It is worth noting that the actual implementation is performed with a cylindrical MMF, which gives ring patterns resembling that of Bianco et al. but our method would also work with other types of fibers (for example square core fibers) as long as patterns corresponding from different sources are different. Lastly, multimode fiber modal dispersion doesn't necessarily limit our method, but it can be engineered to tune our method performance.

Moreover, computational learning tools have already been applied on short MMF scattering fingerprints to retrieve back the image of deep brain neurons from a fixed thick brain slice[54,55]. However, it is not yet proven that this approach could correctly assign the neuronal activity in living mice or any artificial condition. On the temporal resolution side, the authors tracked the movement of a fluorescently labeled worm in 2D with a framerate of 2 Hz, which is relatively low for typical calcium imaging experiments performed in living mice. In the latter case, experiments are typically carried out with a framerate of at least 10 Hz (such as the one available in the miniscope DAQ software). Besides, any learning algorithm depends on the specific conditions of the training set (fiber properties, imaging optical components properties, etc.), which is less general and therefore less applicable to any other new neuroscience experiment.

In our case, we used a simple unconstrained NMF, which is a general mathematical method that simply decomposes a matrix into a product of two matrices where all the entries are non-negative. This has been applied in many different fields, such as astronomy, audio processing or computer vision[53,55,57–59,69,70]. In particular, we transformed our recordings into this matrix form where the columns (rows) represent spatial (temporal) information, but the method is blind to this physical interpretation and just operates numerically. Importantly, we wanted to establish as clearly as possible a general proof-of-principle for the technique, focusing on a solid ground truth, keeping the algorithm in its standard version (as general as possible), so as to underlay the physical concepts. As we previously mentioned, a more tailored NMF implementation, for example using sparsity or other image-based and/or temporal constraints based on the specific nature

of our experiments, would probably yield better results in terms of fidelity or achieving a higher number or retrieved traces[60,61]. On the optical engineering side, one could tailor new miniscopes for this specific method or conveniently explore different multimode fibers types for the optical recordings. Regarding the miniscope potential improvements, when we spatially binned the video recorded by our high-end sCMOS camera (tabletop experiment) until we get a similar number of FoV pixels used in the miniscope, the retrieved results of NMF fingerprints and traces were visually indistinguishable (see Fig. S11) compared to a non-binned analysis. This finding reinforces the idea that the current default design of the miniscope (LED power, camera speed/sensitivity, and FoV magnification) might be already enough to perform a living mouse experiment, although most pixels of the miniscope's camera chip was not used in our case (less than 1/9 of the FoV pixels were used to image a single MMF proximal end tip), and could be explored by simply changing the current magnification lens of the miniscope.

To conclude, we have demonstrated how time traces of individual fluorescent sources can be demixed from spatio-temporal intensity patterns transmitted by short multimode fibers. This is a first step towards measuring the activity of individual sources in fiber photometry experiments that use thin multimode fibers as a microendoscopic probe. Besides, we show that the currently available low-cost miniscope has enough sensitivity to image directly MMF scattering fingerprints from individual sources, suggesting that this experimental configuration could be advantageous for future long-term microendoscopic studies in freely-behaving animals due to the intrinsic rigidity of short fibers ( < 10 mm). Therefore, we believe that this work can open a whole new avenue for novel and affordable minimally invasive deep brain optical microendoscopic studies to probe (potentially several) deep brain regions simultaneously in freely-behaving animals, including experiments that could be conveniently coupled with optogenetics tools to photoactivate or inhibit neurons which are already a routine in many neuroscience labs.

## Methods

### Proof-of-principle setup

The proof-of-principle setup is visually illustrated and fully described in Fig. S1.

### Fiber-ferrule preparation

The multimode fiber was purchased from Thorlabs (FT200UMT, NA = 0.39, 200 ± 5 μm core diameter, 225 ± 5 μm cladding diameter). Around 10 cm of the fiber was initially cleaved with a ruby blade (Thorlabs, S90R), and the quality of the cleaved edge was inspected with a stereomicroscope (LEICA A60F, maximum magnification 30x). The fiber was glued within a 1.25 mm wide and 6 mm long stainless-steel ferrule (Thorlabs SFLC230-10, 230 ± 10 μm bore diameter) using an ultraviolet curing glue (Norland optical adhesive 81) so that the cleaved end was chosen to be the distal end of the fiber (with 2 mm of it sticking out of the ferrule). The excess of the fiber on the other end (proximal end) was cut close to the ferrule edge and then sequentially polished (KRELL-TECH NOVA device) with 3 different silicon carbide polishing disks with gradually descending roughness (30 mm, 3 mm, 0.3 mm, PSA 4" polishing disks from KRELLTECH). The quality of the polished end (proximal end of the fiber) was verified with the LEICA A60F stereomicroscope. We used a customized titanium tweezer and baseplate to hold the ferrule and the miniscope respectively (YMETRY®).

### Preparation of the fluorescent bead sample

In the proof-of-principle experiments, the samples consisted of randomly distributed fluorescent beads on borosilicate glass (22 × 40 mm cover glass, thickness Nb.1.5, purchased from VWR). An aliquot (100x diluted in milli-Q water) of the polystyrene (PS) particles aqueous suspension (PS - FluoGreen - Fi226 – 1 mL, 10.23 ± 0.13 μm size,

abs/em = 502/518 nm, purchase from microParticles GmbH, Germany) was used to randomly distribute the beads on the cover glass. The beads' size was chosen to have similar size to typical neuron soma probed in calcium imaging experiments. Besides, the emission spectrum of the beads is close to the common GFP calcium indicators. For the single-bead sample, a more diluted aliquot (from $10^6$ to $10^7$ times dilution) was used to guarantee a very spatially sparse bead sample with density ≤1 bead/cm², so that there would be only a single bead throughout the Fiber FoV whenever translating the bead laterally in the miniDART experiment (easily inspected by the miniscope itself without the fiber). The sample with high spatial density of beads was done by mixing concentrated bead aliquot with 5% Agarose aqueous solution with proportion 1:1 in volume (Sigma-Aldrich product# A2576, ultra-low gelling temperature, biology grade).

### Preparation of the fixed brain slice sample

The work includes data from one GAD65-EGFP transgenic mouse (heterozygote; male; aged 6 months) expressing EGFP in Gad65-positive interneurons in the brain and spinal cord. All procedures involving this animal complied with French and European legislations relative to the protection of animals used for experimental and other scientific purposes (2010/63/UE) and were approved by the "Charles Darwin" institutional ethics committee under the direction of the French National Committee of Ethical Reflection on Animal Experimentation under authorization number APAFIS 26667. The mouse was euthanized by cervical dislocation performed on the terminally anesthetized animal (5% isoflurane for 5 min in an induction chamber) and the brain was quickly removed from the skull and immediately transferred into a fixation solution (4% paraformaldehyde; pH 6.9 buffered; Sigma-Aldrich #1004965000). After 12 h at 5° the brain was transferred into phosphate-buffered saline (PBS) and cut into tangential slices of 50 to 100 μm thickness using a vibratome (Leica VT1000). Obtained slices were stored in multi-well plates with PBS and assessed with a fluorometric microscope before the experiment with the DMD.

### NMF Analysis in Python

The raw video data [2D image, time] was reshaped to a 2D matrix [1D image, time] to fit the input format of NMF. We used the scikit-learn decomposition NMF package freely available online (Python), which is also explained in the supp info of previous work[52]. In the code pipeline, an optional pre-processing step was added, which included an option for pixel binning (see Fig. S11). The NMF parameters chosen ignored the additional parametric terms that are introduced to account for the sparsity of the data. The parameters chosen were: *init* = 'nndsvd', *random_state* = 0, *max_iter* = 3000, *solver* = 'cd', *l1_ratio* = 1, *beta_loss* = 2, *alpha_W* = 0, *alpha_H* = 0. The parameter *n_components* (which is the chosen NMF rank) changes depending on the experiment and a more detailed discussion about how to choose this value can be found in the Supplementary Note 1. In particular, the rank used in Fig. 2 was $\text{rank}_{Fig2} = 9$ (explained in the Supplementary Note 1), and the rank for Fig. 6 was $\text{rank}_{Fig3} = 5$. The analysis of the data presented in Fig. 2 with different ranks is illustrated in Figs. S2, S3, S4, S5 and Supplementary Note 1.

The non-diagonal coefficients mean and standard deviation of the temporal correlation results:

The matrix of the non-diagonal elements $\zeta = \zeta_{i \neq j}$, where each element is the absolute error value between GT-NMF and GT-GT coefficients is detailed described in the Supplementary Note 2. The absolute error ($AE_{i,j}$) value of a given element in $\zeta_{i \neq j}$ is given by:

$$AE_{i,j} = \left| \nu_{i,j} - \gamma_{i,j} \right|, where : i \neq j \tag{1}$$

Where, $\nu_{i,j}$ is the (i,j) non-diagonal coefficient of GT-NMF temporal correlations, and is $\gamma_{i,j}$ the corresponding (i,j) non-diagonal coefficient of the GT-GT temporal correlations. The average value ($\zeta_{avg}$) of all

these non-diagonal elements (Mean Absolute Error, MAE) and the standard deviation ($\sigma_\zeta$) is given by:

$$\zeta_{avg} = mean\left(\zeta_{i \neq j}\right) = \frac{1}{2(N_s - 1)} \left( \sum_{i \neq j}^{N_s - 1} \left| \boldsymbol{v}_{i,j} - \boldsymbol{\gamma}_{i,j} \right| \right) \qquad (2)$$

$$\sigma_\zeta = std(\zeta_{i \neq j}) \qquad (3)$$

Where $N_s$ is the total number of sources. These two values ($\zeta_{avg} \pm \sigma_\zeta$), together with the diagonal values ($\delta_{avg} \pm \sigma_\delta$), give us an estimation of whole experiment quality since ($\zeta_{avg} \pm \sigma_\zeta$) should ideally approach to zero.

## Data availability
All the data used for this manuscript have been deposited on the Zenodo database and is available under the accession code: https://doi.org/10.5281/zenodo.12087382. Raw movies for all the figures are also available as tiff files in the same repository.

## Code availability
Analysis scripts are available at: https://github.com/comediaLKB/DemixedFiberPhotometry (https://doi.org/10.5281/zenodo.12125030) https://github.com/RimoliCV/NMF_DemixedFiberPhotometry. Hardware control scripts are available at: https://github.com/laboGigan/SpeckledNeuronsControl.

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

## Acknowledgements

We are grateful to Serena Bovetti, Stefano Zucca, and Laurent Bourdieu for productive discussions about the applications of the concept to in-vivo imaging. We are very thankful to Walther Akemann for the mouse

brain extraction and fixation. In addition, we are grateful to Guillaume Dugué, Thomas Pujol, Arnaud Leclercq, and Maurice Debray for the fiber-ferrule assembly discussions and technical support (Lab. Kastler Brossel and IBENS' FabLab facilities, the latter which received support from the Fédération pour la Recherche sur le Cerveau - Rotary International France (2018)). This work was funded by the following grants: HFSP project N°RGP0003/2020 (S.G.), H2020 European Research Council (724473 SMARTIES) (S.G.) and France's Agence Nationale de la Recherche ANR-19-CE37-0007-02 (C.V.).

## Author contributions

C.V.R. adapted the previous setup and performed the optical experiments, prepared the samples, adapted the code, performed the NMF analysis, and wrote the article. C.M. built the excitation ground-truth part of the setup (DMD alignment) and wrote the Matlab control code. F.S. wrote the core code in Python for the NMF analysis. E.B. and C.V.R. assembled fiber-ferrules and performed experiments with the Mini-DART. C.V. and S.G. supervised and initiated the project. C.M. and S.G. conceived the idea. All authors discussed the results and commented on the paper.

## Competing interests

The authors declare no competing interests.
