## [Peer Review File · Nature Communications]

Demixing fluorescence time traces transmitted by multimode fibersREVIEWER COMMENTS

Reviewer #1 (Remarks to the Author):

Rimoli and Moretti et. al. describe a proof-of-principle approach to discover "point-like" fluorescent sources scrambled by the propagation of light through the multimode optical fiber. The manuscript is well written, the idea is simple and nice. Indeed, for many neuroscience applications we do not necessarily need the exact image of neurons. I support publication of this work, however not in the present form. Authors will need to do more experiments to substantiate claims for applicability to in-vivo setting. I fear the transition from these experiments to in-vivo might be a much more challenging step. I comment on this below.

Major concerns:

1. Line 81-86, could authors provide estimates of how many "scattering fingerprints" would be available for NNMF depending on the fiber length? In-vivo experiments may have a range of 5-15mm, assuming 5 mm is needed to mount in the optical system.

2. Line 91-92, authors claims here to validate the approach with a realistic conditions. However, in my opinion, realistic conditions include:

2.1 An attempt to model/mimic brain tissue scattering.

2.2. An attempt to model different density of "neuronal" labeling (I will comment on it below).

2.3. An attempt to model or at least have an estimate of how blood vessels dilation would scramble the expected steady-state patterns discovered by NNMF and would introduce additional unique NNMF patterns.

Therefore, before making a claim on the p 97- 98 that the method is "very suitable to be used in long-term freely-moving mice neuroscience experiments", I would suggest to do several experiments.

3. Proof of principle experiment in Figure 1 is nicely designed and gives the inquire opportunity to understand in-depth the advantages and the limitations of the proposed approach. First, could authors provide more details in the illustration: were excitation-emission filters used (it is important to separate strong excitation light from emission); did the camera have the objective (if so, what was the magnification) ? Please add as a schematic drawing to the Figure 1 all components. What was the LED light power per unit area for the excitation of each point source?

4. I would encourage authors to assess how many point sources could be resolved at once. For this, the proof of principle experiment in Figure 1 should be repeated with the sample containing higher density of beads, the scattering of the sample can be adjusted to closer reproduce the brain tissue (for example by embedding non-fluorescent spheres of different diameters into the sample. I am not an expert in how to mimic tissue scattering, but there are papers discussing this aspect) and, finally excite different proportion of fluorescent beads (at the same time) to create partially overlapping patterns and see how well NNMF would disentangle those.

4.1 To mimic the activity in the brain, groups of "neurons" (fluorescent beads) should be excited nearly simultaneously. Simultaneous excitation in the group might be difficult for NNMF, but you may give a variance in timing of approximately 5-20 ms within a group (to profit from this variance, the camera frame rate should be close to 100 fps). Groups of 1-5, 10, 15, 20, 30 sources should be excited in separate experiments . For example, for the iteration of 5 nearly simultaneously excited beads , I would use some time intervals for one 5-beads composition and another time interval for another 5-beads composition. Run NNMF and extract number of unique sources. Do next iteration with higher number of simultaneously excited beads.

From this experiment authors will be able to estimate: 1. number of simultaneously detected sources (at some moment in time) vs the true number of excited sources and 2. overall number of NNMF detected unique sources vs the true number of overall excited sources. It would be

important to know how well the approach can demix spatial modes when they overlap. Importantly, it is ok if it does not, but authors should identify and discuss this limitation in the text as it is crucial to in-vivo application. Also, if demixing large groups of neurons is not possible, the method should not be called high-throughput (standards for high-throughput are high).

4.2. Same experiment can be done for samples with scattering medium and without to compare NNMF performance.

5. Fig. 2-3 no reported excitation intensity per source.

6. I find correlations on figure 2-3 g-h to be not informative. It is of course nice that the pattern is similar. I would find more useful to report prediction accuracy of NNMF relative to GT. For example, training the model on GT (if the source activation is binary) and predicting with the model from the unseen NNMF data.

7. Line 188-194, discuss about saturation intensity is interesting. I wonder how much of the limitation that would be when you have multiple sources distributed across the fiber field-of-view?

8. Line 214-223, for the discussion a recent paper from Pisanello lab might be useful where they also attempt to gain spatial resolution (Bianco et al. "Orthogonalization of Far-Field Detection in Tapered Optical Fibers for Depth-Selective Fiber Photometry in Brain Tissue." *APL Photonics* 7, 2 (2022): 026106. <https://doi.org/10.1063/5.0073594>.) Their prediction accuracy along the taper length was still relatively low, probably due to the fact that imaging across many fiber modes (degrees of freedom) can deliver a lot of information?

9. Line 219-223, the ability to identify the displacement of the fiber relative to one bead is described as some sort of advantage, but in reality cylindrical symmetry of the fiber makes it difficult to identify multiple sources placed equidistantly from the fiber center (all would contribute to one ring). How big limitation would that be?

10. Finally, in multiple parts of the manuscript authors stress that they "expand the capacity of fiber photometry in resolving neuronal activity" (line 250-252 etc.) however at the moment I see little experiments supporting this statement. Majority of my suggestions for proof of concept experiment target those improvements and will guide the experimental choices for in vivo application. Ideally, I would like to suggest using miniscope-coupled to fiber approach in-vivo, where the fiber is inserted or implanted in the mouse brain above neurons expressing GCaMP calcium indicator. Authors, could record from such mouse, NNMF estimate sources, perfuse the mouse and count GCaMP-labeled cells and compare with NNMF result. Note, for such experiment better use somata-localized GCaMP, as labeled neuronal dendritic trees may essentially spread across all field-of-view. I also understand that in-vivo experiment is possible to do only in the lab dedicated to such work (this includes already obtained animal licenses and many other skills) and this lab may not be able to do that alone but only in collaboration. Therefore, I do not suggest this as a mandatory experiment, but to enhance trust and generate more interest in this approach.

Minor concerns:

1. It would be nice to cite earlier works (one of the papers from Papadopoulos et al. and from Turtaev et al.) which achieved spatial resolution by digitally scanning the excitation light at the fiber output.

Papadopoulos, Ioannis N., Salma Farahi, Christophe Moser, and Demetri Psaltis. "High-Resolution, Lensless Endoscope Based on Digital Scanning through a Multimodeoptical Fiber." *Biomedical Optics Express* 4, 2 (2013): 260–70. <https://doi.org/10.1364/BOE.4.000260>.

Papadopoulos, Ioannis N., Salma Farahi, Christophe Moser, and Demetri Psaltis. "Focusing and Scanning Light through a Multimode Optical Fiber Using Digital Phase Conjugation." *Optics Express* 20, 10 (2012): 10583–90. <https://doi.org/10.1364/OE.20.010583>.

Turtaev, Sergey, Ivo T. Leite, Tristan Altwegg-Boussac, Janelle M. P. Pakan, Nathalie L. Rochefort,

and Tomáš Čížmár. "High-Fidelity Multimode Fibre-Based Endoscopy for Deep Brain in Vivo Imaging." *Light: Science & Applications* 7, 1 (2018): 92. <https://doi.org/10.1038/s41377-018-0094-x>.

Reviewer #2 (Remarks to the Author):

The manuscript by Rimoli et al presents an experiment in which a short multimode optical fiber is used to collect and demix fluorescence from different fluorescent beads in a sample at the fiber output. The idea is that, in experiments in which fluorescent indicators, such as calcium or voltage indicators, are expressed in a neuronal population, a very simple optical setup, essentially composed by a short and minimally invasive optical fiber and a CMOS camera, can demix the signals from different neurons, as they come temporally sparse and have a defined spatial signature on the camera detector. The promise of the manuscript is that, with this simple idea, the broad neuroscience community that relies on fiber photometry to measure population activity in ensemble of neurons could now have the resolution to disentangle activity from individual neurons, and do that in a mouse that is freely moving and thus free to perform unrestrained behavioural task.

The manuscript applies several concepts already demonstrated in the group of S. Gigan (see for example Ref. 43, 44, 48 of the manuscript), which already allowed them to demix fluorescent signals from a scattering tissue (which results in fully developed speckle patterns), and to reconstruct the spatial locations of hidden objects by using correlation strategies on the different speckle patterns. Mathematically, the demixing is carried out by using a non-negative matrix factorization algorithm. In this manuscript, the authors apply the same strategy and the same algorithm to demix traces that come from non-developed speckle patterns, which are characteristic signatures of short multimode fibers. As the fiber is short, the spatial signature is supposedly not sensitive to bending, vibrations or conformational changes of the fiber, which is a strong advantage for freely moving studies.

Although I agree with the authors that changing a fiber photometry setup in something that can give single neuron resolution (essentially for free) by only adding a camera detector can be a game changer in neuroscience, my honest opinion is that the current manuscript is still far from demonstrating that. Indeed, no real biological experiment is performed, nor the reported results can be extrapolated to infer what would happen in a real in vivo experiment in the mouse brain. In my opinion, because the biology is not there, the current manuscript does not add too much to what the authors have previously demonstrated in scattering tissue (Ref 43, 44, 48), in the sense that, if demixing can be carried out through the mouse skull, then I would expect the same process to work through a multimode fiber too.

At the same time, since biological experiments are not carried out, I would have expected a more thorough explanation (with e.g. simulations) of the observed spatial signatures in terms of modes propagating through the fiber. As far as I understood, the relationship between the position of the beads with respect to the fiber centre and the detected fluorescence patterns, should be related to the spatial modes excited by the beads. However, this is not at all explained in the manuscript. A better description of the process would make the story stronger at least from the physics point of view.

Suggestions for possible changes.

From my point of view, the main limitation of the work is the absence of relevant experiments from which stronger conclusions on the performances of the system in real biological conditions can be inferred. The demonstration presented in Figure 03 is not sufficient, as only 4 neurons are selectively illuminated, whereas in a real case scenario, widefield illumination would excite several more neurons distributed in more planes. A more realistic scenario would at least involve a widefield illumination of the whole field of view (which can come from the fiber itself),

superimposed to the variable illumination sent by the DMD to selected neurons. To simulate the neuropil fluorescence variation, the widefield illumination should also vary in intensity over time. Ideally the authors should take a raw video recorded with a 1P miniscope with widefield illumination and try to reproduce with the DMD the variation of neuropil fluorescence and that of the neurons. This would give a better estimate of the performances of the device, and it would be best carried out in the miniDART configuration, as this is one of the main novelties of the manuscript.

I am aware that going for a full in vivo experiment in mice would probably take a very long time (even if one of the corresponding authors have already expertise in similar experiments). However, only a real biological experiment will ultimately allow one to judge on the use of the presented method. The experiment could perhaps be carried out in brain slices expressing GCaMP, or in smaller animals, such as drosophila or zebrafish. If biological experiments are out of question for this manuscript, the authors should at least provide the more relevant experiment I outlined before in the most convincing way possible. Even in that case however, I am not sure that with no biology at all, this work should be published in Nature Communications. A specialised optical journal would, in my opinion, be more appropriate.

The next main point that I think is missing in the current version of the paper is a more thorough explanation of the spatial signatures measured at the fiber output. As far as I understood the fluorescence signatures are a result of the excitation of different modes through the fiber. Spatially resolved fiber photometry was already demonstrated through tapered fibers by the groups of M. de Vittorio and F. Pisanello, albeit in that case the variations corresponded to variations in the z axis, rather than in the xy plane. If one takes for example this reference (Bianco et al. Orthogonalization of far-field detection in tapered optical fibers for depth-selective fiber photometry in brain tissue, APL Photonics 2022), the far field fluorescence patterns recorded on a camera detector display a characteristic ring like shape as a function of the z position (which also correspond to a different xy distance from the fiber center), which reminds me of the spatial signatures measured here by Rimoli et al. However, in Bianco et al, the signatures were measured in the Fourier plane of the fiber, whereas here the authors directly look at the fiber end surface. How are the two experiments linked to one another? My impression is that the overall effect should be pretty similar. I think that the authors should at least discuss, if any, the differences with respect to what already demonstrated through a tapered fiber, from a theoretical point of view, and in terms of advantages of their methods, so that the readers can better appreciate the novelty of their configuration. If the mechanism behind the two methods is really the same, I would expect a detailed description of why the current method is better than the previous one and which problems can it solve that could not be solved with a tapered fiber.

Other points:

- Related to a better explanation of the spatial signatures, theoretically, what is the maximum number of spatial signatures that the authors should be able to unmix. Or, in other words, with reference to the Supplementary figures 3-4, which is the minimal radial and angular distances that would generate sufficiently different spatial signatures?
- There is no mention in the manuscript of the z axis at the sample plane. Is the method designed to work at one particular z plane? What would happen to the spatial signatures if the authors would displace the sample in z? I think that a proper calibration of the instrument (e.g. repeating the measurements in Supp. Fig. 3-4 at different z) as a function of the focal plane would be a very good idea.
- Related to the previous point, an experiment in which different beads at different z are excited and their fluorescence unmixed would be interesting.
- Still related to the z aspect, if one increases the distance between the fiber tip and the sample, would it be possible to extend the xy field of view beyond the diameter of the fiber (with a loss in the detected signal)? It is hard to me to understand that the field of view always stays constant and equal to the fiber diameter no matter the z plane.
- The authors, contrary to their previous works, attribute all the scattering effect to the presence of the fiber and mode mixing. However, in a real experiment, in which fluorescence sources are buried at depth in scattering tissues, the fluorescence that hits the fiber tip would be also scrambled and not clearly assignable to a particular spatial location. In the extreme case of a very deep fluorescence source, the fiber tip would be hit by a speckle pattern, so that probably, many

different modes of the fiber would be excited by individual neurons. Do the authors expect to be able to demix such more complex fluorescence patterns? Or in this case, two very deep neurons (that could still be excited in 1P widefield excitation) would generate very similar spatial signatures at the fiber output?

- In Fig 2-3 the authors show signals that they send to the beads (or neurons) to simulate neuronal activity. However, the y scale in the traces is missing. What is the typical and minimal variation of fluorescence they can detect (the so called $\Delta F/F$)? It is not clear if the ground truth is a signal that is zero all the time (so the target is mostly not excited) and has a higher fluorescence value at certain (very short time), or if the authors allow for the resting fluorescence. In other words, neurons have typically a certain resting fluorescence, which is not zero, and sparsely increase their fluorescence value in correspondence to neuronal activity. The resting value would generate a continuous spatial signature background (similar to Fig.2a) on top of which different spatial signature would variate when more activity is present. Is this what the authors actually do in the experiment? If not, the authors should repeat the experiment by sending as ground truth raw data (not polished $\Delta F/F$ data) from 1P miniscopes.

- It would also be nice to see raw fluorescence videos of the experiments (e.g, of Fig 2).

In summary, I believe that the current manuscript represents a nice optical story, that adds up to the previous works of the group and previous works on fiber photometry. However, I believe that the biological conclusions the authors suggest, 'this work can open a whole new avenue for novel and affordable minimally invasive deep brain microendoscopic studies', are not demonstrated in the current manuscript. To me, it remains completely unknown what kind of patterns would one see in a real experiment in which many neurons expressing GCaMP (or voltage indicators) at multiple planes are excited simultaneously in a wide field manner through the fiber, and if the sensitivity of the demixing algorithm will allow different neurons to be resolved. In the absence of real biological demonstrations, in my opinion, the current work does not add sufficient novelty to what already shown by the group of Gigan (and possibly the groups of de Vittorio and Pisanello on tapered fiber), to be published in Nature Communications. I would instead recommend publishing it in an optical journal.

Reviewer #3 (Remarks to the Author):

The manuscript presents an innovative method for the separation of fluorescence signals through thin multimode fibers, and could in theory lead to a breakthrough in fiber photometry. In conventional fiber photometry, the signals transmitted through the multimode fibers become scrambled, thus restricting the ability to obtain temporal readings at the cellular level. The authors propose a solution to this problem by separating signals from several fluorescent sources by applying an unconstrained non-negative matrix factorization (NMF) algorithm directly to the raw video data. Because fibers are less bulky than GRIN lenses, this approach could ultimately be very useful for recording activity from deep brain areas.

I found this to be a very nice and creative idea which could make a good optics paper. However, there is a strong mismatch between the proof-of-principle demonstration and the claims of biological feasibility. This is potentially problematic in a multidisciplinary journal, because biologists would read this paper thinking that the magical solution is around the corner. This is possible, but the experiments shown here do not yet support this conclusion.

I believe there are two main paths to go forward:

- (1) remove all claims of biological feasibility as well as the suggestive but unphysiological results in sections 2.2. and 2.3. Discuss limitations in detail. Then publish a streamlined and still interesting (in my opinion) optics paper.
- (2) do more realistic biological work to support the claims of this being a useful tool for biology.

The work has potential and I look forward to seeing it after modifications, if the authors decide to take one of those paths.

Major:

At several locations the paper either explicitly claims or strongly suggests that biological applications are almost within reach, but the actual experiments don't support this. For example, figure 4e shows a mouse with an implanted miniscope and MMF, but the data panels in this figure are about radial displacement of beads in agarose.

Time is given in "a.u.". Is that a typo or does this mean that recordings were not made in real time (presumably at longer exposure times)?

It's a clever idea to use projection of temporal patterns onto beads as a first ground-truth evaluation (section 2.1), but the added benefit of repeating this approach with a fixed slice (section 2.2) is not clear. The slice is basically being used as a projection screen.

Similarly, the purpose of section 2.3. is not clear. It shows that data from sparse bright beads can be recorded and unmixed on a miniscope camera. Based on section 2.1 we already know that this approach works in general. Presumably the purpose of using a miniscope was to make conclusions about realistic imaging scenarios, but what new information do we actually get about realistic configurations when the samples are sparsely illuminated bright beads without background (see below)?

The scenarios that are supposed to mimic physiological conditions are quite unrealistic and some main limitations for in vivo imaging are not discussed. Most notably, the projected and unmixed traces seem almost noise-free. Real data has plenty of noise, mostly due to shot noise from a limited number of signal and background photons, which would very likely affect the ability to unmix the signals. As far as I can tell (not stated in the Methods), the ground truth illumination focused light on the sources only, avoiding any background fluorescence. Exposure times and illumination intensities are not given. I suggest a detailed study of how signal quality is affected by fluorescence background level, camera well depth limitations and labeling sparsity. At least, this should be treated in theory or by simulations with realistic parameters, but the manuscript would further benefit from realistic practical demonstrations. For example, authors could use a real miniscope video recording (published or recorded by the authors), tune light intensity to realistic levels and project it onto the input face of the MMF, while trying to unmix the signals on the other end.

The robustness of the approach against movement of the sample and against bending of the MMF should be evaluated.

Other:

Please check language and style throughout the document. The discussion is particularly notable for its language and requires editing.

"there is plenty of room for improvement, not only optically wise" (see also comment above), "simply add some extra data processing steps...". If the proposed improvements are as straightforward as the discussion suggests, it would be beneficial to see these implemented in the study. Alternatively, please rephrase.

The discussion is very verbose on possible improvements, but not much space is used to critically discuss key limitations for the main target application.

I suggest removing figure 4e unless you plan on including such recordings. Otherwise it can mislead the reader.

Rebuttal to the reviewers of Nature Communications

Initial message: the changes in the main text

The authors would like to thank the reviewers for all the insightful experiment suggestions and also the editors for accepting to postpone the rebuttal deadline, which allowed us to perform a series of new experiments aiming at addressing the main criticisms of the reviewers to the best of our possibilities. Before giving a point-by-point answer to the reviewers' comments, we first want to give a general overview of the new results we present and their motivation.

A general comment across the review is the lack of biological verification. We agree with the reviewers that *in vivo* data would be ideal to give a final piece of evidence to this manuscript. However, the method we introduce here is very novel from the conceptual and optical design point of view, and we believe that a thorough proof-of-principle demonstration *in vitro* is already a very important step. We strived throughout our manuscript to keep the biological constraints in mind. We believe we overall provide a compelling case for the general interest of our approach, and its potential to be useful in miniature endoscopy experiments. However, to perform a proof-of-principle experiment *in vivo* is well beyond the scope of this work, and would require resources and facilities well beyond our capabilities. An ideal *in vivo* experiment design would have to include simultaneous genetically encoded calcium indicators (GECI) recordings at that same region (same FoV) in a living mouse brain: one video recording to image the ground truth (conventional imaging method, such as 2-photon fluorescence imaging) and another recording with the multi-mode fiber (MMF) (MiniDART). Otherwise, one does not have access to the ground-truth activity in the brain to validate our method. Thus, we chose an alternative and robust way to prove the idea and test possible limitations of the method: we generated the ground truth *in vitro*, mimicking the brain activity with synthetic GECI time trace profiles generated by MLSpice from real *in vivo* electrophysiology data (more details about how do we generate the analog synthetic time traces with binary modulation of the DMD can be found in previous works from our team (Moretti C. and Gigan S. Nature Photonics (2020), <https://doi.org/10.1038/s41566-020-0612-2>). An *in vivo* demonstration is definitely on our roadmap, we are currently discussing with potential collaborators to carry out these experiments, but we believe this is well beyond the scope of the current work. We hope the current paper, thanks to its compelling concept and the simplicity of the setup, will also motivate neuroscientists to test our method and hopefully adopt it widely.

We do agree that our first reported results, although they demonstrated the concept, failed short on addressing some major concerns about the applicability in the brain. We thus followed the suggestions of the reviewers to further substantiate the claims of our work and we designed and performed novel *in vitro* experiments aiming at mimicking more realistically the real brain. In particular, the current version of the manuscript now includes results from novel experiments that can be found in Figure 02.2 and Figure 02.3 in the main text, reporting on much higher densities and number of sources, and imaging through a scattering layer. Besides, there are some extra details on supporting info, such as Figure S07, where we show that NMF naturally denoises the spatial fingerprint of the sources. We will clarify all the changes below. The results from 6 beads (Figure 02.1) and fixed neurons (Figure 03) in a fixed brain slice are still in the main text, with extra details (as suggested by the reviewers).

We first detail the changes in the manuscript, and then we give a point-by-point answer to the reviewers' questions. Please note that all the changes in the manuscript have been highlighted in blue.

Updated Figure 01: New illustration of the idea behind the proof-of-principle setup

The main modifications of this first figure were: (1) we added the ensemble time trace (fiber photometry, black time trace) below each corresponding superimposed pattern (#1, #2, #3 pattern signals). In addition, we added the non-negative matrix factorization (NMF) optimization equation on the bottom left, with labels (h,w) to be associated with the spatial fingerprint (w(s)) and time traces (h(t)). The arrow pointing down represents the signal decomposition via NMF demixing.

Figure 01 – Concept of **single-source resolved fiber photometry (demixed fiber photometry)**. From left to right: ground-truth excitation mimicking neuronal activity is performed by using a DMD, which can selectively excite a set of fluorescent emitters on the sample with a given time trace, likewise in^{50,51}. A short (8 mm long) multimode fiber typically implanted in optogenetics or fiber photometry experiments (NA = 0.39, 200 μm core diameter, step-index fiber) is placed almost touching the sample (distance of ≈ 50 μm) to collect the fluorescence dynamics of each source. Due to its proximity to the sample, the fiber's effective FoV is expected to be slightly larger than the core size. Fluorescence light inside the multimode fiber is subject to multimodal mixing during propagation, which scrambles/mixes the emitters' wavefront similarly to any scattering media. The transmitted superimposed signal (**#1, #2, and #3**) consists of fluorescence transient patterns, i.e., 2D patterns (**w1, w2,w3**) that fluctuate in intensity over time **with typical calcium transient profiles (h1,h2,h3)**^{65,66}. A video is recorded with a camera and a post-processing step using a spatio-temporal demixing algorithm (unconstrained NMF) is applied to disentangle the overlapped transient patterns into individual 2D spatial fingerprints and their corresponding singular time trace profiles that should match the GT excitations. **For optical setup and raw data videos details, see supp info.**

Minor changes in Figure 02.1: the proof-of-principle result with 6 beads

In the new version of this figure, we added the fiber photometry (ensemble) time trace to be compared with the individual time traces retrieved via NMF. It is worth noting that in the description of this figure (in the main text), we tried to make it clear that we quantified the amount of temporal trace cross-talk among the retrieved time traces taking care of subtracting the average ground-truth intrinsic correlations (GT-GT). **A tiff file of the video we used as input data in the unconstrained NMF was added to the supp info.**

Figure 02.1 – Results of a proof-of-principle experiment performed with 6 fluorescent beads. From (a) to (d) we have: (a) the ensemble temporal activity (fiber photometry), (b) the superimposed pattern image of 6 fluorescent bead fingerprints when simultaneously excited (imaged detected via sCMOS camera (see methods and supp info)); (c) the short MMF located at a distance of $60 \pm 10 \mu\text{m}$ from the fluorescent beads; (d) a CMOS Basler camera with the ground truth image of the sample (see supp info for setup scheme details). (e) The ground truth (GT) fingerprint patterns obtained from each bead when they were individually excited. (f) The fingerprint patterns obtained via NMF demixing are to be compared with the GT patterns in (e). (g) The individual temporal activity traces of the sources obtained with NMF (blue) and their corresponding GT traces (gray). The NMF trace (#6) was not recovered well by NMF since bead #6 was localized very close to the fiber core edge, therefore yielding low signal/contrast of its pattern (see GT scattering fingerprint of bead #6 in (e)). (h) The GT-NMF time trace correlations. The average diagonal value of the first 5 beads was $\langle \delta_{g,n} \rangle = \delta_{\text{avg}} = 85.4\%$ with $\sigma_{\delta} = 3.6\%$. To better evaluate the off-diagonal elements (time trace cross-talk), we subtract them from their corresponding GT-GT coefficients. Then, we averaged the absolute values of these differences and we obtained the mean absolute error of $\zeta_{\text{avg}} = 7.06\%$ with a standard deviation of $\sigma_{\zeta} = 7.29\%$ for the first 5 beads (see supp info). (i) The GT-GT temporal trace correlation table. Importantly, the GT-GT correlation coefficients show that although each GT trace was unique over time (singular profile), GTs from different sources were not fully uncorrelated. For example, GT traces of beads #3 and #4 were fairly correlated ($v_{3,4} = v_{4,3} = 31.2\%$, in (i)) and had a very clear spatial overlap (see GT and NMF scattering fingerprints #3 and #4 in (e) and (f)).

New Figure 02.2: multiple beads in agarose + neuropil

The new Figure 02.2 shows the new results when we challenged NMF by recording signals from tens of sources embedded in agarose (where a few of them are touching each other and have similar radial distances from the central axis of the fiber). The GT temporal traces had multiple overlapping transient peaks and the ensemble signal exhibits rapid fluctuations. We also mimicked the effect of the neuropil, by including one fluorescence source (bead #20) that had a spatiotemporal signal overlapped with many other probed sources. As a consequence, the neuropil signal has a non-sparse and noisy dynamic activity background whose signal is continuously non-zero and of the same order of magnitude as the other sources. NMF algorithm managed to successfully demix the time traces of 22 out of 26 sources, including the neuropil, even when the beads were touching each other (which were more challenging to demix because they had very similar fingerprint patterns). Time traces that were not retrieved well by NMF were due to sources close

to the fiber core edge (likewise the previous experiment with 6 beads, in Figure 02.1). A tiff file of the video we used as input data in the unconstrained NMF has also been added to the supp info.

Figure 02.2 – Results of a proof-of-principle experiment performed with 26 fluorescent beads including a neuropil background source. The bead #20 is modeling the neuropil. From (a) to (d) we have: (a) the photometric (ensemble) time trace, which is the sum of 26 time traces; (b) the sCMOS detected image of the spatially overlapped fingerprint patterns from 26 fluorescent beads probed by the short MMF (see methods); (c) the short MMF located at a distance of $60 \pm 10 \mu\text{m}$ from the sample; (d) the ground truth image of the sample (backpropagated fluorescence image detected from a CMOS Basler camera, see details of the setup in the supp info). (e) The ground truth (GT) fingerprint patterns obtained from each bead when they were individually excited. (f) The fingerprint patterns obtained via NMF are to be compared with the GT patterns in e. (g) Top: the individual temporal activity traces obtained with NMF (blue) and their corresponding GT traces (gray). Bottom: the photometric ensemble signal from the recorded video (black line), which is the sum of all individual traces. The fluorescence intensity in all traces in the figure are normalized to 1. (h) The GT-NMF time trace correlations. The average diagonal value of the first 22 beads was $\langle \delta_{g,n} \rangle = \delta_{avg} = 86.0\%$ with $\sigma_{\delta} = 5.4\%$. To better evaluate the off-diagonal elements (time trace cross-talk), we subtract them from their corresponding GT-GT coefficients. Then, we averaged the absolute values of these differences and we obtained the mean cross-talk of $\zeta_{avg} = 4.4\%$ with a standard deviation of $\sigma_{\zeta} = 3.7\%$ for the first 22 beads (see supp info). (i) The GT-GT temporal trace correlation table showing that the ground truth traces were not orthogonal.

We expect this set of experiments (Figure 02.2) to mimic well a case where the probed neurons are very close to the fiber tip (for which light undergoes little scattering). It is worth noting that the agarose medium in which the beads are embedded has a small scattering effect on the fluorescence light. However, since the fingerprint patterns obtained in this new set of experiments from multiple beads in agarose (the new Figure 02.2) were virtually the same as the fingerprint patterns from the 6-bead experiment (Figure 02.1, without agarose) and from the patterns of fixed neurons the thin $50 \mu\text{m}$ brain slices we had before (Figure 03), we decided to get one step closer to the situation of a real brain by introducing a scattering layer between the beads and the fiber. Indeed, as mentioned by the reviewers, brain tissue is very scattering, and it would be good to know if NMF could demix signals at some distance from the fiber tip from sources buried below some scattering layer. This is what we show next.

New Figure 02.3: multiple beads in agarose + neuropil signal from sources buried below Parafilm M® (scattering layer)

The new Figure 02.3 shows the results when we challenged our method by adding a 120 μm layer of a plastic Parafilm M® (scattering media) in between the bead sample and the MMF tip. We used the same fluorescent bead sample (with a high density of beads embedded in agarose) and the same GT time traces (including the neuropil) as in Figure 02.2. Moreover, this time we searched in the sample for a FoV region where we could find several beads touching each other and we centered the fiber so that they had very similar radial distance about the fiber axis (to obtain very similar patterns). Again, the NMF algorithm managed to successfully demix most of the time traces (20 out of 26 sources). Here, most of the beads were touching each other and had a pattern overlapping with the neuropil (which was more challenging to demix). Interestingly, due to the scattering layer (Parafilm M®), patterns in this experiment had different morphological details (less symmetric) than without Parafilm M®, indicating that scattering may help in making the signal from sources buried below a scattering layer more easily separable. Again, time traces that were not retrieved well by NMF were mostly due to sources close to the fiber core edge (likewise the previous experiments without Parafilm M®, in Figure 02.1 and Figure 02.2). **A tiff file of the video we used as input data in the unconstrained NMF has been added to the supp info.**

Figure 02.3 – Results of a proof-of-principle experiment performed with 26 fluorescent beads behind a parafilm layer (thickness : 120 μm). The bead #13 is the source mimicking neuropil background signal. From (a) to (d) we have: (a) the photometric (ensemble) time trace, which is the sum of 26 time traces; (b) the sCMOS detected image of the spatially overlapped fingerprint patterns from 26 fluorescent beads simultaneously probed by the short MMF (see methods); (c) the short MMF located at a distance of $60 \pm 10 \mu\text{m}$ from the sample; (d) the ground truth image of the sample (backpropagated fluorescence image detected from a CMOS Basler camera, see setup in the supp info). (e) The ground truth (GT) fingerprint patterns obtained from each bead when they were individually excited. (f) The fingerprint patterns obtained via NMF are to be compared with the GT patterns in (e). (g) Top: the individual temporal activity traces obtained with NMF (blue) and their corresponding GT traces (gray). Bottom: the photometric signal from the recorded video (black line), which is the sum of all individual traces. The fluorescence intensity in all traces in the figure are normalized to 1. (h) The GT-NMF time trace correlations. The average diagonal value of the first 20 beads was $\langle \delta_{g,n} \rangle = \delta_{avg} = 77.0\%$ with $\sigma_{\delta} = 11.9\%$. To better evaluate the off-diagonal elements (time trace cross-talk), we subtract them from their corresponding GT-GT coefficients. Then, we averaged the absolute values of these differences and we obtained the mean cross-talk of $\zeta_{avg} = 5.9\%$ with a standard deviation of $\sigma_{\zeta} = 5.6\%$ for the first 20 beads (see supp info). (i) The GT-GT temporal trace correlation table showing that the ground truth traces were not orthogonal - some of them were correlated.

Minor changes in Figure 03: the proof-of-principle result in a fixed brain slice

In the new version of this figure, we added the fiber photometry (ensemble) time trace to be compared with the individual time traces retrieved via NMF. It is worth noting that in the description of this figure (in the main text), we tried to make it clear that we quantified the amount of temporal trace cross-talk among the retrieved time traces taking care of subtracting the average GT-GT correlations. **A tiff file of the video we used as input data in the unconstrained NMF was added to the supp info.**

Figure 03 – Validation of the concept of single-activity resolved fiber photometry with short multimode fibers (short MMF) **in a brain tissue environment (*in vitro*)**. Sample: Gad-EGFP neurons fixed in a 50 μm brain slice, sealed in between 2 coverslips to keep the humidity of the tissue (see methods). From (a) to (c) we have: (a) the fiber proximal end image of 4 neurons' fingerprint patterns spatially overlapped on the sCMOS camera chip; (b) an illustration of the short MMF placed above the top coverslip of the sample, at a distance of $\approx 60 \pm 10 \mu\text{m}$ from it; and (c) the GT image of the sample highlighting the 4 selected neurons to be excited (structurally labeled). (d) The GT fingerprint patterns are obtained from each neuron when individually excited. (e) The fingerprint patterns retrieved via NMF are in good agreement with the GT patterns in (d). (f) The demixed temporal activity traces are sorted in descending GT-NMF correlation order. Traces in blue are retrieved by NMF and temporal traces in gray are their GT. (g) The GT-NMF temporal trace correlation coefficients. The average diagonal value of the 4 neurons was $\langle \delta_{g,n} \rangle = \delta_{\text{avg}} = 86.7\%$, with standard deviation of $\sigma_{\delta} = 2.8\%$. Regarding the non-diagonal elements **cross-talk**, the mean absolute error **taking into account** the GT-GT coefficients was $\zeta_{\text{avg}} = 8.95\%$ with a standard deviation of $\sigma_{\zeta} = 8.02\%$ (see supp info). Again, although each GT trace was unique in time (singular), they were not fully uncorrelated as we can see in the GT-GT correlation traces (h). Interestingly, neurons #1 and #2 (i.e., the two best NMF retrieved results) were also the most temporally correlated ones in the GT excitation ($\nu_{1,2} = \nu_{2,1} = 25.0\%$).

Minor changes in the Figure 04: the miniscope sensitivity results and MiniDART concept

In this version of Figure 04, we present the same data as before, but when we introduce the MiniDART concept (where we exemplify how the method would work in living mice experiments), we added a clear description that it is a prospect to avoid a misleading interpretation that the patterns we show are from *in vivo* experiments.

Figure 04 – Novel microendoscopy concept using a short MMF and a miniscope: the MiniDART. (a) A typical fingerprint pattern from a single-fluorescent bead ($10\ \mu\text{m}$ diameter) probed by using only the miniscope excitation and miniscope detection through the multimode fiber. (b) The experimental setup to probe scattering patterns from the short MMF includes (i) the miniscope, (ii) a customized titanium base plate (YMETRY®) to hold the miniscope, (iii) a ferrule (Thorlabs SFLC230-10) that rigidly holds the multimode fiber (iv) within it, (v) a sample consisting of a single fluorescent $10\ \mu\text{m}$ bead (spatial density $< 1\ \text{bead}/\text{cm}^2$), and (vi) a customized titanium tweezer (YMETRY®) to hold the ferrule. (c) **Scattering fingerprints patterns at the proximal end (bottom row) depending on the radial position d of the single-bead at the distal end. Position d is indicated on the axis at the top of the figure, and is represented as a red arrow on the distal end pictures (with the bead indicated as a blue dot). Each pattern acquisition in the proximal end corresponds to $10\ \mu\text{m}$ steps from the fiber central axis (black dot in the center at distal end images). The bigger the distance d , the wider the diameter of the bright spiral-ring pattern (of radius p in the proximal end, white arrow). The diagonal white dashed line is the azimuthal orientation of the vector d , which always coincides with the alignment angle of the 2 central bright points of the fingerprint patterns in the proximal end (see supp info Figure S03-S04 for details). The highest LED power values measured at the distal end of the fiber (whose core is $200\ \mu\text{m}$ in diameter) were around $9.5\ \mu\text{W}$, which yields an excitation intensity of $0.3\ \text{mW}/\text{mm}^2$ at the output of the fiber core, and $2.4 \times 10^{-5}\ \text{mW}$ excitation power per bead area. Exposure time: $100\ \text{ms}$ (Miniscope FPS = $10\ \text{Hz}$). For more details, see Figures S03 and S04 in the supporting information.** (d) The concept of doing experiments with a MiniDART device, which combines a miniscope and a short implantable multimode fiber. For future *in vivo* experiments, the MMF and miniscope baseplate should be glued on the mouse skull with dental cement in the same way typical miniscope experiments are performed with GRIN lenses.

The new supplementary Figure S07: Spatial denoising feature of NMF

Finally, in the supporting info, Figure S07 displays the images comparing GT and NMF patterns in more detail, where we could detect some degree of image denoising due to NMF intrinsic optimization process. We had observed this denoising effect in the previous experiments, but only after carrying out more complex noisier experiments suggested by the reviewers could we confirm it. In all experiments, NMF fingerprints seemed to have more contrast than the GT patterns. Since NMF attempts to extract the GT fingerprint pattern over many frames, we hypothesize that it can retrieve spatial features with higher spatial contrast than a GT (single-shot frame of one single source) for sufficiently long video recordings. Such denoising effect of NMF is well-documented in the literature (Aonishi, T. *et al.* Neuroscience Research (2022) <https://doi.org/10.1016/j.neures.2021.12.001>. Varghese, K. *et al.* 3rd International Conference for Convergence in Technology (2018) <https://doi.org/10.1109/I2CT.2018.8529796>. Lin, B. *et al.* IEEE International Geoscience and Remote Sensing Symposium, (2018) <https://doi.org/10.1109/IGARSS.2018.8517388>. In particular, NMF has already been used to simultaneously denoise, deconvolve and extract time traces from calcium imaging experiments (Pnevmatikakis, E.A. *et al.* Neuron (2016). <https://doi.org/10.1016/j.neuron.2015.11.037>).

Figure S07 – NMF denoising effect on the retrieved spatial signal. Top: the ground-truth (GT) images. Bottom: the respective NMF fingerprint pattern. (a) Left column: GT-NMF patterns from bead #10 of Figure 02.2 (experiment without Parafilm M®). (b) Central column: GT-NMF patterns from bead# 09 of Figure 02.3 (experiment with Parafilm M®). (c) Right column: GT-NMF patterns of neuron #03 of Figure 03 from the main text (fixed brain slice experiment with Gad EGP labeled neurons). It is well-documented in the literature that NMF can denoise image data and we hypothesize that a long video recording can help NMF in denoising the pattern photometry data (Aonishi, T. et al. Neuroscience Research (2022); Varghese, K. et al. 3rd International Conference for Convergence in Technology (2018); Lin, B. et al. IEEE International Geoscience and Remote Sensing Symposium, (2018)). Importantly, NMF has already been used to simultaneously denoise, deconvolve and extract time traces from calcium imaging experiments (Pnevmatikakis, E.A. et al. Neuron (2016)).

Now we answer the reviewers' questions point-by-point:

Reviewer #1 (Remarks to the Author):

Rimoli and Moretti et al. describe a proof-of-principle approach to discover "point-like" fluorescent sources scrambled by the propagation of light through the multimode optical fiber. The manuscript is well-written, the idea is simple and nice. Indeed, for many neuroscience applications we do not necessarily need the exact image of neurons. I support publication of this work, however not in the present form. Authors will need to do more experiments to substantiate claims for applicability to in-vivo setting. I fear the transition from these experiments to in-vivo might be a much more challenging step. I comment on this below.

Major Concerns.

R1.Q1 - Line 81-86, could authors provide estimates of how many "scattering fingerprints" would be available for NNMF depending on the fiber length? In-vivo experiments may have a range of 5-15mm, assuming 5 mm is needed to mount in the optical system.

We used fibers of different lengths ranging from ~4 mm to ~10 mm in our initial experiments and we found that patterns were very similar for all fibers (data not shown). This result is not surprising because it is known that large modal dispersion effects start to be significantly present in multimode fibers whose length is one order of magnitude longer than the ones we used (Ploschner M. et al 2015 Nat Photonics). Thus, it is expected that experiments with multimode fibers with the typical length range of 5-15 mm long should yield very similar results.

We show in Figure 4 how patterns depend on the position of the source for a unique source. Patterns from 2 sources that are displaced by 10 μm along a radius of the fiber are already very different. In addition, we show that patterns of 2 sources that are localized with similar radial distances but different azimuthal angles around the fiber axis are also different (see patterns of beads 14 and 16 in Figure 02.2), and are well-demixed. In general, we show in Figure 02.2 that we can demix 10 μm diameter sources that are touching each other, indicating that their corresponding patterns are sufficiently different. Therefore, the number of available "scattering fingerprints" should be on the order of hundreds for fiber lengths of 5-15 mm whose fiber core has a similar area as the one we used. With this type of fibers, we show that we are already able to demix more than 20 sources, with and without a scattering layer between the sources and the fiber tip, because their spatial fingerprints are sufficiently different (Figure 02.2 and Figure 02.3). Interestingly, however, we demonstrate that the presence of a scattering layer changes the spatial features of the fingerprint patterns, making them less similar to each other (less symmetric). This feature could facilitate our method in demixing time traces and suggests that it is possible to tune the total number of the probed sources if one decides to change the optical propagation properties of the system by engineering the implantable fiber. For example, one could add a scattering layer on the fiber distal end (the tip that faces the brain) by controlling its degree of polishment, or by simply choosing a fiber with different intrinsic propagation properties, such as NA, core geometry, etc.

This last paragraph has been added in Supp Info 09 : Number of available scattering fingerprints.

R1.Q2 - Line 91-92, authors claims here to validate the approach with a realistic conditions. However, in my opinion, realistic conditions include:

2.1 An attempt to model/mimic brain tissue scattering.

2.2. An attempt to model **different density of "neuronal" labeling** (I will comment on it below).

2.3. An attempt to model or at least have an estimate of **how blood vessels dilation would scramble the expected steady-state patterns discovered by NNMF and would introduce additional unique NNMF patterns**.

Therefore, before making a claim on the p 97- 98 that the method is "very suitable to be used in long-term freely-moving mice neuroscience experiments", I would suggest to do several experiments.

The authors thank reviewer 1 for all these insightful experiment suggestions. As we mentioned at the beginning of this response, the current version of the manuscript now includes results from experiments in which we attempted to mimic the brain tissue in a more realistic manner. We now validate our method by performing experiments with tens of sources embedded in agarose (where a few of them are touching each other), with multiple overlapping temporal transient peaks signals, and by choosing one source to mimic fluorescent neuropil (i.e., a non-sparse temporal signal whose amplitude is of the same order of magnitude as the other sources) (Figure 02.2). The neuropil bead was chosen to be a source whose spatial fingerprint pattern would overlap with many other sources (bead #20) in the FoV of the fiber's proximal end. We showed that a simple unconstrained NMF algorithm successfully retrieved 22 out of 26 spatiotemporal sources from the sample. We then repeated this experiment by introducing a scattering layer of Parafilm M® between the fiber and the sample (Figure 02.3). In this case, we showed that we were able to successfully retrieve 20 out of 26 sources. We discuss in the manuscript the reasons why only some of the 26 spatiotemporal signals were retrieved well :

"NMF was capable of demixing most of those signals (20 time traces out of 26 sources) as we can see in Figure 02.3. As expected, the scattering patterns from this experiment (Figure 02.3) are different from the previous cases (Figure 02.1 and Figure 02.2). They became less symmetric, confirming that the scattering layer is breaking the cylindrical symmetry of the system (which should facilitate NMF demixing), but spatially noisier (i.e., with decreased intensity contrast). The latter might explain why some sources

could not be demixed well even though they were close to the fiber center (see beads #23 and #24). Indeed, those beads they had a strong spatial overlap with the neuropil (bead #13), which made the signals more difficult to demix. Nevertheless, based on previous works, it is reasonable to expect that a longer acquisition would help on this issue because there would be more frames to be used in the matrix decomposition [Moretti C. and Gigan S. Nature Photonics 2020].

Interestingly, however, NMF is widely employed to denoise image datasets [Pnevmatikakis, E.A. *et al.* Neuron (2016), <https://doi.org/10.1016/j.neuron.2015.11.037> ; Zhou P. *et al.* eLife (2018) <https://doi.org/10.7554/eLife.28728> ; Aonishi, T. *et al.* Neuroscience Research (2022), <https://doi.org/10.1016/j.neures.2021.12.001> ; Varghese, K. *et al.* 3rd International Conference for Convergence in Technology I2CT (2018), <https://doi.org/10.1109/I2CT.2018.8529798> ; Lin, B. *et al.* IEEE International Geoscience and Remote Sensing Symposium (2018), <https://doi.org/10.1109/IGARSS.2018.8517388>], and the new results from the Figure 02.3 strongly suggests that NMF is inherently denoising our data. Such denoising effects were already observed in the previous figures (see the comparison of GT and NMF patterns in Figures 02.1 and 02.2). A few examples have been further highlighted in Figure S07 in the supporting information). Consequently, our findings suggest a multifaceted role for NMF, wherein it not only demixes signals and performs image segmentation (fingerprints), but also naturally denoises the data, thereby exemplifying its potential in simultaneously addressing multiple aspects of fluorescence imaging data extraction [Pnevmatikakis, E.A. *et al.* Neuron (2016) ; Zhou P. *et al.* eLife (2018).”

With these two new experiments, we addressed questions 2.1 and 2.2, showing that the technique works in a scattering medium with a higher density of sources.

In vitro experiments mimicking the presence of blood vessels dynamically absorbing or scattering light, are more difficult to implement, but it is well known from the literature that methods using wavefront shaping techniques and MMFs are capable of focusing light in the living mouse brain [Stibúrek, M. *et al.* Nature Communications (2023) <https://doi.org/10.1038/s41467-023-36889-z>; Park J-H. *et al.* PNAS (2015) <https://doi.org/10.1073/pnas.1505939112> ; Blochet B. *et al.* PNAS (2023) <https://doi.org/10.1073/pnas.2305593120>; Cao H. *et al.* Adv. Opt. Photon. (2023) <https://doi.org/10.1364/AOP.484298>]. These results strongly indicate that light patterns are resilient to dynamic scattering by blood vessels, which should also apply to the source fingerprints in our experiments - especially considering that the working distance will be small (a few tens of microns up to 100 μm) and that the density of blood vessels is small.

A video showing a continuous change of the fingerprint patterns due to the movement of a single bead source is now added in the supp info.

R1.Q3 - proof-of-principle experiment in Figure 1 is nicely designed and gives the inquire opportunity to understand in-depth the advantages and the limitations of the proposed approach. First, could authors **provide more details in the illustration**: were excitation-emission filters used (it is important to separate strong excitation light from emission); did the camera have the objective (if so, what was the magnification) ? Please add as a schematic drawing to the Figure 1 all components. What was the LED light power per unit area for the excitation of each point source?

The authors appreciate the comments of reviewer 1 about Figure 01. This figure was designed to give a general qualitative insight into the physical phenomenon of the method and to give a simple visual introduction. As we mentioned above, we made some subtle changes to Figure 01, trying to make it even clearer the NMF decomposition process from the ensemble to individual spatiotemporal fluorescence signals. However, all the technical details of the experimental setup, including the specification of all the optical components, were already present in the support information of this manuscript. We invite the reviewer to look at Figure S01 (below) which gives all those details.

Figure S01 - The proof-of-principle optical setup. A 473 nm blue laser (LSR-0473-PFM-00100-01, Laserglow Technologies) illuminates a digital micromirror device surface (DMD from Texas Instruments: DLP LightCrafter 6500, same as ⁴³), that was used to create the ground truth (GT) excitation of each fluorescence source present in the sample plane (see sample details in the methods). A tube lens (L1, LA1708-A, Thorlabs) and the lower objective OBJ 2 (Plan-NEOFLUAR × 20, 0.5 NA, Zeiss) were used in the excitation path. For the ground truth excitation, genetically encoded calcium indicator (GECI) traces were obtained from neuronal recordings available in the literature, as we did before in previous experiments ^{43,57}. After excitation, the fluorescent signal propagated through the short multimode fiber (see fiber-ferrule preparation details in the methods) and the fluctuating fluorescent scattering fingerprints on the tip of the fiber were imaged onto a scientific complementary metal-oxide-semiconductor (sCMOS) camera (Iris 15 sCMOS, Teledyne Photometrics) by a microscope objective (OBJ 1, RMS10X PLAN ACHROMAT 0.25NA, Olympus) and a tube lens (TL 1). An emission bandpass filter (MF530-43, Thorlabs) was used in the detection path to block the blue excitation light. In addition, the system has a control path to image the fluorescence sample directly, so that it would be possible to obtain the ground truth image of the sample without passing through the fiber. This is done by imaging the backpropagating fluorescent light from the sample in reflection mode. For this path, a dichroic beam splitter (DM, FF496-SDi01, Semrock) was used to collect the fluorescent signal in reflection and the sample plane was imaged onto a CMOS camera (ACE2014-55um, Basler) after another tube lens (TL 2). **We typically used a laser power of $3.0 \cdot 10^{-8}$ W per $10 \mu\text{m}$ diameter area (bead size), resulting in a local intensity of $3.8 \cdot 10^{-2}$ W/cm². Synthetic fluorescence traces were generated like previous work from our team, using the spike activity dataset from real available datasets acquired in mouse visual cortex, and converting them into calcium (excitation) traces (MLSpike from Vanzetta's group: Deneux, T., et al. Accurate spike estimation from noisy calcium signals for ultrafast three-dimensional imaging of large neuronal populations *in vivo*. *Nat Commun* 7, 12190 (2016). <https://doi.org/10.1038/ncomms12190>) using a GCaMP6s physiological model and resampled at 10 Hz. The analog calcium transient profiles were designed by changing the dwell time of the excitation in relation to a longer, but constant, detection window time - similarly to previous works from our team. The DMD can control the excitation beam fast ($>10\text{kHz}$) concerning the detection window, allowing it to mimic an analog time trace profile very smoothly in the video recording. A higher (resp. lower) fluorescence intensity from a given bead corresponds to a longer (shorter) excitation dwell time during a given detection window. The detection window in the experiments were: 500 ms (Figure 02.1 - the 6 bead experiment, camera dynamic range of $\sim 10,000$), 1000 ms (Figure 02.2 - the 26 bead embedded in $50 \mu\text{m}$ thick agarose layer sandwiched between coverslips, camera dynamic range of $\sim 1,500$), 2000 ms (Figure 02.3 - the 26 bead embedded agarose layer sandwiched between coverslips + single Parafilm M[®] layer on top, camera dynamic range of $\sim 1,500$), 4000 ms (Figure 03 - 4 neurons in $50 \mu\text{m}$ brain tissue slice of the Gad-EGFP labeled neurons from the cortex, camera dynamic range of ~ 450). Therefore, most of the experiments used a moderate dynamic range compared to the available 16-bit.**

We now included in the supplementary information S01 the power per unit area for the excitation of each point source was $3,8 \times 10^{-2}$ W/cm² (ie. 0.38 mW/mm², corresponding to $3,0 \times 10^{-8}$ W for each source). This power is similar to what is used in optical recordings with one-photon imaging, which is around 0.05 - 0.20 mW transmitted by a GRIN lens (typically with ~500 μ m diameter core), whose is equivalent to 0.25-1 mW/mm², (see Resendez, S. L. *et al. Nat Protoc* (2016) <https://doi.org/10.1038/nprot.2016.024>). This information has been included in the caption of Figure S01 (highlighted in blue). Concerning the miniscope settings in the sensitivity experiments, the reviewer can find in the supporting information S03 - miniscope detection sensitivity - all the values for excitation and detection settings to detect a pattern from a single bead source. The highest LED power values measured at the distal end of the fiber (whose core is 200 μ m in diameter) were around 9.5 μ W, which yields an excitation intensity of 0.3 mW/mm² at the output of the fiber core, and 2.4×10^{-5} mW excitation power per bead area. Miniscope exposure time ranged from 100ms to 33ms to detect a single bead source with such relative low LED power (Figure S03).

R1.Q4 - I would encourage authors to assess **how many point sources could be resolved at once**. For this, the proof-of-principle experiment in Figure 1 should be repeated with the sample containing higher density of beads, the scattering of the sample can be adjusted to closer reproduce the brain tissue (for example by embedding non-fluorescent spheres of different diameters into the sample. I am not an expert in how to mimic tissue scattering, but there are papers discussing this aspect) and, finally excite different proportion of fluorescent beads (at the same time) to create partially overlapping patterns and see how well NMF would disentangle those.

Q4.1 To mimic the activity in the brain, **groups of "neurons"** (fluorescent beads) should be **excited nearly simultaneously**. Simultaneous excitation in the group might be difficult for NMF, but you may give a variance in timing of approximately 5-20 ms within a group (to profit from this variance, the camera frame rate should be close to 100 fps). Groups of 1-5, 10, 15, 20, 30 sources should be excited in separate experiments. For example, for the iteration of 5 nearly simultaneously excited beads, I would use some time intervals for one 5-beads composition and another time interval for another 5-beads composition. Run NMF and extract number of unique sources. Do next iteration with higher number of simultaneously excited beads.

From this experiment authors will be able to estimate: **1. number of simultaneously detected sources (at some moment in time) vs the true number of excited sources and 2. overall number of NMF detected unique sources vs the true number of overall excited sources. It would be important to know how well the approach can demix spatial modes when they overlap.** Importantly, it is ok if it does not, but authors should identify and discuss this limitation in the text as it is crucial to in-vivo application. Also, if demixing large groups of neurons is not possible, the method should not be called high-throughput (standards for high-throughput are high).

Q4.2 Same experiment can be done for samples with scattering medium and without to compare NMF performance.

As we detailed in the initial message and in the previous response (R1Q1), we designed and performed new experiments aiming at mimicking more realistically the GECI-based neuronal signal (Figures 02.2 and 02.3). We demonstrated that NMF can demix simultaneous spatiotemporal signals coming from tens of sources with coincident overlapping transient peaks and with a noisier dynamic background. Importantly, we carried out an experiment on a sample in which many sources were touching each other in the distal FoV, which resulted in many similar overlapping patterns in the proximal FoV of the fiber. We found that individual sources with overlapping spatial patterns were well demixed. These new experiments now give a lower estimate of the maximum number of sources that NMF could potentially demix depending on the area of the fiber core. Moreover, we showed that the signal coming from sources located below a scattering layer can also be demixed by NMF, and the corresponding fingerprints are modified compared to the fingerprints obtained without a scattering layer (they are more asymmetric and they have stronger spatial overlap). Again, in this experiment, we find that individual sources with overlapping spatial patterns were well demixed. In addition, this latter result suggests that the method is sensitive to neurons present at different depths, with a scattering layer (brain tissue) in between them. Thus, the spatial details of the fluorescence fingerprint patterns depend on the source's 3D localization (x, y, and z coordinates), consequently a larger number of neurons than the 2D maximum estimate (number of neuronal somas that fit the MMF core area) could be potentially demixed by NMF. Importantly, Parafilm M® is a strong scattering media and it can be considered a good model to mimic biological tissue. Parafilm M® scattering properties have been already studied by our lab (see supp info of Boniface A. et al. Optica

(2019) <https://doi.org/10.1364/OPTICA.6.001381>). In particular, one layer of Parafilm M® is approximately 120 µm thick, having a scattering mean free path of around $l_s \approx 170 \mu\text{m}$ and its transport mean free path close to $l^* \approx 0.7 \mu\text{m}$, which leads to an anisotropy factor of $g \approx 0.8$ for green light (532 nm). These values, published in Optica (2019), and well-detailed in Antoine Boniface's PhD dissertation (2020), are reasonably close to the values obtained in biological tissues, such as the brain tissue. Thus, one layer of Parafilm M® is approximately as scattering as 120 µm of brain tissue - and definitely more scattering than a 50 µm brain slice. A comprehensive explanation can be found at Antoine Boniface's PhD dissertation (Boniface, A. Light control in scattering media and computational fluorescence imaging: towards microscopy deep inside biological tissues. (Laboratoire Kastler Brossel, L'Université Pierre et Marie Curie, 2020)).

For these two experiments (Figures 02.2 and 02.3), it is worth emphasizing that we added a neuropil background with a clear spatiotemporal overlap with the other time traces. The neuropil signal has an amplitude of the same order of magnitude as the fluorescence time trace activity coming from the individual sources. In addition, previous work from the team already addressed the case of NMF performance depending on a fluorescence background [Moretti C. and Gigan S. Nature Photonics (2020); Soldevilla F. et al. Optics Express (2023)]. The NMF performance was evaluated by quantifying the cross-correlation statistics of the retrieved time traces over different experimental background conditions. Typically, NMF performance starts to slowly get impaired when the max value of the background signal is 1.5x bigger than the max value of the sources' activity time trace signal. In this case, the median temporal cross-correlation gets lower than 80%, with the first quartile of ~65%. Nevertheless, even when the max background over the max activity ratio is of ~2, the median cross-correlation in the time trace is still above 70%, with the first quartile above 60% [see supp info of Moretti C. and Gigan S. Nature Photonics (2020)]. In our new experiments we present here, we now included a fast oscillating neuropil source with a dynamic non-sparse signal which has a strong overlap with the other sources and has the same order of magnitude as the transient peaks of the sources (see the ensemble time trace of Figure 02.2 and Figure 02.3). Typically, the neuropil amplitude signal is not 2x higher than the transient peak maximum, hence we believe NMF could easily decompose its signal as an extra component source. For cases in which the background buries the signal, it has already been shown in the literature that temporally constrained NMF (CNMF) can retrieve well the time traces from neurons in such condition measured by wide-field 1-photon microscopy [Pnevmatikakis E.A. et al. Neuron (2016). <https://doi.org/10.1016/j.neuron.2015.11.037>; Zhou P. et al. eLife (2018) <https://doi.org/10.7554/eLife.28728>]. In addition, in Moretti C. and Gigan S. Nature Photonics (2020), we showed that the NMF performance improves for longer video recordings, that uses more frames for NMF optimization. Endoscopy is typically well adapted for long recordings, therefore it provides another possible improvement strategy. As a final remark, our work demonstrates for the first time how one could decompose ensemble time traces transmitted by multimode fiber into their individual components by using the simplest and most general formulation of NMF. We wanted to establish as clearly as possible the proof-of-principle for the technique, focusing on a solid ground truth, keeping the algorithms as "vanilla" as possible, so as to underlay the physical concepts. It is known from the literature that the performance of NMF can be significantly improved by adding prior signal constraints, which could also be implemented in our method. For example, it has already been shown in the literature that constrained NMF (CNMF) can significantly improve the detection of GECl time traces of partially overlapping neurons in noisy 1-photon microscopy data compared to PCA-ICA and unconstrained NMF [Pnevmatikakis E.A. et al. Neuron (2016); Zhou P. et al. eLife (2018)].

R1.Q5 - Fig. 2-3 no reported excitation intensity per source.

As we mentioned in the answer of the question R1.Q3, we included in the supp info S01 the intensity we typically used during the experiment, which was $3,0 \cdot 10^{-8} \text{ W}$ per $10 \mu\text{m}$ diameter bead area, resulting in a local intensity of $3,8 \times 10^{-2} \text{ W/cm}^2$ per bead or fixed neuron. In addition, we added a brief explanation of how we generate the synthetic GT time traces.

The new text in the caption of supp info Figure S01 reads: "We typically used a laser power of $3,0 \cdot 10^{-8} \text{ W}$ per $10 \mu\text{m}$ diameter area (bead size), resulting in a local intensity of $3,8 \cdot 10^{-2} \text{ W/cm}^2$. Synthetic fluorescence traces were generated like previous work from our team, using the spike activity dataset from real available datasets acquired in mouse visual cortex, and converting them into calcium (excitation) traces (MLSpikes from Vanzetta's group: Deneux, T., et al. Accurate spike estimation from noisy calcium signals for ultrafast three-dimensional imaging

of large neuronal populations *in vivo*. *Nat Commun* 7, 12190 (2016). <https://doi.org/10.1038/ncomms12190>) using a GCaMP6s physiological model and resampled at 10 Hz. The analog calcium transient profiles were designed by changing the dwell time of the excitation in relation to a longer, but constant, detection window time - similarly to previous works from our team. The DMD can control the excitation beam fast (>10kHz) concerning the detection window, allowing it to mimic an analog time trace profile very smoothly in the video recording. A higher (resp. lower) fluorescence intensity from a given bead corresponds to a longer (shorter) excitation dwell time during a given detection window. The detection window in the experiments were: 500 ms (Figure 02.1 - the 6 bead experiment, camera dynamic range of ~10.000), 1000 ms (Figure 02.2 - the 26 bead embedded in 50 μm thick agarose layer sandwiched between coverslips, camera dynamic range of ~1.500), 2000 ms (Figure 02.3 - the 26 bead embedded agarose layer sandwiched between coverslips + single Parafilm M® layer on top, camera dynamic range of ~1.500), 4000 ms (Figure 03 - 4 neurons in 50 μm brain tissue slice of the Gad-EGFP labeled neurons from the cortex, camera dynamic range of ~450). Therefore, most of the experiments used a moderate dynamic range compared to the available 16-bit."

R1.Q6 - I find correlations on figure 2-3 g-h to be not informative. It is of course nice that the pattern is similar. I would find more useful to report prediction accuracy of NNMF relative to GT. For example, training the model on GT (if the source activation is binary) and predicting with the model from the unseen NNMF data.

It is not clear for us what exactly the reviewer means by accuracy in the model. Non-negative Matrix Factorization is an unsupervised optimization method for signal processing problems and not a neural network where we can extract the confusion matrix (accuracy, recall, etc) and/or model/predict any results. The data is not used to train the algorithm, but is processed by it. However, we did quantify and include in the captions of the result figures the values of similarity between the recovered demixed traces and the ground truth traces, and the residual of the NMF and cross-talk between the recovered time traces using correlations, which is to the best of our knowledge the standard benchmark for this kind of techniques. For the latter, the off-diagonal values of NMF-GT temporal correlations were subtracted by the off-diagonal values GT-GT correlation that contributed for each retrieved source. Residual NMF cross-talk could be estimated by evaluating these off-diagonal elements of the correlation table, which was of order of 5% while the overall NMF-GT agreement in temporal correlations could be evaluated by the diagonal elements, and they were overall above 80%, with average value around $\langle \delta_{g,n} \rangle = \delta_{\text{avg}} = 86.0\%$, with the exception to be in the experiment using Parafilm M® (Figure 02.3), with $\langle \delta_{g,n} \rangle = \delta_{\text{avg}} = 77.0\%$. In addition, It is important to take into account that these time traces are relatively short (between 2000 and 3000 frames), and longer video recordings should improve the quality of NMF optimization (see discussion on Moretti C. and Gigan S. Nat Photonics (2020)). Finally, note that the fluorescence modulation obtained in the recorded video is not a binary signal, but an analog signal as GECl profiles thanks to the way we control the excitation dwell time within the detection time window for each frame to be able to generate a video of exponentially decay fluorescence with the same characteristic time as GCaMP6, whose temporal activity were taken from studies in the virtual cortex. We included a brief explanation about how we generate the synthetic GT time traces in the supp info S01 and we added in the supplementary information the tiff files of the videos that we used as input data in the unconstrained NMF.

R1.Q7 - Line 188-194, discuss about saturation intensity is interesting. I wonder how much of the limitation that would be when you have multiple sources distributed across the fiber field-of-view?

In the miniscope sensitivity experiments, we chose a given fixed video recording settings so that it was possible to measure both bright and dim fluorescence patterns emitted by the same source but located at different radial positions in the distal FoV. Fluorescence patterns generated from sources close to the fiber core edge are dimmer because less light is collected from the fiber in comparison to sources located at the center of the fiber's distal FoV. We demonstrated that the low-cost Miniscope V4.4 has enough sensitivity to probe a single source pattern (even the dim ones), which it was previously observed in our proof-of-principle tabletop experiment (see supp info S01 for setup details). However, the current 8-bit depth CMOS camera of the miniscope might experimentally limit the potential of the method in probing a large number of sources because the miniscope camera can get more easily saturated due to the current short dynamic range (shallow bit depth). Saturated camera pixels should be avoided during the video recording because they do not allow one to distinguish GECl dynamics within the saturated frames. Probably, when N number of pixels are saturated over a finite number of F frames, it might create a cross-talk in all retrieved time traces over the F frames whose fingerprint signals depend on those N saturated pixels. In our tabletop experiment, the sCMOS camera we used had a larger bit depth (16-bit dynamic range) and we demonstrated that a simple unconstrained NMF was able to demix more than 20 sources with a significant spatiotemporal overlap with other sources and non-sparse neuropil activity. In

our experiments the camera dynamic range (DR) in the ensemble time trace were around: DR = 10.000 for Figure 02.1 (Proof-of-principle with 6 beads), DR = 1.500 for Figure 02.2 and Figure 02.3 (Proof-of-principle with 26 beads with and without Parafilm M®), DR = 450 for Figure 03 (Proof-of-principle with 50 µm thick brain slice). Therefore, most of the experiments used a moderate dynamic range compared to the available 16-bit. All the recorded video used as an input on NMF algorithm are available in the supplementary information.

We added the following text in the discussion of the paper :

“The current 8-bit depth CMOS camera of the miniscope might experimentally limit the potential of the method in probing a large number of sources because the miniscope camera can get more easily saturated due to the current short dynamic range (shallow bit depth, see Figure 04d and Figure S03). Saturated camera pixels should be avoided during the video recording because they do not allow one to distinguish GECl dynamics within the saturated frames. Probably, when N number of pixels are saturated over a finite number of F frames, it might create a cross-talk in all retrieved time traces over the F frames whose fingerprint signals depend on those N saturated pixels. In our tabletop experiment, the sCMOS camera we used had a larger bit depth (16-bit dynamic range) and we demonstrated that a simple unconstrained NMF was able to demix more than 20 sources with a significant spatiotemporal overlap with other sources and non-sparse neuropil activity. In our experiments the camera dynamic range (DR) in the ensemble time trace were around: DR = 10.000 for Figure 02.1 (Proof-of-principle with 6 beads), DR = 1.500 for Figure 02.2 and Figure 02.3 (Proof-of-principle with 26 beads with and without Parafilm M®), DR = 450 for Figure 03 (Proof-of-principle with 50 µm thick brain slice). Therefore, most of the experiments used a moderate dynamic range compared to the available 16-bit.”

R1.Q8 - Line 214-223, for the discussion a recent paper from Pisanello lab might be useful where they also attempt to gain spatial resolution (Bianco et al. “Orthogonalization of Far-Field Detection in Tapered Optical Fibers for Depth-Selective Fiber Photometry in Brain Tissue.” *APL Photonics* 7, 2 (2022): 026106. <https://doi.org/10.1063/5.0073594>.) Their prediction accuracy along the taper length was still relatively low, probably due to the fact that imaging across many fiber modes (degrees of freedom) can deliver a lot of information?

We thank the reviewer for mentioning the very interesting approach of Bianco M. et al. *APL Photonics* (2022) <https://doi.org/10.1063/5.0073594>. Indeed, their work shares some similar features with our method, such as the use of multimode fiber to collect fluorescence readouts and the overall goal of decomposing the signal of fiber photometry methods. In their work, they managed to distinguish ensemble integrated fiber photometry signals from groups of neurons located at different penetration depths by using a multimode fiber with a tapered end. Different ensemble signals were detected at different positions of the tapered edge (distal end) and these could be ensemble mapped, after choosing an intensity threshold in the signal, to different average rings in the proximal end with a camera. However, these rings do not correspond to the signal of individual sources, but to the signal of a population of neurons in a given tissue depth. In our case, we are presenting the first proof-of-principle of how one could obtain single-source resolution in fiber photometry methods. In their work, they characterized the cross-talk from different tapered end segmented regions and their method accuracy is highly dependent on the modal dispersion of the fiber, which might explain their results. In our case, modal dispersion does not limit our method, but could even help NMF in demixing patterns by making the probed spatial fingerprint less symmetric, thus easier to be demixed. Extreme cases of modal dispersion in fibers generate speckle patterns, and previous works from our group already demonstrated that NMF works well in demixing them (Moretti C. and Gigan S., *Nature Photonics*, 2020). Another difference is that they use a patch fiber to image in the far field the ensemble rings, and a rigorous characterization of how much fiber bending affects their results should be ideally implemented for freely-behaving mouse experiments. It is worth mentioning that their approach is very promising and could be possibly combined with our method aiming at improving fiber photometry results at different depths with the tapered end. Nevertheless, combining our method with a tapered end fiber would be one among several other possible experimental modifications with potential to improve the results at the cost of making the accessibility and reproducibility of our method slightly more complex. We added the following discussion in the manuscript briefly mentioning the similarities and differences of both methods: “previous works have demonstrated methods to obtain some degree of readout specificity in photometry experiments. For example, in Bianco M. et al. *APL Photonics* (2022) <https://doi.org/10.1063/5.0073594>, the authors have tapered the end of a multimode fiber so that fluorescence light coming from different depths of the taper is spatially separated in the far field of the optical fiber, at a camera. Although this technique is a nice improvement of fiber photometry, it does not give single source specificity, but rather it allows to obtain signals from

spatial ensembles located at different depths in the tissue. Our approach is very different: we are not relying on spatial information only to separate the sources, but we take advantage of the temporal fluctuations of each source to extract both the spatial fingerprint and the temporal trace corresponding to each source, using a NMF algorithm. As a consequence, we manage to reach single source specificity. Moreover, in our approach we show that the raw video data doesn't need to be transformed or pre-process to obtain more specificity in the photometric readouts, although we could promptly do it as an extra strategy to improve its performance. It is worth noting that the actual implementation is performed with a cylindrical MMF, which gives ring patterns resembling that of Bianco et al, but our method would also work with other types of fibers (for example square core fibers) as long as patterns corresponding from different sources are different. Lastly, multimode fiber modal dispersion doesn't necessarily limit our method, but it can be engineered to tune our method performance."

R1.Q9 - Line 219-223, the ability to identify the displacement of the fiber relative to one bead is described as some sort of advantage, but in reality **cylindrical symmetry** of the fiber makes it difficult to identify multiple sources placed equidistantly from the fiber center (all would contribute to one ring). **How big limitation would that be?**

As we show on Figure 04, the patterns corresponding to sources placed equidistant from the fiber center still differ by the central part of the patterns, which contain two spots aligned with the direction pointing to the bead (same azimuthal angle). This difference should be sufficient to demix the sources, as is illustrated from the example of beads 14 and 16 in figure 02.2. As we mentioned in response to question R1Q1, these beads were located at the same distance from the fiber but at slightly different angles and signals from these beads were well-demixed.

In addition, we also now show in Figure 02.3 that the presence of a scattering layer in between the beads and the fiber, mimicking what should happen when applying our method to brain tissue, breaks the cylindrical symmetry of the patterns and will make them easier to demix. Finally, as we mentioned earlier, we could easily adopt additional strategies to further break the cylindrical symmetry of the patterns, for example using fibers with a different core geometry (e.g. square-core fibers), or using a non-polished fiber tip on the distal side (pointing to the brain).

We added the following texts in the manuscript (results of Figure 02.3) briefly mentioning what we would expect from the signal coming from sources buried behind a scattering layer:

"With this experiment, we expect to show (1) that our method could demix fluorescence spatiotemporal signals transmitted by short MMF coming from sources concealed beneath a scattering layer and (2) that the scattering layer scrambles more the fluorescence wavefront and, consequently, breaks the residual symmetry of the fingerprint patterns we currently obtained, thus affording depth sensitivity to the method. In an extreme case, where a strong scattering medium is present in between the sources and the fiber, multimode fibers are expected to transmit a fully developed speckle as fluorescence patterns, which our team has already demonstrated that NMF can successfully be employed [Moretti C. and Gigan S., Nature Photonics, 2020]."

And

"As expected, the scattering patterns from this experiment [Figure 02.3] are different from the previous cases [Figure 02.1 and Figure 02.2]. They became less symmetric, confirming that the scattering layer is breaking the cylindrical symmetry of the system (which should facilitate NMF demixing), but spatially noisier (i.e., with decreased intensity contrast)."

R1.Q10 - Finally, in multiple parts of the manuscript authors stress that they "expand the capacity of fiber photometry in resolving neuronal activity" (line 250-252 etc.) however at the moment I see little experiments supporting this statement. Majority of my suggestions for proof of concept experiment target those improvements and will guide the experimental choices for in vivo application. Ideally, I would like to suggest using miniscope-coupled to fiber approach in-vivo, where the fiber is inserted or implanted in the mouse brain above neurons expressing GCaMP calcium indicator. Authors, could record from such mouse, NNMF estimate sources, perfuse the mouse and count GCaMP-labeled cells and compare with NNMF result. Note, for such experiment better use somata-localized GCaMP, as labeled neuronal

dendritic trees may essentially spread across all field-of-view. I also understand that *in-vivo* experiment is possible to do only in the lab dedicated to such work (this includes already obtained animal licenses and many other skills) and this lab may not be able to do that alone but only in collaboration. Therefore, I do not suggest this as a mandatory experiment, but to enhance trust and generate more interest in this approach.

As we mentioned in the introduction of these responses, we agree with the reviewer that *in vivo* data would be ideal to give a final proof of concept of our method. However, as the reviewer adverted, performing *in vivo* experiments would require resources and facilities well beyond our current capacities. Nevertheless, we think that our method is very novel from the conceptual and optical design points of view, and we believe that a thorough proof-of-principle demonstration *in vitro* is already a very important and necessary step to prove our concept. In response to the reviewers questions, we added new experiments showing that the technique performs well with a densely labeled sample, with significant overlap of the spatiotemporal recorded signals, and with a scattering media between the fluorescent sources and the fiber. With these new experiments, we hope that we can convince the reviewer that the technique has a strong potential for *in vivo* applications, which will be the subject of a second large-scale project involving a new collaboration.

Minor concerns:

1. It would be nice to cite earlier works (one of the papers from Papadopoulos et al. and from Turtaev et al.) which achieved spatial resolution by digitally scanning the excitation light at the fiber output.

Papadopoulos, Ioannis N., Salma Farahi, Christophe Moser, and Demetri Psaltis. "High-Resolution, Lensless Endoscope Based on Digital Scanning through a Multimodeoptical Fiber." *Biomedical Optics Express* 4, 2 (2013): 260–70. <https://doi.org/10.1364/BOE.4.000260>.

Papadopoulos, Ioannis N., Salma Farahi, Christophe Moser, and Demetri Psaltis. "Focusing and Scanning Light through a Multimode Optical Fiber Using Digital Phase Conjugation." *Optics Express* 20, 10 (2012): 10583–90. <https://doi.org/10.1364/OE.20.010583>.

Turtaev, Sergey, Ivo T. Leite, Tristan Altwegg-Boussac, Janelle M. P. Pakan, Nathalie L. Rochefort, and Tomáš Čížmár. "High-Fidelity Multimode Fibre-Based Endoscopy for Deep Brain *In Vivo* Imaging." *Light: Science & Applications* 7, 1 (2018): 92. <https://doi.org/10.1038/s41377-018-0094-x>.

We thank the reviewer for listing some relevant related works for us to cite. Those references were added to the main text.

Reviewer #2 (Remarks to the Author):

R2.Q1 - The manuscript by Rimoli et al presents an experiment in which a short multimode optical fiber is used to collect and demix fluorescence from different fluorescent beads in a sample at the fiber output. The idea is that, in experiments in which fluorescent indicators, such as calcium or voltage indicators, are expressed in a neuronal population, a very simple optical setup, essentially composed by a short and minimally invasive optical fiber and a CMOS camera, can demix the signals from different neurons, as they come temporally sparse and have a defined spatial signature on the camera detector. The promise of the manuscript is that, with this simple idea, the broad neuroscience community that relies on fiber photometry to measure population activity in ensemble of neurons could now have the resolution to disentangle activity from individual neurons, and do that in a mouse that is freely moving and thus free to perform unrestrained behavioural task.

The manuscript applies several concepts already demonstrated in the group of S. Gigan (see for example Ref. 43, 44, 48 of the manuscript), which already allowed them to demix fluorescent signals from a scattering tissue (which results in fully developed speckle patterns), and to reconstruct the spatial locations of hidden objects by using correlation strategies on the different speckle patterns. Mathematically, the demixing is carried out by using a non-negative matrix factorization algorithm. In this manuscript, the authors apply the same strategy and the same algorithm to demix traces that come from non-developed speckle patterns, which are characteristic signatures of short multimode fibers. As the fiber is short, the spatial signature is supposedly not sensitive to bending, vibrations or conformational changes of the fiber, which is a strong advantage for freely moving studies.

Although I agree with the authors that changing a fiber photometry setup in something that can give single neuron resolution (essentially for free) by only adding a camera detector can be a game changer in neuroscience, my honest opinion is that the current manuscript is still far from demonstrating that. Indeed, **no real biological experiment is performed**, nor the reported results can be extrapolated to infer what would happen in a real in vivo experiment in the mouse brain. In my opinion, because the biology is not there, the current manuscript does not add too much to what the authors have previously demonstrated in scattering tissue (Ref 43, 44, 48), in the sense that, if demixing can be carried out through the mouse skull, then I would expect the same process to work through a multimode fiber too.

At the same time, since biological experiments are not carried out, I would have expected a more thorough explanation (with e.g. simulations) of the observed spatial signatures in terms of modes propagating through the fiber. As far as I understood, the relationship between the position of the beads with respect to the fiber centre and the detected fluorescence patterns, should be related to the spatial modes excited by the beads. However, this is not at all explained in the manuscript. A better description of the process would make the story stronger at least from the physics point of view.

Suggestions for possible changes.

From my point of view, the main limitation of the work is the absence of relevant experiments from which stronger conclusions on the performances of the system **in real biological conditions** can be inferred. The demonstration presented in Figure 03 is not sufficient, as only **4 neurons** are selectively illuminated, whereas in a real case scenario, widefield illumination would excite several more neurons distributed in more planes. A more realistic scenario would at least involve a widefield illumination of the whole field of view (which can come from the fiber itself), superimposed to the variable illumination sent by the DMD to selected neurons. To **simulate the neuropil fluorescence variation**, the widefield illumination should also vary in intensity over time. Ideally the authors should take a **raw video recorded** with a 1P miniscope with widefield illumination and try to reproduce with the DMD the variation of neuropil fluorescence and that of the neurons. This would give a better estimate of the performances of the device, and it would be best carried out in the miniDART configuration, as this is one of the main novelties of the manuscript.

I am aware that going for a full in vivo experiment in mice would probably take a very long time (even if one of the corresponding authors have already expertise in similar experiments). However, only **a real biological experiment**

will ultimately allow one to judge on the use of the presented method. The experiment could perhaps be carried out in brain slices expressing GCaMP, or in smaller animals, such as drosophila or zebrafish. If biological experiments are out of question for this manuscript, the authors should at least provide the more relevant experiment I outlined before in the most convincing way possible. Even in that case however, I am not sure that with no biology at all, this work should be published in Nature Communications. A specialised optical journal would, in my opinion, be more appropriate.

We thank the reviewer for his general assessment of our work, with whom we generally concur. Regarding a biological demonstration, unfortunately, none of the paper's authors has the skills to carry out *in vivo* experiments. Therefore, performing these experiments should involve setting up a collaboration or recruiting someone qualified, which we are planning to do in the medium term. In the meantime, to answer the reviewers' questions as best as we could, we have chosen instead to flesh out the *in vitro* proofs of concept, mimicking more closely what will happen in a brain. We prepared a sample with a higher source density, we experimentally modeled the presence of a neuropil, and we introduced a scattering medium between the fluorescent sources and the optical fiber. With these experiments, we were able to show that we could demix more than 20 sources (out of 26) with high fidelity, with or without scattering. We could also show how the fingerprint patterns would transform in the presence of scattering, becoming more asymmetric. Sources with spatial fingerprints highly overlapping that of the neuropil source were well-demixed, showing some robustness of our method to neuropil signal. Nevertheless, when applying this method to *in vivo* conditions, we believe that the most promising experimental design is to label neurons using soma-targeted GCaMP8 (Grodem S. et al. Nature Communications (2023) <https://doi.org/10.1038/s41467-023-36324-3>) already available as AAV vectors from addgene), thus avoiding neuropil signal. The proof-of-concept that we designed *in vitro* is our best effort to mimic this situation, and we hope that we can convince the reviewer that this proof of concept is relevant. Finally, we kindly point out the novelty of our work in the optics domain, and the fact that Nature Communications is a generalist journal, not a biology-focused journal, and does indeed publish quite a lot of technical papers in fundamental optics, imaging and microscopy, without biological demonstration. Although we don't provide a real biological experiment, we are thus convinced our results are within the scope of Nature Communications.

R2.Q2 - The next main point that I think is missing in the current version of the paper is a more thorough explanation of the spatial signatures measured at the fiber output. As far as I understood the fluorescence signatures are a result of the excitation of different modes through the fiber. **Spatially resolved fiber photometry was already demonstrated through tapered fibers by the groups of M. de Vittorio and F. Pisanello, albeit in that case the variations corresponded to variations in the z axis, rather than in the xy plane.** If one takes for example this reference (Bianco et al. Orthogonalization of far-field detection in tapered optical fibers for depth-selective fiber photometry in brain tissue, APL Photonics 2022), the far field fluorescence patterns recorded on a camera detector display a characteristic ring like shape as a function of the z position (which also correspond to a different xy distance from the fiber center), which reminds me of the spatial signatures measured here by Rimoli et al. However, in **Bianco et al, the signatures were measured in the Fourier plane of the fiber**, whereas here the authors directly look at the fiber end surface. How are the two experiments linked to one another? My impression is that the overall effect should be pretty similar. I think that the authors should at least discuss, if any, the differences with respect to what already demonstrated through a tapered fiber, from a theoretical point of view, and in terms of advantages of their methods, so that the readers can better appreciate the novelty of their configuration. If the mechanism behind the two methods is really the same, I would expect a detailed description of why the current method is better than the previous one and which problems can it solve that could not be solved with a tapered fiber.

The reviewer is right, the fingerprint patterns emerge as the projection of the fluorescent light emitted by each source on the different spatial modes of the fiber. Nevertheless, we want to point out that our method is general and does not depend on the geometric properties of the patterns, as long as the patterns from the different sources are different from one another. The method should also work with other types of fibers (eg. square core fibers) or customized fibers (eg. fiber with a non-polished surface on the sample side to produce fully developed speckles). Indeed, we showed in our previous work that fully developed speckle patterns were also suitable (Moretti C. and Gigan S., Nature Photonics 2020). Therefore, in the paper, we decided not to focus on the exact nature of the patterns but rather on the robustness of the method to different properties of the sample, including the source density and the presence of a scattering layer.

Nevertheless, as the reviewer suggests, it is very interesting to compare our method with the work from Bianco et al, APL Photonics 2022. Although the final goal of the two methods is the same (getting some level of neuronal specificity from photometry experiments), the methods are in essence very different:

- As the reviewer understood well, In Bianco et al, the authors taper the fiber so that fluorescence light coming from different depths of the taper is spatially separated in the far field of the optical fiber. Therefore they don't use any temporal information to obtain spatial specificity: they rely only on spatial information.
- In our case, the patterns from the different sources that we want to unmix are spatially overlapped, and we take advantage of the temporal fluctuations of each source to extract both the spatial fingerprint and the temporal trace corresponding to each source, using a NMF algorithm. It is true that the rings of the patterns that we obtain with the short cylindrical fiber resemble the rings that Bianco et al obtain in the far-field, but 1- the mechanism for obtaining the ring is different (even though in both cases is relies on the projection of fluorescent light on the spatial modes of the fiber, the geometry used is different); 2- we don't use the fact that our patterns resemble rings to demix the signal. In fact, as mentioned just above, our method would work with any kind of pattern as long as patterns from different sources are different; 3- we showed that we don't need to transform or pre-process our raw video data to obtain single-source time trace specificity, although we could promptly do it as an extra strategy to improve the performance of the method; 4- multimode fiber modal dispersion doesn't necessarily limit our method, but it can be engineered to tune the method performance.

We now include in the manuscript a comparison with Bianco et al. : “previous works have demonstrated methods to obtain some degree of readout specificity in photometry experiments. For example, in Bianco M. et al. APL Photonics (2022) <https://doi.org/10.1063/1.5007150>, the authors have tapered the end of a multimode fiber so that fluorescence light coming from different depths of the taper is spatially separated in the far field of the optical fiber, at a camera. Although this technique is a nice improvement of fiber photometry, it does not give single source specificity, but rather it allows to obtain signals from spatial ensembles located at different depths in the tissue. Our approach is very different: we are not relying on spatial information only to separate the sources, but we take advantage of the temporal fluctuations of each source to extract both the spatial fingerprint and the temporal trace corresponding to each source, using a NMF algorithm. As a consequence, we manage to reach single source specificity. Moreover, in our approach we show that the raw video data doesn't need to be transformed or pre-process to obtain more specificity in the photometric readouts, although we could promptly do it as an extra strategy to improve its performance. It is worth noting that the actual implementation is performed with a cylindrical MMF, which gives ring patterns resembling that of Bianco et al, but our method would also work with other types of fibers (for example square core fibers) as long as patterns corresponding from different sources are different. Lastly, multimode fiber modal dispersion doesn't necessarily limit our method, but it can be engineered to tune our method performance.”

R2.Q3 - Other points: - Related to a better explanation of the spatial signatures, **theoretically**, what is the maximum number of spatial signatures that the authors should be able to unmix. Or, in other words, with reference to the Supplementary figures 3-4, **which is the minimal radial and angular distances that would generate sufficiently different spatial signatures?**

We chose to answer this question experimentally instead of theoretically, because we thought that it would be more convincing. As we mentioned already, we performed *in vitro* experiments with a higher density of beads, with beads touching one another without (Figure 02.2) and with (Figure 02.3) a layer of scattering media in between the fluorescent sources and the optical fiber. In these experiments, we showed that we were able to unmix many time traces from beads that were touching each other. For example, in Figure 02.2 we demixed signals from beads 14 and 16 that are located the same radial distance from the fiber center but at slightly different angles. We also measured patterns from sources separated by steps of 10 μm along a radius of the fiber and showed that they are already very different (Figure 04). Therefore, we estimate that we would be able to unmix tens of spatial signatures (or even hundreds) depending on the signal to noise ratio of the recordings, the sparseness of the activity, and the length of the recordings. Here, with very short recordings, we showed that we could demix more than 20 sources, with or without a scattering layer between the sources and the fiber.

As we answered to reviewer 1, interestingly, we confirmed that the presence of a scattering layer changes the spatial features of the fingerprint patterns, making them less similar to each other (less symmetric). This feature could facilitate our method in demixing time traces and suggests that it is possible to tune up the total number of the probed sources if one decides to change the optical propagation properties of the system by engineering the implantable fiber. For example, one could add a scattering layer on the fiber distal end (the tip that faces the brain) by controlling its degree of polishing, or by simply choosing a fiber with different intrinsic propagation properties, such as NA, core geometry, etc.

R2.Q4 - There is no mention in the manuscript of the **z axis** at the sample plane. Is the method designed to work at one particular **z plane**? What would happen to the spatial signatures if the authors would displace the sample in **z**? I think that a proper calibration of the instrument (e.g. repeating the measurements in Supp. Fig. 3-4 at different **z**) as a function of the focal plane would be a very good idea.

As we mentioned in the caption of the figures, the fiber was always approximately 50 μm away from the coverslip of the sample to avoid mechanical damage on the fiber. Multimode fibers deliver and collect light as a wide field method, with a characteristic light cone given by their numerical aperture. In fact, we expect the performances of the technique to be similar for a large **z-range** in the free space, with only a decrease in collection efficiency when the beads get farther than the diameter of the core. Indeed, in a transparent media, we found that the spatial fingerprints were only slowly depending on **z**, mostly decreasing in signal contrast but not shape over large ($> 500 \mu\text{m}$) distances away from the fiber (data not shown). In the presence of a scattering medium, however, it is expected that sources placed at different **z** will have slightly different fingerprints. To show this, we measured fingerprints with a scattering layer (Figure 02.3) and without (Figure 02.2). As we expected, patterns obtained with the scattering layer lost symmetry. This effect could be used to distinguish sources that are at the same **x,y** position but at different **z**, as the fluorescence signal coming from the deeper sources will cross a thicker scattering medium before reaching the fiber. Thus, although in principle there is no particular **z-plane** sectioning by the fiber, our new results suggest that sources at different depths in the brain would have different fingerprints and thus could be demixed.

R2.Q5 - Related to the previous point, an experiment in which different beads at different **z** are excited and their fluorescence unmixed would be interesting.

The experiment proposed by the referee is not straightforward to perform, because the DMD is only conjugated to one specific plane in the sample, and therefore we can only accurately control the intensity of beads in this plane. Nevertheless, because the spatial fingerprints only slowly depend on **z** in free space, we should obtain similar results to those shown in Figure 02.2 if beads were arranged in 3D rather than in 2D, such that the projection on a particular **z-plane** is similar to the 2D distribution of beads that we used in this experiment (shown in Figure 02-2d). Therefore, we anticipate that our method should work with sources placed in 3D, with a labeling density such that the number of sources in the vicinity of the fiber is similar to what we have shown here. Please note that we haven't tested yet how the method would perform for a higher density of sources. In theory, an infinitely long recording would demix all the probed sources with unique spatial and temporal signatures. In practice, the maximum number of sources one can demix will depend on the unique features of the activity for each source, the signal-to-noise ratio, and the length of the recordings. Here, we use very short recordings (3000 frames), and the activity rate is the one observed in the mouse cortex.

As a final note, if the sample would contain sources positioned exactly on top of one another, the patterns would be similar so the beads would be difficult to demix. However, as we mentioned just above (R2.Q4), in a scattering tissue such as the brain we expect fingerprints to slightly depend on **z**, which should help distinguish sources located on the same **x,y** position but on a different **z plane**.

We now changed the manuscript and include a comment about the **z-axis** : "Because of experimental constraints (the DMD being conjugated to only one plane at the sample), all the experiments were performed with sources located at the same axial plane. Nevertheless, the spatial fingerprints measured for single sources in free space only slowly depend on the distance along the propagation axis **z** over large ($> 500 \mu\text{m}$) distances away from the fiber (data not shown). Thus, we should obtain similar

outcomes to those shown here if the sources were arranged in 3D in a transparent medium rather than in 2D, such that the projection on a particular z-plane is similar to the 2D distribution of sources used in this experiment (shown for example in Figure 02-2d). Therefore, we anticipate that our method should work with sources placed in 3D (free space), with a labeling density such that the number of sources in the vicinity of the fiber is similar to what we have shown here."

In fact, unconstrained NMF is a general mathematical method that simply decomposes a matrix into a product of two matrices where all the entries are non-negative, with minimal residual. This has been applied in many different fields, such as astronomy, audio processing or computer vision. In our case, we transform our recordings into this matrix form where the columns (rows) represent spatial (temporal) information, but the method is blind to this physical interpretation and just operates numerically. Whether or not the fingerprints come from the same axial position, the procedure would demix them given that their spatial fingerprint and their temporal activity is different (as in the case of sources sharing the same axial position). One might check the previous work of our team (Moretti C. and Gigan S. *Nature Photonics* (2020) <https://doi.org/10.1038/s41566-020-0612-2>) where emulated the outcome of spatiotemporal signals coming from different z planes being simultaneously.

R2.Q6 - Still related to the z aspect, if one increases the distance between the fiber tip and the sample, would it be possible to extend the xy field of view beyond the diameter of the fiber (with a loss in the detected signal)? It is hard to me to understand that the field of view always stays constant and equal to the fiber diameter no matter the z plane.

The reviewer is correct, if one increases the distance z between the fiber tip and the sample, it would be possible to extend the xy field of view (FoV) beyond the diameter of the fiber, with a loss of detected signal. With the NA used here (0.39), this would result in an increase of the FoV diameter of ~60 μm for each 100 μm increase in z-distance, and even larger in the presence of scattering. However, this increase in FoV would of course correspond to a strong decrease in the excitation efficiency (proportional to the square of the FoV diameter). In addition the detection efficiency at the border of this increased FoV should be very small (as the isotropic emission from photons emitted by such a source will be larger than the effective fiber acceptance angle). Therefore, we expect to collect only little signal from sources located at the border of this increased FoV. In fact, we believe that the technique should be used as a priority to detect sources in the vicinity of the fiber tip (in the first ~100 μm). A more precise study of this effect is outside the scope of this paper, but could be performed in future studies.

R2.Q7 - The authors, contrary to their previous works, attribute all the scattering effect to the presence of the fiber and mode mixing. However, in a real experiment, in which fluorescence sources are buried at depth in scattering tissues, the fluorescence that hits the fiber tip would be also scrambled and not clearly assignable to a particular spatial location. In the extreme case of a very deep fluorescence source, the fiber tip would be hit by a speckle pattern, so that probably, many different modes of the fiber would be excited by individual neurons. Do the authors expect to be able to demix such more complex fluorescence patterns? Or in this case, two very deep neurons (that could still be excited in 1P widefield excitation) would generate very similar spatial signatures at the fiber output?

The authors thank the reviewer for making these insightful questions. These questions motivated us to probe how the fluorescence patterns would be if a scattering media is present in between the tip of the fiber and the sources. As already answered in the previous questions (R1.Q4 ; R2.Q4-Q6), we were expecting that the more modal dispersion or tissue scattering in the light propagation the more scrambled would be the fluorescence pattern transmitted by the fiber. Scrambled patterns actually help NMF optimization, because the spatial fingerprints are less symmetric, thus easier to demix. A fully developed speckle is the extreme case of wavefront scrambling and our group has already shown that NMF is capable of demixing those complex fluorescence patterns, including when originating from different planes (Moretti C. et al *Nature Photonics* (2020) <https://doi.org/10.1038/s41566-020-0612-2>, Soldevila F. et al *Optics Express* (2023) <https://doi.org/10.36468/OE.487764>). However, we didn't know how much closer to a fully developed speckle the patterns transmitted by short MMFs would be if we added a strong scattering medium (e.g., Parafilm M®) in front of the sources. As we mentioned in the answer of R1.Q4, Parafilm M® can be considered a good material to mimic biological tissue, and we roughly expect that a single layer of Parafilm M® to be as scattering as ~120 μm layer of brain tissue (i.e, mimicking well a brain tissue layer thicker than 50 μm). We showed in Figure 02.3 that a simple unconstrained NMF can demix close and equidistant sources which were buried behind a Parafilm M® layer. The

fingerprint patterns of these sources were less symmetric, but not a fully developed speckle. Interestingly, however, this result indicates that more scattering layers could be added (or engineered) in front of the fiber to finally obtain a fully developed speckle - meaning that the probing depth can be larger than what the Parafilm M® mimics (if sufficient signal is collected to detect pattern transients in the recorded video).

R2.Q8 - In Fig 2-3 the authors show signals that they send to the beads (or neurons) to simulate neuronal activity. However, the y scale in the traces is missing. **What is the typical and minimal variation of fluorescence they can detect (the so called $\Delta F/F$)?** It is not clear if the ground truth is a signal that is zero all the time (so the target is mostly not excited) and has a higher fluorescence value at certain (very short time), or if the authors allow for the resting fluorescence. In other words, neurons have typically a certain resting fluorescence, which is not zero, and sparsely increase their fluorescence value in correspondence to neuronal activity. The resting value would generate a continuous spatial signature background (similar to Fig.2a) on top of which different spatial signature would vary when more activity is present. Is this what the authors actually do in the experiment? If not, the authors should repeat the experiment by sending as ground truth raw data (not polished $\Delta F/F$ data) from 1P miniscopes.

In calcium imaging experiments, there are two types of background: a static background (corresponding to the neurons resting fluorescence F_0) and a dynamic background (typically the activity of the neuropil). In our experiments, we didn't model the resting fluorescence of the neurons ($F_0=0$). However, new GECI indicators such as GCaMP8 have low resting fluorescence and show large transient signals, with a $\Delta F/F_0$ corresponding to one action potential ranging between 40% and 100% depending on the variant used (Zhang, Y. *et al.* Nature (2023). <https://doi.org/10.1038/s41586-023-05026-4>). Therefore, this static background should remain moderate, and we expect NMF to be able to extract it (for example, using rank-1 matrix factorization, see *supp info* in Nöbauer, T. *et al.* Nature Methods, 2017). In fact, we believe that the dynamic background caused by the neuropil could bring more difficult challenges, and could prevent NMF to reliably extract spatiotemporal signals from individual sources if their fingerprints are overlapped with that of the neuropil. In our *in vitro* experiments, to evaluate if NMF could find and separate a non-sparse dynamic background component from the signal, we included an additional source to model a neuropil signal (Figure 02.2 and Figure 02.3). As we mentioned in R1.Q2, the neuropil source (bead #20 in Figure 02.2, and bead #13 in Figure 02.03) was chosen to have a spatial fingerprint overlapped with many other sources in the FoV. It exhibits a non-sparse, fast oscillating signal, with amplitudes on the same order of magnitude as the transient peaks of the other sources. In both experiments, we showed that a simple NMF algorithm could successfully retrieve more than 20 out of 26 spatiotemporal sources from the sample, including the neuropil source. As a final remark, when designing *in vivo* experiments, we would advise using GCaMP8 indicators targeted to the soma (Grodem S. *et al.* Nature Communications (2023) <https://doi.org/10.1038/s41467-023-36324-4>) already available as AAV vectors from addgene) to minimize the neuropil component and therefore the dynamic background and facilitate signal analysis.

In addition, previous works from our team have already evaluated the performance of NMF depending on the noise, background, and the length of the video recording (Moretti C. and Gigan S. Nature Photonics (2020) <https://doi.org/10.1038/s41586-020-0612-2>; Soldevila F. *et al.* Optics Express (2023) <https://doi.org/10.1364/OE.487768>). As previously mentioned in the question R1.Q4, NMF performance starts to slowly get impaired when the max value of the background signal is 1.5x bigger than the max value of the sources' activity time trace signal. In this case, the median temporal cross-correlation gets lower than 80%, with the first quartile of ~65%. Nevertheless, even when the max background over the max activity ratio is of ~2, the median cross-correlation in the time trace is still above 70%, with the first quartile above 60% (see Moretti C. and Gigan S. Nature Photonics (2020)).

Finally, the performance of NMF could be improved compared to what has been shown in our experiments. First, our team has already shown that the cross-correlation between NMF and GT traces increases with the number of frames in the recording (Moretti C. and Gigan S. Nature Photonics (2020)), thus longer recordings are preferable for noisier data. In addition, it has been shown in the literature that for cases in which the background is larger than the signal

from the somata, constrained NMF (CNMF) could retrieve well the time traces from neurons in wide-field 1-photon microscopy [Pnevmatikakis, E.A. *et al.* *Neuron* (2016) <https://doi.org/10.1016/j.neuron.2015.11.037> ; Zhou P. *et al.* *eLife* (2018) <https://doi.org/10.7554/eLife.28728>]. In our case, we could readily apply similar time constraints, which should allow significant improvement of demixing performances. In a second step, we could also design spatial constraints adapted to the structure of our patterns.

This answer has been copied in the discussion part of the manuscript.

R2.Q9 - In summary, I believe that the current manuscript represents a nice optical story, that adds up to the previous works of the group and previous works on fiber photometry. However, I believe that the biological conclusions the authors suggest, 'this work can open a whole new avenue for novel and affordable minimally invasive deep brain microendoscopic studies', are not demonstrated in the current manuscript. To me, it remains completely unknown what kind of patterns would one see in a real experiment in which many neurons expressing GCaMP (or voltage indicators) at multiple planes are excited simultaneously in a wide field manner through the fiber, and if the sensitivity of the demixing algorithm will allow different neurons to be resolved. In the absence of real biological demonstrations, in my opinion, the current work does not add sufficient novelty to what already shown by the group of Gigan (and possibly the groups of de Vittorio and Pisanello on tapered fiber), to be published in *Nature Communications*. I would instead recommend publishing it in an optical journal.

We thank the reviewer for his comments, and for challenging us to push our experiments to more difficult situations, with scattering, more beads, etc. We hope the new experiments and analysis we provided in this revised version convince him further of the future applicability *in vivo* (it certainly convinced us even more to pursue our work towards *in vivo* demonstration). One important technical limitation for *in vivo* demonstration with the miniscope is the limited dynamic range of the camera, but in recent discussion with experts (group of Liangyi CHEN and Changliang GUO in Beijing) it appears that different sensors could be used, with better sensitivity, speed and dynamic range.

We have also highlighted the significant conceptual and technical differences with the De Vittorio and Pisanello work. Regarding indicators, we have used realistic GCaMP signals, but voltage indicators are indeed very promising alternatives for our technique, thanks to their sparse and fast response, but will come with their own challenges (stronger background).

We hope that with this revised version, the reviewer will appreciate the significant novelty in the optical domain, and the prospect for *in vivo* applications, and be more positive in his recommendation for *Nature Communications*.

Reviewer #3 (Remarks to the Author):

R3.Q1 - The manuscript presents an innovative method for the separation of fluorescence signals through thin multimode fibers, and could in theory lead to a breakthrough in fiber photometry. In conventional fiber photometry, the signals transmitted through the multimode fibers become scrambled, thus restricting the ability to obtain temporal readings at the cellular level. The authors propose a solution to this problem by separating signals from several fluorescent sources by applying an unconstrained non-negative matrix factorization (NMF) algorithm directly to the raw video data. Because fibers are less bulky than GRIN lenses, this approach could ultimately be very useful for recording activity from deep brain areas.

I found this to be a very nice and creative idea which could make a good optics paper. However, there is a strong mismatch between the proof-of-principle demonstration and the claims of biological feasibility. This is potentially problematic in a multidisciplinary journal, because biologists would read this paper thinking that the magical solution is around the corner. This is possible, but the experiments shown here do not yet support this conclusion.

I believe there are two main paths to go forward:

- (1) **remove all claims of biological feasibility** as well as the suggestive but unphysiological results in sections 2.2. and 2.3. Discuss limitations in detail. Then publish a streamlined and still interesting (in my opinion) optics paper.
- (2) **do more realistic biological work** to support the claims of this being a useful tool for biology.

The work has potential and I look forward to seeing it after modifications, if the authors decide to take one of those paths.

The authors thank reviewer 3 for all positive comments and insightful observations. As we mentioned at the beginning of this response, the current version of the manuscript now includes results from experiments in which we attempted to mimic the brain tissue more realistically. We agree our claims of biological feasibility may have been a bit too enthusiastic, and we opted to lower down a bit our claims as you will see in the next answers. We complemented the previous results by performing more complex experiments aiming at demonstrating the general applicability of the method, emphasizing that its strength relies on its simplicity and easy accessibility. The new results demonstrate that several individual sources could be equidistantly probed around the fiber axis and still generate different fingerprint patterns that can be demixed by our approach. Even touching sources can get their individual time traces disentangled by our method. In addition, we challenged our method against a much more realistic ensemble time trace signal, with multiple overlapping temporal transient peaks, and by choosing one source to mimic fluorescent neuropil (i.e., a non-sparse temporal signal whose amplitude is of the same order of magnitude as the other sources) (Figure 02.2). We then repeated this experiment by introducing a scattering layer of Parafilm M® between the fiber and the sample (Figure 02.3), demonstrating that sources at different depths would have different fingerprint patterns and could be demixed. Importantly, our method is not limited by modal dispersion of the multimode fiber like other methods. Interestingly, however, modal dispersion and scattering can be conveniently engineered for our approach, changing the spatial profile of the generated patterns and, therefore, changing the sensitivity of the method. Due its simplicity and low-cost, several other optical engineering modifications or even more complex NMF analysis could be used and combined, thus conveniently explored for neuroscience applications. For example, one simple feasible modification would be to test if a fiber with a tapered end would improve the NMF signal decomposition from sources at different depths in relation to the tip of the fiber (similar to Pisanello's approach mentioned by the other reviewers). Therefore, we believe the current work might be of interest to a broad multidisciplinary scientific community, hence suitable for the Nature Communications journal.

R3.Q2 - Major: at several locations the paper either explicitly claims or strongly suggests that biological applications are almost within reach, but the actual experiments don't support this. For example, figure 4e shows a mouse with an implanted miniscope and MMF, but the data panels in this figure are about radial displacement of beads in agarose.

As discussed previously, we have alleviated all claims of direct applicability in biology. We also changed the Figure 4 of the manuscript to avoid misleading interpretations (see below). We reduced the size of the illustration of the MiniDART concept, we explicitly wrote on top of it that it is a "future prospect", and we detailed in the caption how the MiniDART would work *in vivo* for future experiments. We agree that fig 4e was misleading as it was previously.

Figure 04 – Novel microendoscopy concept using a short MMF and a miniscope: the MiniDART. (a) A typical fingerprint pattern from a single-fluorescent bead (10 μm diameter) probed by using only the miniscope excitation and miniscope detection through the multimode fiber. (b) The experimental setup to probe scattering patterns from the short MMF includes (i) the miniscope, (ii) a customized titanium base plate (YMETRY®) to hold the miniscope, (iii) a ferrule (Thorlabs SFLC230-10) that rigidly holds the multimode fiber (iv) within it, (v) a sample consisting of a single fluorescent 10 μm bead (spatial density < 1 bead/cm²), and (vi) a customized titanium tweezer (YMETRY®) to hold the ferrule. (c) Scattering fingerprints patterns at the proximal end (bottom row) depending on the radial position d of the single-bead at the distal end. Position d is indicated on the axis at the top of the figure, and is represented as a red arrow on the distal end pictures (with the bead indicated as a blue dot). Each pattern acquisition in the proximal end corresponds to 10 μm steps from the fiber central axis (black dot in the center at distal end images). The bigger the distance d , the wider the diameter of the bright spiral-ring pattern (of radius p in the proximal end, white arrow). The diagonal white dashed line is the azimuthal orientation of the vector d , which always coincides with the alignment angle of the 2 central bright points of the fingerprint patterns in the proximal end (see supp info Figure S03-S04 for details). The highest LED power values measured at the distal end of the fiber (whose core is 200 μm in diameter) were around 9.5 μW , which yields an excitation intensity of 0.3 mW/mm^2 at the output of the fiber core, and 2.4×10^{-5} mW excitation power per bead area. Exposure time: 100 ms (Miniscope FPS = 10 Hz). For more details, see Figures S03 and S04 in the supporting information. (d) The concept of doing experiments with a MiniDART device, which combines a miniscope and a short implantable multimode fiber. For future *in vivo* experiments, the MMF and miniscope baseplate should be glued on the mouse skull with dental cement in the same way typical miniscope experiments are performed with GRIN lenses.

R3.Q3 - Time is given in "a.u.". Is that a typo or does this mean that recordings were not made in real time (presumably at longer exposure times)?

We modified the figures in the manuscript by changing a.u. to video frames. Indeed, to generate the synthetic fluorescence time traces, the recorded fingerprint patterns should display pixels whose the amplitude signal have an exponential decay over the video frames, equivalent to GCaMP6 time decay. A brief explanation on how to generate video frames with such analog fluorescence dynamics profiles by modulating a fast DMD is now included in the supp info S01. For each transient peak, the excitation dwell time was progressively smaller in relation to a constant and longer camera exposure time, allowing to create video frames with a characteristic fluorescence transient profiles typical from the literature. We added in the supporting info the detection exposure time per frame used on each experiment from the main text.

The resulting exposure times used in our experiment are therefore longer than typical exposure times used in calcium imaging experiments. Nevertheless, we have shown in a previous work that NMF ability to demix spatiotemporal signals was mostly independent of exposure time, for exposure times ranging between 50 ms and 400 ms, in conditions similar to the experiments shown here (see suppl Fig 10 in Moretti C. and Gigan S., *Nature Photonics* (2020)). In the figures we displayed scattering patterns detected with the miniscope (Figure 04, Figure S03, and Figure S04), we show that the current low-cost miniscope V4.4 manages to detect a single source with 100ms and 33ms exposure times (Miniscope FPS = 10 Hz and 30 Hz respectively). Finally, efficient denoising algorithms such as deepCAD (Li X. *et al.*, *Nature Methods* (2021), <https://doi.org/10.1038/s41592-021-01225-0>) could readily be used in our data to improve SNR before demixing spatio-temporal signals if needed. Therefore, we believe that the use of longer exposure times does not affect the applicability of our experiments to real calcium imaging conditions.

R3.Q4 - It's a clever idea to use projection of temporal patterns onto beads as a first ground-truth evaluation (section 2.1), but the added benefit of repeating this approach with a fixed slice (section 2.2) is not clear. The slice is basically being used as a projection screen.

The proof-of-principle experiment with 6 beads (Figure 02.1) exemplifies the general idea of the method and indicates some possible limitations when low SNR patterns are detected in the video recording due to the position of sources close to the fiber core edge. Hence, it would be important to know if in a real labeled brain we would obtain patterns with enough contrast and signal for NMF to demix. Indeed, in brain tissue, neurons are surrounded by a scattering tissue with fluorescence background, and the typical labels (GFP) are 10x dimmer than fluorescence beads. In addition, neuron somata are not perfectly round. . The experiment of Figure 03 mimics these conditions: it is carried out in a fixed brain slice with dim labels, excitation light is scattering through the sample and therefore illuminating not perfectly round shapes, and fluorescence out-of-focus background can reach the camera; in these conditions, , we show that NMF can demix individual time traces from 4 neurons even though the GT time traces were well correlated.

R3.Q5 - Similarly, the purpose of section 2.3. is not clear. It shows that data from sparse bright beads can be recorded and unmixed on a miniscope camera. Based on section 2.1 we already know that this approach works in general. Presumably the purpose of using a miniscope was to make conclusions about realistic imaging scenarios, but what new information do we actually get about realistic configurations when the samples are sparsely illuminated bright beads without background (see below)?

As we mentioned in the manuscript, the performance of the NMF depends on detecting spatial fingerprints from the sources, thus it is important to know if currently available miniscopes would have enough sensitivity and resolution power to detect the details of the spatial fingerprints that emerge from the MMFs. Therefore, we show in Figure 04 and in the supplementary information that a current version of a low cost miniscope has enough sensitivity to detect a single source fingerprint and also that the morphology of the fingerprint is strongly dependent on the source position along a radius of the fiber.

R3.Q6 - The scenarios that are supposed to mimic physiological conditions are quite unrealistic and some main limitations for in vivo imaging are not discussed. Most notably, the projected and unmixed traces seem almost noise-free. Real data has plenty of noise, mostly due to shot noise from a limited number of signal and background photons, which would very likely affect the ability to unmix the signals. As far as I can tell (not stated in the Methods), the ground truth illumination focused light on the sources only, avoiding any background fluorescence. Exposure times and illumination intensities are not given. I suggest a detailed study of how signal quality is affected by fluorescence background level, camera well depth limitations and labeling sparsity. At least, this should be treated in theory or by simulations with realistic parameters, but the manuscript would further benefit from realistic practical demonstrations. For example, authors could use a real miniscope video recording (published or recorded by the authors), tune light intensity to realistic levels and project it onto the input face of the MMF, while trying to unmix the signals on the other end.

We agreed with the reviewer that in our previous results we had too simplistic time traces and low source number to be demixed by NMF. Here, we now present new results (Figure 02.2 and Figure02.3) from new experiments that experimentally mimicked more realistically the possible fluorescence signal coming from a living brain, as already mentioned in previous answers (R1.Q2; R1.Q4; R2.Q3; R3.Q4) and at the initial message. We chose to answer the reviewers' questions experimentally instead of theoretically, because we thought that it would be more convincing. The quantitative values of exposure time and camera dynamic range are now given in the discussion of main text and in the supp info S01.

In addition, previous works from our team have already evaluated the performance of NMF depending on the noise, background, and the length of the video recording (Moretti C. and Gigan S. *Nature Photonics* (2020); Soldevila, F. *et al. Optics Express* (2023)). As previously mentioned in the question R1.Q4, NMF performance starts to slowly get impaired when the max value of the background signal is 1.5x bigger than the max value of the sources' activity time trace signal. In this case, the median temporal cross-correlation gets lower than 80%, with the first quartile of ~65%. Nevertheless, even when the max background over the max activity ratio is of 2.0, the median cross-correlation in the time trace is still above 70%, with the first quartile above 60% (Moretti C. and Gigan S. *Nature Photonics* (2020) <https://doi.org/10.1038/s41566-020-0612-7>). In our new experiments we present here, we now included a fast oscillating neuropil source with a dynamic non-sparse signal which has a strong overlap with the other sources and has the same order of magnitude as the transient peaks of the sources (see the ensemble time trace of Figure 02.2 and Figure02.3). Typically, the neuropil amplitude signal is not 2x higher than the transient peak maximum, hence we believe NMF could easily decompose its signal as an extra component source. For extreme cases in which the background buries the signal, it has already been shown in the literature that temporally constrained NMF (CNMF) can retrieve well the time traces from neurons in wide-field 1-photon microscopy [Pnevmatikakis, E.A. *et al. Neuron* (2016). <https://doi.org/10.1016/j.neuron.2015.11.037>; Zhou P. *et al. eLife* (2018) <https://doi.org/10.7554/eLife.28728>]. NMF could also be constrained in the spatial domain, profiting from the deterministic nature of the signal, such as the one in CNMF-E with a double sparsity constraint (in time and space) (Zhou P. *et al. eLife* (2018)), with significant improvement in NMF performance capacity in extracting calcium transient signals. We chose to use the simplest version of NMF in this first proof-of-principle paper to keep as general as possible. As a final remark, our team has already shown that the NMF performance improves for longer video recordings (Moretti C. and Gigan S. *Nature Photonics* (2020)), thus longer recordings are preferable for noisier data.

R3.Q7 - The robustness of the approach against movement of the sample and against bending of the MMF should be evaluated.

Regarding movements of the sample, in the case of 2-photon imaging experiments in the cortex with a cranial window, motion artifacts were 2-4 μm at z distances shorter than 150 μm from the optical window (Akemann, W. *et al. Fast optical recording of neuronal activity by three-dimensional custom-access serial holography. Nat Methods* **19**, (2022)). In our case, we expect similar motion artifacts when exploring shallow regions of the brain, and smaller artifacts for deep regions. Indeed, this is what has been observed for 2-photon imaging with GRIN lenses (Bocarsly M.E., *et al. Biomedical Optics Express* (2015) <https://doi.org/10.1364/BOE.6.004546>; Meng G. *et al. eLife* (2019) <https://doi.org/10.7554/eLife.40809>). In addition, since in our experiments the patterns smoothly change upon source motion in the distal FoV (see Figure 04, supplementary information Figures S03 and S04, and tiff video of pattern dependence on radial position of the source), we expect that motion artifact to remain small, periodic, and restricted in space, and we expect NMF to extract an average pattern for each source. However, we agree that motion artifacts should be properly quantified in the future with a rigorous proof-of-principle experiment *in vivo*.

Regarding the flexibility of the fiber, typical studies on multimode fiber bending effects are done with relatively long fibers (> 100 mm, typically around 300 mm long), which are quite flexible to be bent (Thorlabs manufacturer recommends bending fibers until a maximum of 21 mm of radius of curvature for 200 μm diameter core fibers). However, light propagation studies dealing with extreme bending cases of fibers with similar core as the one we used (200 μm diameter) display typical smallest bending radius of around 5 mm (Cao H. *et al. Adv. Opt. Photon.* (2023) <https://doi.org/10.1364/AOP.484298>), which is the typical length of the short MMF we use. Like any other solid material, shorter multimode fibers (~10 mm) are way stiffer to bending than long fibers (> 100 mm) (Euler–Bernoulli beam theory of solid materials mechanics; Glaesemann G.S. *Review of Research at Corning's Optical Fiber Strength Laboratory* 2017; Glaesemann G.S. *Proc. Int. Wire Cable Symp.*(1991), Mallinder F.P. *et al. Phys. Chem. Glasses* (1964),

Matthewson J.M. Proc. SPIE Critical Review Series CR50 (1993)). Typically, even long fibers have critical ~6 mm bending radius (with typical bending stress of ~700 MPa), where the Young modulus of multimode fibers changes very little (less than 1%) and its value is for practical purposes considered constant (see Figure R1 below, which is the same as Figure 3 from Glaesemann G.S. Optical Fiber Mechanical Reliability. Review of Research at Corning's Optical Fiber Strenght Laboratory. White Paper WP8002. July 2017. <https://www.corning.com/media/worldwide/coc/documents/Fiber/white-paper/WP8002.pdf>). Therefore, short fibers as the ones we used (~ 8mm long) are significantly rigid and very difficult to be bent. This is particularly true for in living mouse experiments, where one end of the fiber (proximal) will be glued on the mouse's skull, and only the distal end of the fiber will be "free" to be bent (i.e., this end will be actually be surrounded by the mouse brain, dumping the small internal movements, see below). In fact, most of the fiber length is rigidly glued within the ferrule, letting only a small portion of the fiber "free" to be bent (typically, in between ~1mm to ~4mm long sticking out of the ferrule, which is below the critical length of the fiber). Therefore, we expect very little bending of the fiber during in vivo recordings.

Figure R1 - Maximum bending stress vs bend radius of multimode fibers for 80, 125, and 200 μm diameter fibers (figure taken from: Glaesemann G.S. Optical Fiber Mechanical Reliability. Review of Research at Corning's Optical Fiber Strenght Laboratory. White Paper WP8002. July 2017. <https://www.corning.com/media/worldwide/coc/documents/Fiber/white-paper/WP8002.pdf>). Changing the curvature for values below 20mm need significantly higher bending stress (MPa) until the fiber breaks with approximately 4mm.

We added in the following paragraph about fiber bending in the discussion section of the manuscript: "Typical studies on multimode fiber bending effects are done with relatively long fibers (> 100 mm, typically around 300 mm long), which are quite flexible to be bent (Thorlabs manufacturer recommends bending fibers until a maximum of 21 mm of radius of curvature for 200 μm diameter core fibers). However, light propagation studies dealing with extreme bending cases of fibers with similar core as the one we used (200 μm diameter) display typical smallest bending radius of around 5 mm, which is the typical length of the short MMF we use⁴⁷. Like any other solid material, shorter multimode fibers (~10 mm) are way stiffer to bending than long fibers (> 100 mm) (Euler–Bernoulli beam theory of solid materials mechanics)[Glaesemann G.S. Review of Research at Corning's Optical Fiber Strenght Laboratory 2017; Glaesemann G.S. Proc. Int. Wire Cable Symp.(1991), Mallinder F.P. *et al.* Phys. Chem. Glasses (1964), Matthewson J.M. Proc. SPIE Critical Review Series CR50 (1993)]. Typically, even long fibers have critical ~6 mm bending radius (with typical bending stress of ~700 MPa), where the Young modulus of multimode fibers changes very little (less than 1%) and its value is for practical purposes considered constant [Glaesemann G.S. Review of Research at Corning's Optical Fiber Strenght Laboratory 2017]. Therefore, short fibers as the ones we used (~ 8mm long) are significantly rigid and very difficult to be bent. This is particularly true for in living mouse experiments, where one end of the fiber (proximal) will be glued on the mouse's skull, and only the distal end of the fiber will be "free" to be bent (i.e., this end will be actually be surrounded by the

mouse brain, dumping the small internal movements [Bocarsly M.E., *et al.* *Biomedical Optics Express* (2015) <https://doi.org/10.1364/BOE.6.004546>; Meng G. *et al.* *eLife* (2019) <https://doi.org/10.7554/eLife.40805>]. In fact, most of the fiber length is rigidly glued within the ferrule, letting only a small portion of the fiber “free” to be bent (typically, in between ~1mm to ~4mm long sticking out of the ferrule, which is below the critical length of the fiber). Therefore, we expect very little bending of the fiber during in vivo recordings.

R3.Q8 – Other

Please check language and style throughout the document. The discussion is particularly notable for its language and requires editing.

"there is plenty of room for improvement, not only optically wise" (see also comment above), "simply add some extra data processing steps...". If the proposed improvements are as straightforward as the discussion suggests, it would be beneficial to see these implemented in the study. Alternatively, please rephrase.

The discussion is very verbose on possible improvements, but not much space is used to critically discuss key limitations for the main target application.

I suggest removing figure 4e unless you plan on including such recordings. Otherwise it can mislead the reader.

We have thoroughly gone through the discussion and edited for language. The general reason while we left some technical steps such as data processing improvement for later is that we wanted to establish as clearly as possible a general proof-of-principle for the technique, focusing on a solid ground truth, keeping the algorithms as “vanilla” as possible, so as to underlay the physical concepts. In fact, the unconstrained NMF used here is a general mathematical method that simply decomposes a matrix into a product of two matrices where all the entries are non-negative. More tailored NMF implementations, for example using sparsity or other image-based and/or temporal constraints based on the specific nature of our experiments, would probably yield better results in terms of fidelity or achieving a higher number or retrieved traces [Pnevmatikakis, E.A. *et al.* *Neuron* (2016). <https://doi.org/10.1016/j.neuron.2015.11.037>; Zhou P. *et al.* *eLife* (2018) <https://doi.org/10.7554/eLife.23728>]. However, even though we expect significant quantitative improvements from those approaches, their implementation is not technically straightforward and it will be the aim of future work. We hope that with the updated version, and the extra experiments we presented, the discussion is now more balanced and reflects better the pros and cons of the new method.

Finally, regarding Figure 04-e (now Figure 04-d), we decided to keep it in the paper to show the perspective of the work and justify why the last characterization has been performed using a miniscope, but we significantly reduced its size, and indicated on top of the figure that it corresponds to future prospects.

References used in this rebuttal (in alphabetical order):

- Aonishi, T. *et al.* Imaging data analysis using non-negative matrix factorization. *Neuroscience Research* vol. 179 51–56 Preprint at <https://doi.org/10.1016/j.neures.2021.12.001> (2022).
- Bianco, M. *et al.* Orthogonalization of far-field detection in tapered optical fibers for depth-selective fiber photometry in brain tissue. *APL Photonics* 1 February 2022; 7 (2): 026106. <https://doi.org/10.1063/5.0073594>
- Blochet B, Akemann W, Gigan S, Bourdieu L. Fast wavefront shaping for two-photon brain imaging with multipatch correction. *Proc Natl Acad Sci U S A.* 2023 Dec 19;120(51):e2305593120. doi: 10.1073/pnas.2305593120. Epub 2023 Dec 15. PMID: 38100413; PMCID: PMC10743372.

Bocarsly M.E., et al. Minimally invasive microendoscopy system for in vivo functional imaging of deep nuclei in the mouse brain. *Biomed Opt Express* (2015) <https://doi.org/10.1364/BOE.6.004546>

Boniface, A., Dong, J. & Gigan, S. Non-invasive focusing and imaging in scattering media with a fluorescence-based transmission matrix. *Nat Commun* 11, (2020). <https://doi.org/10.1364/OPTICA.6.001381>

Boniface, A. Light control in scattering media and computational fluorescence imaging: towards microscopy deep inside biological tissues. (Laboratoire Kastler Brossel, L'Université Pierre et Marie Curie, 2020)

Cao, H., Čížmár, T., Turtaev, S., Tyc, T. & Rotter, S. Controlling light propagation in multimode fibers for imaging, spectroscopy, and beyond. *Adv Opt Photonics* 15, 524 (2023) <https://doi.org/10.1364/AOP.484298>

Glaesemann, G. S. Optical Fiber Mechanical Reliability. Review of Research at Corning's Optical Fiber Strength Laboratory. White paper. *Corning Incorporated* vol. WP8002 1–62 Preprint at <https://www.corning.com/media/worldwide/coc/documents/Fiber/white-paper/WP8002.pdf> (2017).

Glaesemann, G. S. Optical fiber failure probability predictions from long-length strength distributions. in *The 40th international wire and cable symposium proceedings* (Corning Incorporated, 1991). *Proc. Int. Wire Cable Symp.*, 40, 819-825 (1991)

Grødem, S., Nymoen, I., Vatne, G.H. et al. An updated suite of viral vectors for in vivo calcium imaging using intracerebral and retro-orbital injections in male mice. *Nat Commun* 14, 608 (2023). <https://doi.org/10.1038/s41467-023-36324-3>

Li, X. et al. Reinforcing neuron extraction and spike inference in calcium imaging using deep self-supervised denoising. *Nat Methods* 18, 1395–1400 (2021). <https://doi.org/10.1038/s41592-021-01225-0>

LIN, B., TAO, X., QIN, X., DUAN, Y. & LU, J. Hyperspectral image denoising via nonnegative matrix factorization and convolutional neural networks. in *IEEE International Geoscience and Remote Sensing Symposium (IGARSS)* 4023–4026 (IEEE, 2018). <https://doi.org/10.1109/IGARSS.2018.8517388> .

Mallinder, F. P. & Proctor, B. A. Elastic constants of fused silica as a function of large tensile strain. in *Phys, Chem. Glasses* vol. 5 [4] 91–103 (1964).

Matthewson, M. J. Optical fiber mechanical testing techniques. in *Fiber Optics Reliability and Testing: A Critical Review* vol. 10272 1027205 (SPIE, 1993).

Meng G. et al. High-throughput synapse-resolving two-photon fluorescence microendoscopy for deep-brain volumetric imaging in vivo. *eLife* (2019) <https://doi.org/10.7554/eLife.40805>

Nöbauer, T., Skocek, O., Pernía-Andrade, A. et al. Video rate volumetric Ca²⁺ imaging across cortex using seeded iterative demixing (SID) microscopy. *Nat Methods* 14, 811–818 (2017). <https://doi.org/10.1038/nmeth.4341>

Papadopoulos IN, Farahi S, Moser C, Psaltis D. High-resolution, lensless endoscope based on digital scanning through a multimode optical fiber. *Biomed Opt Express*. 2013 Feb 1;4(2):260-70. <https://doi.org/10.1364/BOE.4.000260>. Epub 2013 Jan 17. PMID: 23411747; PMCID: PMC3567713.

Papadopoulos IN, Farahi S, Moser C, Psaltis D. Focusing and scanning light through a multimode optical fiber using digital phase conjugation. *Opt Express*. 2012 May 7;20(10):10583-90. <https://doi.org/10.1364/OE.20.010583> . PMID: 22565684.

Pnevmatikakis, E. A. et al. Simultaneous Denoising, Deconvolution, and Demixing of Calcium Imaging Data. *Neuron* 89, 285 (2016). <https://doi.org/10.1016/j.neuron.2015.11.037>.

Turtaev, S., Leite, I.T., Altwegg-Boussac, T. *et al.* High-fidelity multimode fibre-based endoscopy for deep brain in vivo imaging. *Light Sci Appl* 7, 92 (2018). <https://doi.org/10.1038/s41377-018-0094-x>

Varghese, K., Kolhekar, M. M. & Hande, S. Denoising of Facial Images Using Non-Negative Matrix Factorization with Sparseness Constraint. in *2018 3rd International Conference for Convergence in Technology (I2CT)* 1–4 (IEEE, 2018). <https://doi.org/10.1109/I2CT.2018.8529796> .

Zhang, Y., Rózsa, M., Liang, Y. *et al.* Fast and sensitive GCaMP calcium indicators for imaging neural populations. *Nature* 615, 884–891 (2023). <https://doi.org/10.1038/s41586-023-05828-9>

Zhou, P. *et al.* Efficient and accurate extraction of in vivo calcium signals from microendoscopic video data. *Elife* 7, (2018) <https://doi.org/10.7554/eLife.28728>

REVIEWER COMMENTS

Reviewer #1 (Remarks to the Author):

I would suggest the revised manuscript for the publication. Although there is no biological demonstration the idea is very interesting. There is also an appeal to a wider imaging community in bio/ neuroscience.

I have few more comments that might be helpful for the authors.

1. Please revise the newly written text, avoid jargon expressions as line 424: "keeping the algorithms as "vanilla" as possible".

2. I do not really agree with the authors model of the neuropil background (line 203). Neuropil is not only temporally but also spatially distributed signal (often highly correlated), fluorescence in one bead could hardly account for it. Please see the example of neuropil from miniscope recording here: <https://www.youtube.com/watch?v=RslP-WQUIK8> . The signal is bright and is distributed across the field of view. Usually spatial filtering is used to get rid of neuropil, in addition to extracting neurons as rois.

3. I am not sure prafilm (scattering) brakes the symmetry of the fiber (line 205), instead it modifies the wavefront such that it changes the composition of modes the fluorescence pattern is coupled to.

Reviewer #2 (Remarks to the Author):

I would like to thank the authors for the articulated response they gave to the referees comments. I appreciate that, based on our suggestions, they decided to add more experiments and discussions within the paper.

I believe that the neuropil like experiment and the scattering through the parafilm, add valuable insight to evaluate the potential of the method.

I also think we could debate forever if these results can or cannot be extrapolated to real in vivo imaging of calcium activity. Only one such experiment will give the final answer. My idea is still that in vivo imaging will be much more challenging. I've been doing in vivo calcium imaging myself for many years now and I can guarantee that the signals are much noisier and have much stronger backgrounds than what the authors consider.

However, because I believe strategies similar to the one presented in this paper can really make the calcium imaging community leap forward, I am ready to recommend publication, but I would still ask a couple of changes and one experiment, which I believe should not be too difficult for the authors to perform.

Thanks to Fig.2 and 3 we now know that background signals can confuse the algorithm, especially when it is close to neurons and in the presence of scattering. Now, 1P widefield calcium imaging in the cortex for example, is characterized by very high background fluorescence. Look for instance at the videos of this recent paper 'Rapid detection of neurons in widefield calcium imaging datasets after training with synthetic data'. There is a uniform fluorescence background everywhere in the raw data, which also can vary in time, and then dim cells appear on top. GCaMP8 might make the game a bit easier, but widefield 1P excitation is bound to give high background. Even if we assume that the authors can demix well cells at different planes thanks to the scattering effect that would change the look of the fingerprint (nice idea), the neuropil and resting fluorescence of certain cells will be a much stronger problem, I believe, than what the authors assume. The neuropil is not just one point in the field of view, but it is rather a distributed background. Not even in 2P microscopy you find areas on one plane that have zero counts.

My recommendations to the authors are thus the following:

Please discuss more and better this problem from the beginning (and again in the discussion). When you describe the proof of principle experiment, it is already the moment to say that it does not take into account several aspects that would make the process less reliable in vivo. 1) The time traces used are not raw time traces, but $\Delta F/F$ in which F_0 is set to zero; 2) The effect of widefield excitation that could excite at the same time hundreds (or thousands) of cells at different planes, with non-zero resting fluorescence and a distributed neuropil effect is only marginally taken into account.

I typically would not ask more experiments in a second round of revision, but I think there is clearly one more important point that can be added to Fig.3. What happens if you now, on the same FOV, start to add more independent neuropil like components? My idea of this experiment is the following. Take a sample with the maximum number of beads that you can find. Select 20-30 beads as 'neurons' and ~ 5 as neuropil, in different positions, not very far from the neuron beads. Now run the demixing routine while only sending neuronal signals. You should get all the 20-30 neurons. Next, start one by one to add the neuropils and see each time how many neurons traces you actually lose. This experiment is not designed to say that the method is wrong if after 5 neuropils no neurons is found anymore. My publication recommendation will not change depending on the output of the experiment. But it will add all the context necessary so that people will be more able to understand if it could or not work for their applications.

I believe that after this new round of revision the paper will be ready for publication.

Reviewer #3 (Remarks to the Author):

The rebuttal was very thorough and addressed all main concerns regarding the presentation, claims and discussion. I support publication.

Rebuttal #2 to the reviewers of Nature Communications

Initial message:

The authors would like to thank all the reviewers for their positive assessment of our previous version of the manuscript and for their constructive criticism during the revision process. Due to this positive interaction and all the insightful discussions, we believe that the quality of the current version of the manuscript has improved significantly and we are grateful for that. With this response, we hope to address the final concerns of the reviewers. Before responding point-by-point to the reviewers' comments, we would first like to give a general overview of the new results we present.

First, the concerns of reviewers #1 and #2 about how we modeled the neuropil are valid. Indeed, the neuropil is typically a fluctuating fluorescence background signal that surrounds the soma (neuron cell body). Neuropil fluorescence is an unwanted signal which is typically difficult to isolate via image segmentation for the extraction of the GECI traces from the target ROIs (somas). As rightly mentioned by the reviewers, neuropil signals are very strong in one-photon (1P) calcium imaging *without* soma-tag labeling. Although there are many advances described in the literature on how to reduce the influence of neuropil (experimentally with soma-tag, and computationally with constrained NMF, CNMF), we agree that it would be of general interest to know to what point the standard NMF would work without relying on a specific labeling strategy (soma-tag). Thus, we followed the reviewer's recommendations and designed a more realistic model of neuropil in a new set of experiments to test their last concern.

In this new set of experiments, we designed a new neuropil signal in which (1) the excitation amplitude (photon budget) of the "neuropil sources" was dominant compared to the ensemble signal generated by the target sources; and (2) the "neuropil sources" should have a delocalized distribution over the FoV, i.e., the signal should come from different beads around the FoV. Moreover, the selected neuropil sources should be at the close vicinity of the target sources. To do this, we searched for ROIs where we could find a good number of aggregated beads. Neighbor beads were assigned to be either "target" or "neuropil".

The new results are presented in the main text and in the supporting information of the manuscript and are detailed here in rebuttal#2. We have a total of 21 beads in the FoV, and, following the recommendation of reviewer 2, we progressively increased the number of "neuropil" sources, with respect to "target" sources. In the most challenging condition we tested, we chose 11 neuropil sources and 10 target sources, resulting in a total neuropil signal 6 times larger than the total target signal. Nevertheless, in these conditions, the standard unconstrained NMF was able to retrieve 9 of the 10 target sources with very high accuracy (temporal correlation with GT time traces larger than 80%, with an average value of 86% and standard deviation of 5.5%).

We now will present the main changes in the manuscript (highlighted in **yellow**), and then answer point-by-point the reviewers' comments. Text in *italic* indicates the old main text.

Changes in the MAIN TEXT:

- **In the introduction, we included a sentence indicating that we validated the case when we challenged NMF with a strong neuropil:**

*"In the present work, we designed a proof-of-principle experiment (Figure 01) and we confirmed that a simple unconstrained NMF could also disentangle the short MMFs scattering fingerprints signals and retrieve their corresponding time traces. As a consequence, one may now temporally resolve and count the number of sources with singular time traces transmitted by a minimally invasive multimode fiber. Thus, the results of this paper consist of a proof of concept on how to obtain individual time trace resolution in fiber photometry methods. Additionally, we progressively validated our approach towards more realistic conditions, such as demixing fluorescence signal from tens of bead sources buried behind a scattering media (plastic paraffin: Parafilm M®) including a component for neuropil activity, and by selectively probing a few structurally Gad-eGFP labeled neurons in a 50 µm fixed brain slice with literature-available time traces to mimic neuronal activity. **We also validated the method when the signal from the mimicked neuropil is dominant compared to the signal corresponding to the mimicked cell bodies (soma).** Finally, we propose a novel method for probing neuronal microendoscopic signals by simply combining a miniscope and an implantable short multimode fiber, which we called MiniDART (for Miniaturized Deep Activity Recording with high Throughput).*

For that, we demonstrate that the inexpensive and commercially available open-source miniscope (Open Ephys Miniscope-v4.4) has already enough sensitivity and illumination power to detect the typical intricate patterns of short MMFs. This new way of measuring individual fluorescence time traces from an ensemble of fluctuating sources profits from the short length of multimode fibers that are naturally more rigid (bending resistant) and therefore very suitable to be used in long-term freely-moving mice neuroscience experiments.”

- **In the result section 2.1, we included a paragraph detailing the time traces we modelled (as suggested by reviewer#2):**

*“2.1 Proof-of-principle **experiment** using phantom samples made of 10 μm diameter fluorescent beads*

*To demonstrate the validity of the method, we implemented an optical setup using a digital micromirror device (DMD), which was used to generate different excitation ground truth (GT)^{51,52,66,67} activity traces for each fluorescent source (10 μm diameter fluorescent beads \approx neuron soma size). Each source emits fluorescence that is collected and transmitted by the multimode fiber (see Figure 01 and methods for details). Upon propagating through the MMF, the fluorescence wavefront undergoes scrambling, resulting in the emergence of fluorescence patterns upon exiting the fiber. The controlled excitation guarantees that each fluorescent source generates a fingerprint pattern whose intensity transiently fluctuates accordingly with the chosen GT time trace profile (see transient patterns in Figure 01). **We designed GT time traces to be equivalent to optical recording experiments of GECl time traces where calcium signals had FO set to zero. We then recorded a video of the transient patterns that emerge from a short multimode fiber and applied NMF to the recorded raw data without doing any pre-processing step. In other sets of experiments later, we selected one or more sources available in the FoV to mimick neuropil signal, by exciting them using a non-sparse GT signal (see sections below).***

- **In the result section 2.2 and 2.3, we changed Figure 02.2 and Figure02.3 by highlighting in orange the neuropil source, pattern and time trace in the figures.**

2.2 NMF demixing of densely superimposed spatiotemporal signals including neuropil dynamic background

Figure 02.2 – Results of a proof-of-principle experiment performed with 26 fluorescent beads including a neuropil background source. The bead #20 is modeling the neuropil (highlighted in orange). From (a) to (d) we have: (a) the photometric (ensemble) time trace, which is the sum of 26 time traces; (b) the sCMOS detected image of the spatially overlapped fingerprint patterns from 26 fluorescent beads probed by the short MMF (see methods); (c) the short MMF located at a distance of $60 \pm 10 \mu\text{m}$ from the sample; (d) the ground truth image of the sample (backpropagated fluorescence image detected from a CMOS Basler camera, see details of the setup in the supp info). (e) The ground truth (GT) fingerprint patterns obtained from each bead when they were individually excited. (f) The fingerprint patterns obtained via NMF are to be compared with the GT patterns in e. (g) Top: the individual temporal activity traces obtained with NMF (blue) and their corresponding GT traces (gray). Bottom: the photometric ensemble signal from the recorded video (black line), which is the sum of all individual traces. The fluorescence intensity in all traces in the figure are normalized to 1. (h) The GT-NMF time trace correlations. The average diagonal value of the first 22 beads was $\langle \delta_{g,n} \rangle = \delta_{avg} = 86.0\%$ with $\sigma_{\delta} = 5.4\%$. To better evaluate the off-diagonal elements (time trace cross-talk), we subtract them from their corresponding GT-GT coefficients. Then, we averaged the absolute values of these differences and we obtained the mean cross-talk of $\zeta_{avg} = 4.4\%$ with a standard deviation of $\sigma_{\zeta} = 3.7\%$ for the first 22 beads (see supp info). (i) The GT-GT temporal trace correlation table showing that the ground truth traces were not orthogonal.

2.3 NMF demixing of multiple source photometric signals buried below a scattering layer

Figure 02.3 – Results of a proof-of-principle experiment performed with 26 fluorescent beads behind a Parafilm M® layer. The bead #13 is the source mimicking neuropil background signal (highlighted in orange). From (a) to (d) we have: (a) the photometric (ensemble) time trace, which is the sum of 26 time traces; (b) the sCMOS detected image of the spatially overlapped fingerprint patterns from 26 fluorescent beads simultaneously probed by the short MMF (see methods); (c) the short MMF located at a distance of $60 \pm 10 \mu\text{m}$ from the sample; (d) the ground truth image of the sample (backpropagated fluorescence image detected from a CMOS Basler camera, see setup in the supp info). (e) The ground truth (GT) fingerprint patterns obtained from each bead when they were individually excited. (f) The fingerprint patterns obtained via NMF are to be compared with the GT patterns in (e). (g) Top: the individual temporal activity traces obtained with NMF (blue) and their corresponding GT traces (gray). Bottom: the photometric signal from the recorded video (black line), which is the sum of all individual traces. The fluorescence intensity in all traces in the figure are normalized to 1. (h) The GT-NMF time trace correlations. The average diagonal value of the first 20 beads was $\langle \delta_{g,n} \rangle = \delta_{avg} = 77.0\%$ with $\sigma_{\delta} = 11.9\%$. To better evaluate the off-diagonal elements (time trace cross-talk), we subtract them from their corresponding GT-GT coefficients. Then, we averaged the absolute values of these differences and we obtained the mean cross-talk of $\zeta_{avg} = 5.9\%$ with a standard deviation of $\sigma_{\zeta} = 5.6\%$ for the first 20 beads (see supp info). (i) The GT-GT temporal trace correlation table showing that the ground truth traces were not orthogonal - some of them were correlated.

- Added a new result section in the main text (Results 2.4): testing NMF in conditions where the neuropil signal dominates

2.4 NMF demixing of signal from multiple sources hidden by dominant neuropil activity

Previous work by some of the authors has already shown that unconstrained NMF can successfully demix fluorescence time traces from overlapping speckle patterns from a high level of fluorescence background^{51,52}. In the last experiments we showed here, we chose to have only one source mimicking neuropil signal, with an amplitude comparable to the amplitude of the bright peaks of each of the other individual time traces. Thus, it remained an open question whether unconstrained NMF could demix the time traces from overlapping scattering fingerprints transmitted by short multimode fibers in conditions where the fluorescence background is dominant, as demonstrated in the previous works.

To address this question, we performed a new set of experiments in which we progressively increased the number of sources mimicking neuropil, and thus progressively increased the strength of the neuropil signal compared to the target sources. More specifically, we chose a new FoV containing 21 beads in total, and we excited 1, 5, or 11 sources with the same neuropil-like non-sparse temporal signal (see figures O2.4, S08, and S09). As expected, unconstrained NMF was able to successfully demix most of the individual time traces of the target beads mimicking cell bodies (“target sources”) with high temporal correlation accuracy ($> 80\%$), even when 11 of 21 sources were mimicking neuropil (see Figure O2.4). In such an extreme case, the signal from the total neuropil-like background was approximately 6x stronger on average than the whole ensemble target signal (see Figure S08m), and approximately 10x larger than the maximum peak of each target source signal. Yet, NMF was able to retrieve 9 of the 10 remaining target sources with very high accuracy (temporal correlation with GT time traces larger than 80%, with an average value of 86% and standard deviation of 5.5%, see Figures O2.4). Importantly, as can be seen in Figure O2.4d, the target sources were spatially aggregated with many neuropil sources, which is often the case in GECI imaging experiments.

Figure O2.4 – Results of a proof-of-principle experiment performed with 21 fluorescent beads, where 10 beads had sparse and unique time traces and the other 11 had the same non-sparse neuropil-like time trace (dynamic background). The beads with “neuropil-like” background activity are highlighted in orange (#1, #11, #12, #13, #14, #15, #16, #17, #18, #19, and #20). The remaining beads (#2, #3, #4, #5, #6, #7, #8, #9, #10, and #21) had unique sparse neuronal activity time traces mimicking signal from neuronal cell bodies (target sources). (a) the photometric (ensemble) time trace, which is the sum of all the 21 time traces; (b) the sCMOS detected image of the spatially overlapped fingerprint patterns from 21 fluorescent beads simultaneously probed by the short MMF (see methods); (c) the short MMF located at a distance of $60 \pm 10 \mu\text{m}$ from the sample; (d) the ground truth image of the sample (backpropagated fluorescence image detected from a CMOS Basler camera, see setup in the supp info Figure S01). (e) The ground truth (GT) fingerprint patterns obtained from each bead when they were individually excited. (f) The fingerprint patterns obtained via NMF. Note that the NMF pattern #1 is the neuropil pattern due to the spatial overlap of 11 sources (highlighted with orange squared boxes). (g) Top: the individual time traces obtained with NMF (blue) and their corresponding GT traces (gray). Bottom: the photometric signal from the recorded video (black line), which is the sum of all individual traces. The fluorescence intensity in all traces in the figure are normalized to 1. (h) The GT-NMF time trace correlations. The average diagonal value of the first 10 beads was $\langle \delta_{g,n} \rangle = \delta_{avg} = 86.0\%$ with $\sigma_\delta = 5.5\%$. To better evaluate the off-diagonal elements (time trace cross-talk), we subtract them from their corresponding GT-GT coefficients. Then, we averaged the absolute values of these differences and we obtained the mean cross-talk of $\zeta_{avg} = 6.1\%$ with a standard deviation of $\sigma_\zeta = 5.7\%$ for the first 10 beads (see supp info). (i) The GT-GT temporal trace correlation table showing that the ground truth traces were not orthogonal.

- In the result section 2.5, we corrected some typos in the Figure O3 caption

2.5 Validation of the method while probing structurally GFP-labeled neurons in a 50 μm thick fixed brain slice

Figure 03 – Validation of the concept of single-activity resolved fiber photometry with short multimode fibers (short MMF) in a brain tissue environment (*in vitro*). Sample: Gad-EGFP neurons fixed in a 50 μm brain slice, sealed in between 2 coverslips to keep the humidity of the tissue (see methods). (a) The ensemble photometric time trace of this experiment. (b) the fiber proximal end image of 4 neurons' fingerprint patterns spatially overlapped on the sCMOS camera chip. (c) an illustration of the short MMF placed above the top coverslip of the sample, at a distance of $\approx 60 \pm 10 \mu\text{m}$ from it; and (d) the GT image of the sample highlighting the 4 selected neurons to be excited (structurally labeled). (e) The GT fingerprint patterns are obtained from each neuron when individually excited. (f) The fingerprint patterns retrieved via NMF are in good agreement with the GT patterns in (e). (g) The demixed temporal activity traces are sorted in descending GT-NMF correlation order (from the best to the worst retrieved). Traces in blue are retrieved by NMF and temporal traces in gray are their GT. (h) The GT-NMF temporal trace correlation coefficients. (i) The GT-GT temporal correlations. The average diagonal value in (h) of the 4 neurons was $\langle \delta_{g,n} \rangle = \delta_{\text{avg}} = 86.7\%$, with standard deviation of $= 2.8\%$. Regarding the non-diagonal elements (cross-talk), the mean absolute error taking into account the GT-GT coefficients was $\zeta_{\text{avg}} = 8.95\%$ with a standard deviation of $\sigma_{\zeta} = 8.02\%$ (see supp info). Again, although each GT trace was unique in time (singular), they were not fully uncorrelated as we can see in the GT-GT correlation traces (i). Interestingly, neurons #1 and #2 (i.e., the two best NMF retrieved results) were also the most temporally correlated ones in the GT excitation ($\nu_{1,2} = \nu_{2,1} = 25.0\%$).

- We improved the text in the caption of Figure 04 from the result section 2.6

2.6 Pattern sensitivity evaluation of low-cost miniscopes while probing a single fluorescence source

Figure 04 – Novel microendoscopy concept using a short MMF and a miniscope: the MiniDART. (a) A typical fingerprint pattern from a single-fluorescent bead ($10\ \mu\text{m}$ diameter) probed by using only the miniscope excitation and miniscope detection through the multimode fiber. (b) The experimental setup to probe scattering patterns from the short MMF includes (i) the miniscope, (ii) a customized titanium base plate (YMETRY®) to hold the miniscope, (iii) a ferrule (Thorlabs SFLC230-10) that rigidly holds the multimode fiber (iv) within it, (v) a sample consisting of a single fluorescent $10\ \mu\text{m}$ bead (spatial density $< 1\ \text{bead}/\text{cm}^2$), and (vi) a customized titanium tweezer (YMETRY®) to hold the ferrule. (c) Scattering fingerprint patterns at the proximal end (bottom row) depending on the radial position d of the single-bead at the distal end (bottom row). Position d (red arrow) is indicated in relation to the fiber axis (the bead is represented as a blue spot in the zoom of (b) and the top row images of (c), while the axial center of the fiber is represented as a fixed black dot in the top row images of (c)). Each pattern acquisition in the proximal end (bottom row of (c)) corresponds to $10\ \mu\text{m}$ steps of the bead from the fiber central axis in the distal end (top row of (c)). The bigger the distance d (red arrow; top) of the bead from the center of the fiber, the larger the radius ρ (white arrow; bottom) of the bright spiral-ring pattern in the proximal end. The diagonal white dashed line is the azimuthal orientation of the red vector d , which always coincides with the alignment angle of the 2 central bright points of the fingerprint patterns in the proximal end (see supp info Figure S03-S04 for details). The highest LED power values measured at the distal end of the fiber (whose core is $200\ \mu\text{m}$ in diameter) were around $9.5\ \mu\text{W}$, which yields an excitation intensity of $0.3\ \text{mW}/\text{mm}^2$ at the output of the fiber core, and $2.4 \times 10^{-5}\ \text{mW}$ excitation power per bead area. Exposure time: $100\ \text{ms}$ (Miniscope FPS = $10\ \text{Hz}$). For more details, see Figures S03 and S04 in the supporting Information. (d) The concept of doing experiments with a MiniDART device, which combines a miniscope and a short implantable multimode fiber. For future in vivo experiments, the MMF and miniscope baseplate should be glued on the mouse skull with dental cement in the same way typical miniscope experiments are performed with GRIN lenses.

- We changed the Discussion section to include the information of the new experimental results

Lines 360-374:

“To this end, we designed an experiment with a known ground truth excitation to be able to evaluate if NMF could demix the individual spatio-temporal readouts characteristic of short MMFs and GECI recordings. In this paper, we demonstrate that it is possible to demix such spatio-temporal signals in vitro, using a simple and general unconstrained NMF algorithm on the video data recorded in our proof-of-principle experiment. We designed in vitro samples to mimic as much as possible the real brain: we used tens of sources embedded in agarose (where a few of them are touching each other), with multiple overlapping temporal transient peaks signals, and we chose one or several sources to mimic fluorescent neuropil. In one experiment, we added a scattering layer between the sources and the fiber, with scattering properties equivalent to that of a $120\ \mu\text{m}$ brain layer. In all of these experiments, we demonstrate that we are able to successfully demix most of the sources with NMF. In one challenging case,

where the sources were touching one another below a scattering layer, we show that NMF retrieved 20 of the 26 sources with high fidelity. In another extreme case, we included a dominant neuropil-like background signal to compete against the signal from the target sources (mimicking activity from neuronal cell bodies). We designed this “neuropil-like” signal to be a surrounding, non-sparse, and dynamic fluorescent background emitted from sources that were in the vicinity or even aggregating with the target sources (Figure 02.4d), with a total amplitude that was 6 times larger than the ensemble signal from the target sources. Despite these stringent conditions, we demonstrated that a general unconstrained NMF algorithm could successfully demix the signals from most of the target sources (9 out of 10). Therefore, this work opens a promising direction to improve fiber photometry fluorescence experiments by reaching single-source temporal activity resolution.

Lines 390-412:

"To apply this method in vivo, a few difficulties could arise, but we think they can be circumvented:

1- Fluorescence background. In calcium imaging experiments, there are two types of background: a static background (corresponding to the neurons resting fluorescence F_0) and a dynamic background (typically the activity of the neuropil). In our experiments, we didn't model the resting fluorescence of the neurons ($F_0=0$). However, new GECI indicators such as GCaMP8 have low resting fluorescence and show large transient signals, with a $\Delta F/F_0$ corresponding to one action potential ranging between 40% and 100% depending on the variant used⁷³. Therefore, this static background should remain moderate, and we expect NMF to be able to extract it (for example using rank-1 matrix factorization, see supp info of⁷⁹). On the other hand, dynamic background should be less straightforward to subtract. In this work, to evaluate if NMF could find and remove a non-sparse dynamic background component from our current signal, we included one (Figure 02.2 and Figure 02.3) or several (Figure 02.4, Figure S08, and Figure S09) additional sources to model a neuropil signal. These sources were chosen to have a spatial fingerprint overlapped with many other sources in the FoV, and to exhibit a non-sparse, fast oscillating signal, with amplitudes on the same order of magnitude as the transient peaks of the other individual sources. In the two experiments with one neuropil source, we first showed that a simple NMF algorithm could successfully retrieve more than 20 out of 26 spatiotemporal sources from the sample, including the neuropil-like source. Then, in a more challenging experiment (Figure 02.4, Figure S08, and Figure S09) where 11 out of 21 sources were mimicking synchronous neuropil, NMF successfully demixed 9 out of 10 remaining target sources with 86% average temporal correlation accuracy. These results suggest that our method could be suitable to extract activity from individual neurons in conditions where the average neuropil fluorescence is several times larger in amplitude than the remaining ensemble signal from the neurons in the FoV.

In addition, previous work from the team addressed NMF performance to extract activity from target sources in strong fluorescence background^{51,52}. NMF performance was evaluated by quantifying the cross-correlation statistics of the retrieved time traces over different experimental background conditions. Typically, NMF performance starts to slowly get impaired when the max value of the background signal is 1.5x bigger than the max value of the sources' activity time trace signal. In this case, the median temporal cross-correlation gets lower than 80%, with the first quartile of ~65%. Nevertheless, even when the max background over the max activity ratio is of 2.0, the median cross-correlation in the time trace is still above 70%, with the first quartile above 60%⁵¹. “

The changes in the SUPPLEMENTARY INFORMATION:

- We included 2 new sections in the supp info (contents)

"Supplementary materials - list of contents:

- Supplementary Information 01 – The proof of principle setup
- Supplementary Information 02 – Estimating the number of fluorescence sources with NMF: the NMF rank study
- Supplementary Information 03 – The miniscope detection sensitivity to a single source
- Supplementary Information 04 – Pattern shape dependence on symmetrically positioned beads (with miniscope)
- Supplementary Information 05 – NMF analysis test when binning the image

- Supplementary Information 06 – Comparing GT-NMF and GT-GT correlation coefficients
- Supplementary Information 07 – NMF denoising effect on spatial fingerprints
- Supplementary Information 08 – Neuropil experiment results
- Supplementary Information 09 – Neuropil experiment estimated GT ranks: the fidelity plots
- Supplementary Information 10 – Scattering properties of Parafilm M®
- Supplementary Information 11 – Number of available scattering fingerprints (sources) to be demixed”

Supp info 08: Neuropil experiment results

Figure S08 – Neuropil experiment results. NMF demixing with an increasing number of sources simulating neuropil signal. (a,b,c) The distal FOV with target sources (in white) and neuropil sources generating a dynamic background (in orange). The beads' indices are sorted from the highest to the lowest correlation between GT and NMF time traces. (d,e,f) Ensemble temporal signal (as in fiber photometry), (g,h,i) GT-NMF temporal correlations, (j,k,l) GT-GT temporal correlation of a proof-of-principle experiment designed with N=1, N=5, and N=11 beads (respectively) generating neuropil signal of out all the 21 sources. (m) Ground truth excitation photon budget of the experiment with N=11 neuropil sources over

the video frames. In blue, the ensemble excitation time trace of 10 target sources added together; in red, the ensemble excitation time trace of the 11 neuropil sources; in black, the total ensemble excitation number of photons (sum of blue and red profiles). The average number of photons delivered to the neuropil sources (red trace mean value) was 86% of the average number of photons delivered to all beads (black trace mean value).

Supp info 09: Neuropil experiments estimated GT ranks: the fidelity plots

Fidelity plots

Figure S09 – Fidelity plots for different experiments performed with various number of neuropil sources, representing an estimation of NMF residual error as a function of the factorization rank (rank = 3, 4, 5, ..., 32). Not-normalized (top) and normalized (bottom) residuals of 3 proof-of-principle experiments performed with 21 beads, including N=1 (blue), N=5 (green), N=11 (red) neuropil sources, estimated with Frobenius norm (F) in the unconstrained NMF loss function. The inset in the normalized plot is the zoom of the highlighted rectangular region. The dashed-pointed horizontal line in the normalized plot (bottom), with value ~ 0.0321 taken as a common threshold in the normalized residuals over the 3 experiments (N=1, 5, and 11) (see below). The rank corresponding to this arbitrary threshold is a good estimation of the number of the well-retrieved target

sources in the experiments. For example: 13 or 14 well-retrieved targets for $N=1$ (see also Figure S08.g), 11 or 12 targets for $N=5$ (see also Figure S08.h), and 10 of 11 targets for ($N=11$, see also figure S08.i)). The value 0,0321 is the exact residual of the rank 10 in the red curve ($N=11$) and it was arbitrarily chosen to represent the threshold line. A subsequent and more detailed inspection of the patterns and individual time traces obtained with these estimated GT ranks would help to conclude the exact rank. For example, patterns with 2 clear defined rings are only generated by 2 sources at different radii, and time traces with non-realistic GECl profiles (baseline with negative peaks) should be disregarded in the analysis.

ANSWERING POINT-BY-POINT TO REVIEWERS COMMENTS

Reviewer #1 (Remarks to the Author):

I would suggest the revised manuscript for the publication. Although there is no biological demonstration the idea is very interesting. There is also an appeal to a wider imaging community in bio/ neuroscience.

I have few more comments that might be helpful for the authors.

1. Please revise the newly written text, avoid jargon expressions as line 424: "keeping the algorithms as "vanilla" as possible".

We thank the reviewer #1 for considering our manuscript for the publication. We substituted the term "vanilla" in the main text by "standard version" and the sentence became:

Lines 475-476:

*"Importantly, we wanted to establish as clearly as possible a general proof-of-principle for the technique, focusing on a solid ground truth, keeping the algorithm in **its standard version (as general as possible)**, so as to underlay the physical concepts."*

2. I do not really agree with the authors model of the neuropil background (line 203). Neuropil is not only temporally but also spatially distributed signal (often highly correlated), fluorescence in one bead could hardly account for it. Please see the example of neuropil from miniscope recording here: <https://www.youtube.com/watch?v=RslP-WQUIK8>. The signal is bright and is distributed across the field of view. Usually spatial filtering is used to get rid of neuropil, in addition to extracting neurons as rois.

The concerns of reviewer#1's about how we modeled the neuropil are valid. Indeed, the neuropil signal is a fluctuating fluorescence background signal emitted by cellular processes surrounding the soma (neuron cell body), and is typically very strong in one photon (1P) calcium imaging experiments *without* soma-tag labeling - making it difficult to extract signals from the somata. Hence, we followed the reviewer's recommendations and designed a more realistic model of the neuropil in new set of experiments.

In our new set of experiments, the total GT neuropil signal was mimicked to be much stronger than the signal of the individual target bead, and the chosen neuropil sources were spatially distributed around the FoV in the close neighborhood of the target beads (spatially aggregated sources). We progressively increased the number of sources mimicking the same non-sparse neuropil-like temporal activity, thus progressively increasing the strength of the neuropil signal. More specifically, we chose a new FoV containing 21 beads, and we selected 1, 5, or 11 sources having the exact same neuropil-like signal (see figures 02.4, S08, and S09). Even in the most extreme case where 11 sources were mimicking neuropil (see Figure 02.4), the unconstrained NMF algorithm was able to successfully demix the signal of 9 out of 10 remaining target sources with very high accuracy (temporal correlation with GT time traces larger than 80%, with an average value of 86% and standard deviation of 5.5%, see Figures 02.4). In this case, the signal from the total neuropil-like background was approximately 6x stronger on average than the whole ensemble target signal (see Figure S08m), and approximately 10x larger than the maximum peak of each target source signal.

We believe that these new *in vitro* results not only validate the idea of the method in a very challenging scenario using an unconstrained NMF algorithm, but are compelling for future *in vivo* tests, especially if one considers using a soma-tag labeling strategy where the neuropil fluorescence should remain quite low. For other cases where one does not want to use soma-tag

labeling, a modified version of NMF (a constrained NMF, CNMF) could eventually be tested, since it has already been shown in the literature to demix neuronal target signals hidden by a strong neuropil dynamic background in 1P calcium imaging experiments.

We added a whole new section (2.4) in the main text of the manuscript describing this new experiment (see above), we extended the discussion section considering these results.

- **In between lines 366-372, we added:**

"In another extreme case, we included a dominant neuropil-like background signal to compete against the signal from the target sources (mimicking activity from neuronal cell bodies). We designed this "neuropil-like" signal to be a surrounding, non-sparse, and dynamic fluorescent background emitted from sources that were in the vicinity or even aggregating with the target sources (Figure 02.4d), with a total amplitude that was 6 times larger than the ensemble signal from the target sources. Despite these stringent conditions, we demonstrated that a general unconstrained NMF algorithm could successfully demix the signals from most of the target sources (9 out of 10)."

- **In between lines 394-410, we added:**

"On the other hand, dynamic background should be less straightforward to subtract. In this work, to evaluate if NMF could find and remove a non-sparse dynamic background component from our current signal, we included one (Figure 02.2 and Figure 02.3) or several (Figure 02.4, Figure S08, and Figure S09) additional sources to model a neuropil signal. These sources were chosen to have a spatial fingerprint overlapped with many other sources in the FoV, and to exhibit a non-sparse, fast oscillating signal, with amplitudes on the same order of magnitude as the transient peaks of the other individual sources. In the two experiments with one neuropil source, we first showed that a simple NMF algorithm could successfully retrieve more than 20 out of 26 spatiotemporal sources from the sample, including the neuropil-like source. Then, in a more challenging experiment (Figure 02.4, Figure S08, and Figure S09) where 11 out of 21 sources were mimicking synchronous neuropil, NMF successfully demixed 9 out of 10 remaining target sources with 86% average temporal correlation accuracy. These results suggest that our method could be suitable to extract activity from individual neurons in conditions where the average neuropil fluorescence is several times larger in amplitude than the remaining ensemble signal from the neurons in the FoV."

In addition, previous work from the team addressed NMF performance to extract activity from target sources in strong fluorescence background^{51,52}. NMF performance was evaluated by quantifying the cross-correlation statistics of the retrieved time traces over different experimental background conditions. Typically, NMF performance starts to slowly get impaired when the max value of the background signal is 1.5x bigger than the max value of the sources' activity time trace signal. In this case, the median temporal cross-correlation gets lower than 80%, with the first quartile of ~65%. Nevertheless, even when the max background over the max activity ratio is of 2.0, the median cross-correlation in the time trace is still above 70%, with the first quartile above 60%⁵¹. "

3. I am not sure prafilm (scattering) brakes the symmetry of the fiber (line 205), instead it modifies the wavefront such that it changes the composition of modes the fluorescence pattern is coupled to.

The scattering media does not break the symmetry of optical fiber, but it does break the symmetry of the whole optical system (as it was written in the text). Thus, it is expected to obtain less symmetric patterns in the detection due to the presence of prafilm. To ensure that these ambiguities do not occur, we substituted the term "system" with the term "whole optical system" to better specify what we meant (line 212 of the new manuscript version).

- Before:

*"As expected, the scattering patterns from this experiment (Figure 02.3) are different from the previous cases (Figure 02.1 and Figure 02.2). They became less symmetric, confirming that the scattering layer is breaking the cylindrical symmetry of the **system** (which should facilitate NMF demixing), but spatially noisier (i.e., with decreased intensity contrast)."*

- After:

"As expected, the scattering patterns from this experiment (Figure 02.3) are different from the previous cases (Figure 02.1 and Figure 02.2). They became less symmetric (which facilitates NMF demixing) and spatially noisier (i.e., with decreased intensity

contrast), reflecting the addition of an element in the optical pathway (the parafilm layer) that scatters light and does not exhibit cylindrical symmetry”

Reviewer #2 (Remarks to the Author):

I would like to thank the authors for the articulated response they gave to the referees comments. I appreciate that, based on our suggestions, they decided to add more experiments and discussions within the paper.

I believe that the neuropil like experiment and the scattering through the parafilm, add valuable insight to evaluate the potential of the method.

I also think we could debate forever if these results can or cannot be extrapolated to real in vivo imaging of calcium activity. Only one such experiment will give the final answer. My idea is still that in vivo imaging will be much more challenging. I've been doing in vivo calcium imaging myself for many years now and I can guarantee that the signals are much noisier and **have much stronger backgrounds than what the authors consider.**

However, because I believe strategies similar to the one presented in this paper can really make the calcium imaging community leap forward, I am ready to recommend publication, but I would still ask a couple of changes and one experiment, which I believe should not be too difficult for the authors to perform.

Thanks to Fig.2 and 3 we now know that background signals can confuse the algorithm, especially when it is **close to neurons and in the presence of scattering.** Now, 1P widefield calcium imaging in the cortex for example, is characterized by very high background fluorescence. Look for instance at the videos of this recent paper 'Rapid detection of neurons in widefield calcium imaging datasets after training with synthetic data'. There is a uniform fluorescence background everywhere in the raw data, which also can vary in time, and then dim cells appear on top. GCaMP8 might make the game a bit easier, but widefield 1P excitation is bound to give high background. Even if we assume that the authors can demix well cells at different planes thanks to the scattering effect that would change the look of the fingerprint (nice idea), the neuropil and resting fluorescence of certain cells will be a much stronger problem, I believe, than what the authors assume. **The neuropil is not just one point in the field of view, but it is rather a distributed background.** Not even in 2P microscopy you find areas on one plane that have zero counts.

My recommendations to the authors are thus the following:

Please discuss more and better this problem from the beginning (and again in the discussion). When you describe the proof of principle experiment, it is already the moment to say that it does not take into account several aspects that would make the process less reliable in vivo. 1) **The time traces used are not raw time traces, but $\Delta F/F$ in which F_0 is set to zero;** 2) The effect of widefield excitation that could excite at the same time hundreds (or thousands) of cells at different planes, with non-zero resting fluorescence and a distributed neuropil effect is only marginally taken into account.

We appreciate very much the constructive criticism of the reviewer #2.

Following the reviewer recommendation, we included in the result section (section 2.1) a few sentences detailing the GT time traces, and in particular emphasizing that the resting fluorescence is set to zero. :

- **Main text, Results Section 2.1, Lines 109-113:**

"The controlled excitation guarantees that each fluorescent source generates a fingerprint pattern whose intensity transiently fluctuates accordingly with the chosen GT time trace profile (see transient patterns in Figure O1). We designed GT time traces to be equivalent to optical recording experiments of GECI time traces where calcium signals had F_0 set to zero. We then recorded a video of the transient patterns that emerge from a short multimode fiber and applied NMF to the recorded raw data without doing any pre-processing step. In other sets of experiments later, we selected one or more sources available in the FoV to mimick neuropil signal, by exciting them using a non-sparse GT signal (see sections below) "

We had already mentioned that the resting fluorescence is equal to zero in the discussion following the previous revision : "In our experiments, we didn't model the resting fluorescence of the neurons ($F_0=0$)."

Regarding the second point, we agree with the reviewer that it would be very challenging to apply our method in situations where hundreds or thousands of neurons would be labeled within the excitation volume of the fiber. Instead, we would advise designing the first biological experiments so that only 20 to 30 neurons are labeled within this volume (by adjusting the sparsity of expression). In this case, we expect that spatiotemporal unmixing of most of the sources should be possible. Indeed, as we emphasized in the previous round of revisions, neurons distributed in the 3D scattering tissue should produce unique fingerprint patterns at the camera, and should therefore be properly unmixed, similar to what we demonstrated in *in vitro* proof-of-principle experiments. We would like to emphasize that this situation is very different from imaging with standard widefield microscopes, where neurons that are away from the imaging plane appear blurred on the camera and cannot be separated from each another using their spatial signature: in that case, the signal from all these out-of-focus neurons mixes up into a common background that dominates the signal from the in-focus neurons, and cannot be unmixed. With our method, on the other hand, we should be able to unmix neurons independently of their position in the 3D volume, as long as the labeling is sparse enough. We have added a sentence in the discussion to clarify this point :

Lines 419-423:

"Finally, when designing in vivo experiments, we would advise using GCaMP8 indicators targeted to the soma⁸¹ to minimize the neuropil component and therefore facilitate signal analysis. In the first experiments, sparsity of expression could be adjusted so that only 20 to 30 neurons are labeled within the illumination volume of the fiber. In this case, we expect that spatiotemporal unmixing of most of the sources should be possible. Indeed, neurons distributed in the 3D scattering tissue should produce unique fingerprint patterns at the camera and should therefore be properly unmixed, similar to what we demonstrated in in vitro proof-of-principle experiments."

I typically would not ask more experiments in a second round of revision, but I think there is clearly one more important point that can be added to Fig.3. What happens if you now, on the same FOV, start to add more independent neuropil-like components? My idea of this experiment is the following. Take a sample with the maximum number of beads that you can find. Select 20-30 beads as 'neurons' and ~ 5 as neuropil, in different **positions, not very far from the neuron beads**. Now run the demixing routine while only sending neuronal signals. You should get all the 20-30 neurons. Next, start one by one to add the neuropils and see each time how many neurons traces you actually loose. **This experiment is not designed to say that the method is wrong if after 5 neuropils no neurons is found anymore**. My publication recommendation will not change depending on the output of the experiment. But it will add all the context necessary so that people will be more able to understand if it could or not work for their applications.

I believe that after this new round of revision the paper will be ready for publication.

As we answer to reviewer #1, the concerns about how we modeled the neuropil are valid, despite the current advances in the literature on how to could significantly overcome such limitations experimentally (with soma-tag), and computationally (with constrained NMF). Indeed, neuropil is not an easy challenge to overcome in calcium imaging experiments. Therefore, we decided to follow the reviewer's recommendation and perform a new set of experiments to demonstrate that the idea of the *proof of principle* still holds true for a condition where neuropil is more realistically mimicked (i.e., stronger and coming from different regions of the FoV, but still close to the target sources).

As one can see in the new section (2.4) in the main text of the manuscript, and in the answer to reviewer #1, we designed a new set of experiments in which aggregated sources to the target ones at different regions in the FoV were selected to act as neuropil sources. We progressively increased the number of neuropil sources (1, 5, and 11) compared to the total number of sources (21) so that around half of the sources had a synchronous non-sparse neuropil-like activity (a strong fluctuating background that was generated in the right vicinity of all remaining 10 target sources), while other sources remained as target sources. In this extreme case, the total signal of the neuropil sources is 6 times stronger than the total signal from the target

sources. Nevertheless, we demonstrated that unconstrained NMF was still able to retrieve individual time traces of the 9 of the 10 target sources with a surprisingly high temporal correlation accuracy ($>80\%$ and with an average value of 86%). There is no doubt that this performance can be even further improved with a constrained NMF as it has already been shown in the literature. We described these results in more details at the beginning of this document, and listed all the changes made to the main text and supplementary data of the manuscript.

Reviewer #3 (Remarks to the Author):

The rebuttal was very thorough and addressed all main concerns regarding the presentation, claims and discussion. I support publication.

The authors would like to thank the reviewer#3 for the positive assessment of our manuscript and his constructive criticism during the revision process.

REVIEWERS' COMMENTS

Reviewer #1 (Remarks to the Author):

Thank you for considering more challenging conditions to mimic neuropil signal. I recommend the revised manuscript for publication.

Reviewer #2 (Remarks to the Author):

I would like to thank the authors for the response to the previous round of review. Indeed the new set of experiments are much more convincing and I would now like to recommend the manuscript for publication.

I am still looking forward to see some of these methods used for real in vivo experiments. I hope the authors will eventually get to that point.

For the moment, the current manuscript constitutes a nice proof of principle and new experimental concept. The potential to improve greatly fiber photometry experiments is demonstrated, let's hope this will really be used in the future.